# LABEL SMOOTHING IMPROVES MACHINE UNLEARN-ING

**Zonglin Di**[1]**, Zhaowei Zhu**[2]**, Jinghan Jia**[3*]**, Jiancheng Liu**[3*]**, Zafar Takhirov**[4]**, Bo Jiang**[4]**,
Yuanshun Yao**[5]**, Sijia Liu**[3]**, Yang Liu**[1†]

[1]University of California, Santa Cruz, [2]Docta.ai, [3]Michigan State University,
[4]Independent Contributor, [5]Meta SuperIntelligence Lab
[1]{zdi, yangliu}@ucsc.edu, [2]zzw@docta.ai,
[3]{jiajingh, liujia45, liusiji5}@msu.edu,
[4]{z.tahirov, bjiang518}@gmail.com, [5]kevinyaowork@gmail.com

## ABSTRACT

The objective of machine unlearning (MU) is to eliminate previously learned data from a model. However, it can be challenging to strike a balance between computation cost and performance when using existing MU techniques. Taking inspiration from the influence of label smoothing on model confidence and differential privacy, we propose a simple gradient-based MU approach that uses an inverse process of label smoothing. This work introduces UGradSL, a simple, plug-and-play MU approach that uses smoothed labels. We provide theoretical analyses demonstrating why properly introducing label smoothing improves MU performance. We conducted extensive experiments on several datasets of various sizes and different modalities, demonstrating the effectiveness and robustness of our proposed method. UGradSL is also closely connected to improving the local differential privacy. The consistent improvement in MU performance is only at a marginal cost of additional computations. For instance, UGradSL consistently outperforms the gradient-ascent MU baseline on different unlearning tasks without sacrificing unlearning efficiency. A self-adaptive UGradSL is also given for simple parameter selection. The code is available at https://github.com/UCSC-REAL/Label-Smoothing-Unlearn.

## 1 INTRODUCTION

Building a reliable ML model has become an important topic in this community. Machine unlearning (MU) is a task requiring removing the learned data points from the model. The concept and the technology of MU enable researchers to delete sensitive or improper data in the training set to improve fairness, robustness, and privacy and get a better ML model for real-world use (Chen et al., 2021b; Sekhari et al., 2021). Retraining from scratch (Retrain) is a straightforward method when we want to remove the data from the model; yet it incurs prohibitive computation costs for large models due to computing resource constraints. Therefore, an efficient and effective MU method is desired.

The most straightforward MU approach should be a retraining-based method (Bourtoule et al., 2021), meaning that we retrain the model from scratch without using the data to be forgotten. The method can guarantee privacy protection but the computational cost is intensive. Most existing works (Koh & Liang, 2017; Golatkar et al., 2020; Warnecke et al., 2021; Graves et al., 2021; Thudi et al., 2021; Izzo et al., 2021; Becker & Liebig, 2022; Jia et al., 2023) focus on *approximate MU* to achieve a balance between unlearning efficacy and computational complexity, making them more suitable for real-world applications, meaning that they make the model unlearn the forgetting dataset without retraining the model.

We desire an approach that enjoys both high performance and high efficiency. Since MU can be viewed as the inverse process of ML, we are motivated to think it would be a natural and efficient

---

*Equal Contribution.
†Corresponding author: Yang Liu <yangliu@ucsc.edu>

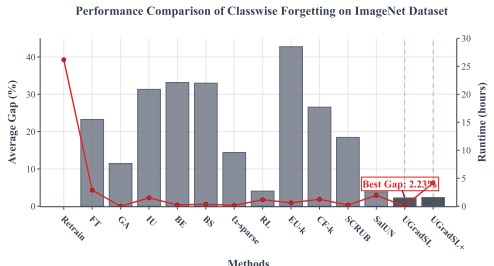 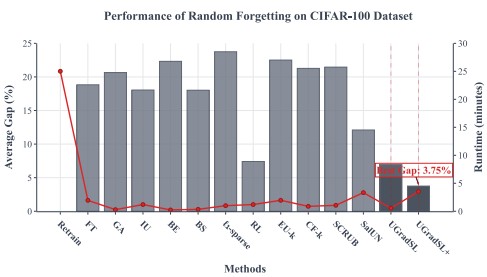

(a) Performance of classwise forgetting on ImageNet.    (b) Performance of random forgetting on CIFAR-100.

Figure 1: The performance comparison of our proposed methods and baseline methods using average gap and runtime (RTE), where lower values indicate better performance. Bars represent average gap while red dotted lines show RTE. Since retraining does not have gap by definition, only RTE is reported for this baseline and the bar is empty. For classwise forgetting on ImageNet, UGradSL achieves the lowest average gap (2.23%) with acceptable RTE increase. For random forgetting on CIFAR-100, UGradSL+ attains the best average gap (3.75%), while UGradSL demonstrates an optimal gap-runtime trade-off.

way to develop an unlearning process that imitates the reverse of gradient descent. Indeed, gradient ascent (GA) (Thudi et al., 2021) is one of the MU methods but unfortunately, it does not fully achieve the potential of this idea. One of the primary reasons is that once the model completes training, the gradient of well-memorized data that was learned during the process diminishes (close to 0 loss) and therefore the effect of GA is rather limited.

Our approach is inspired by the celebrated idea of label smoothing (Szegedy et al., 2016). In the forward problem (gradient descent), the smoothed label proves to be able to improve the model's generalization power. In our setting, we treat the smoothed label term as the regularization in the loss function, making the unlearning more controllable. Specifically, we show that GA with a "negative" label smoothing process (which effectively results in a standard label smoothing term in a descending fashion) can quickly improve the model's deniability in the forgetting dataset, making the model behave close to the retrained model, which is exactly the goal of MU. We name our approach *UGradSL*, **U**nlearning using **Gra**dient-based **S**moothed **L**abels.

Our approach is a plug-and-play method that can improve the gradient-based MU performance consistently and does not hurt the performance of the remaining dataset and the testing dataset in a gradient-mixed way. At the same time, we provide a theoretical analysis of the benefits of our approach for the MU task. The core contributions of this paper are summarized as follows:

- We propose a lightweight tool to improve MU by joining the label smoothing and gradient ascent.
- We theoretically analyze the role of gradient ascent in MU and how negative label smoothing is able to boost MU performance.
- Extensive experiments in six datasets in different modalities and several unlearning paradigms regarding different MU metrics show the robustness and generalization of our method.
- We investigate the relationship between label smoothing and label differential privacy (LDP), showing that label smoothing can aid LDP.

## 2 RELATED WORK

**Machine Unlearning** (MU) was developed to address information leakage concerns related to private data after the completion of model training (Cao & Yang, 2015; Bourtoule et al., 2021; Nguyen et al., 2022), gained prominence with the advent of privacy-focused legislation (Hoofnagle et al., 2019; Pardau, 2018). One direct unlearning method involves retraining the model from scratch after removing the forgetting data from the original training set. It is computationally inefficient, prompting researchers to focus on developing approximate but much faster unlearning techniques (Becker & Liebig, 2022; Golatkar et al., 2020; Warnecke et al., 2021; Graves et al., 2021; Thudi et al., 2021; Izzo et al., 2021; Jia et al., 2023). Beyond unlearning methods, other research efforts aim to create probabilistic unlearning concepts (Ginart et al., 2019; Guo et al., 2019; Neel et al.,

2021; Ullah et al., 2021; Sekhari et al., 2021) and facilitate unlearning with provable error guarantees, particularly in the context of differential privacy (DP) (Dwork et al., 2006; Ji et al., 2014; Hall et al., 2012). However, it typically necessitates stringent model and algorithmic assumptions, potentially compromising effectiveness against practical adversaries, such as membership inference attacks (Graves et al., 2021; Thudi et al., 2021). Additionally, the interest in MU has expanded to encompass various learning tasks and paradigms (Wang et al., 2022b; Liu et al., 2022b; Chen et al., 2022; Chien et al., 2022; Marchant et al., 2022; Di et al., 2022). These applications demonstrate the growing importance of MU techniques in safeguarding privacy. The rest of the related work about influence function label smoothing and differential privacy are given in Appendix B.

## 3 Label Smoothing Enables Fast and Effective Unlearning

This section sets up the analysis and shows that properly performing label smoothing enables fast and effective unlearning. The key ingredients of our approach are gradient ascent (GA) and label smoothing (LS). We start with understanding how GA helps with unlearning and then move on to show the power of LS. At the end of the section, we formally present our algorithm.

### 3.1 Preliminary

**Machine Unlearning** Consider a $K$-class classification problem on the training data distribution $\mathcal{D}_{tr} = (\mathcal{X} \times \mathcal{Y})$, where $\mathcal{X}$ and $\mathcal{Y}$ are the feature and label space, respectively. Due to some privacy regulations, there exists a forgetting data distribution $\mathcal{D}_f$ that the model needs to unlearn. We denote by $\boldsymbol{\theta}_{tr}$ the original model trained on $\mathcal{D}_{tr}$ and $\boldsymbol{\theta}_u$ the model without the influence of $\mathcal{D}_f$. The goal of machine unlearning (MU) is how to generate $\boldsymbol{\theta}_u$ from $\boldsymbol{\theta}_{tr}$.

**Label Smoothing** In a $K$-class classification task, let $\boldsymbol{y}_i$ denote the one-hot encoded vector form of $y_i \in \mathcal{Y}$. Similar to (Wei et al., 2021), we unify positive label smoothing (PLS) and negative label smoothing (NLS) into generalized label smoothing (GLS). The random variable of smoothed label $\boldsymbol{y}_i^{\text{GLS},\alpha}$ with smooth rate $\alpha \in (-\infty, 1]$ is $\boldsymbol{y}_i^{\text{GLS},\alpha} = (1 - \alpha) \cdot \boldsymbol{y}_i + \frac{\alpha}{K} \cdot \mathbf{1} = [\frac{\alpha}{K}, \cdots, \frac{\alpha}{K}, (1 + \frac{1-K}{K}\alpha), \frac{\alpha}{K}, \cdots, \frac{\alpha}{K}]$, where $(1 + \frac{1-K}{K}\alpha)$ is the $y_i$-th element in the encoded label vector. When $\alpha < 0$, GLS becomes NLS.

### 3.2 Gradient Ascent Can Help Gradient-Based Machine Unlearning

We discuss three sets of model parameters in the MU problem: 1) $\boldsymbol{\theta}_{tr}^*$, the optimal parameters trained from $D_{tr} \sim \mathcal{D}_{tr}$, 2) $\boldsymbol{\theta}_r^*$, the optimal parameters trained from $D_r \sim \mathcal{D}_r$, such that $D_r = D_{tr} \backslash D_f$ and 3) $\boldsymbol{\theta}_f^*$, the optimal parameters unlearned using gradient ascent (GA) on $D_f \sim \mathcal{D}_f$. Note $\boldsymbol{\theta}_r^*$ can be viewed as the *exact* MU model. The definitions of $\boldsymbol{\theta}_{tr}^*$ and $\boldsymbol{\theta}_r^*$ follow the standard empirical risk minimization as

$$\boldsymbol{\theta}^* = \arg\min_{\boldsymbol{\theta}} \frac{1}{n} \sum_{z \in D} \ell(h_{\boldsymbol{\theta}}, z). \tag{1}$$

and by using the influence function, $\boldsymbol{\theta}_f^*$ is

$$\boldsymbol{\theta}_f^* = \arg\min_{\boldsymbol{\theta}}\{R_{tr}(\boldsymbol{\theta}) + \varepsilon \sum_{z^f \in D_f} \ell(h_{\boldsymbol{\theta}}, z^f)\}$$

where $R_{tr}(\boldsymbol{\theta}) = \sum_{z^{tr} \in D_{tr}} \ell(h_{\boldsymbol{\theta}}, z^{tr})$ and $R_f(\boldsymbol{\theta}) = \sum_{z^f \in D_f} \ell(h_{\boldsymbol{\theta}}, z^f)$ are the empirical risk on $D_{tr}$ and $D_f$, respectively. We use notations $\ell(h_{\boldsymbol{\theta}}, z)$ to specify the loss of an example $z = (x, y)$ in the dataset. $h_{\boldsymbol{\theta}}$ is a function $h$ parameterized by $\boldsymbol{\theta}$. $\varepsilon$ is the weight of $D_f$ compared with $D_{tr}$. The optimal parameter can be found when the gradient is 0:

$$\nabla_{\boldsymbol{\theta}} R_{tr}(\boldsymbol{\theta}_f^*) + \varepsilon \sum_{z^f \in D_f} \nabla_{\boldsymbol{\theta}} \ell(h_{\boldsymbol{\theta}_f^*}, z^f) = 0. \tag{2}$$

Expanding Eq. (2) at $\boldsymbol{\theta} = \boldsymbol{\theta}_{tr}^*$ using the Taylor series, we have

$$\boldsymbol{\theta}_f^* - \boldsymbol{\theta}_{tr}^* \approx -\left[\sum_{z^{tr} \in D_{tr}} \nabla_{\boldsymbol{\theta}}^2 \ell(h_{\boldsymbol{\theta}_{tr}^*}, z^{tr}) + \varepsilon \sum_{z^f \in D_f} \nabla_{\boldsymbol{\theta}}^2 \ell(h_{\boldsymbol{\theta}_{tr}^*}, z^f)\right]^{-1} \left(\varepsilon \sum_{z^f \in D_f} \nabla_{\boldsymbol{\theta}} \ell(h_{\boldsymbol{\theta}_{tr}^*}, z^f)\right). \tag{3}$$

Here, we ignore the Lagrange Remainder. Similarly, we can expand $\nabla_{\boldsymbol{\theta}} R_{tr}(\boldsymbol{\theta}_{tr}^*)$ at $\boldsymbol{\theta} = \boldsymbol{\theta}_r^*$ and derive $\boldsymbol{\theta}_r^* - \boldsymbol{\theta}_{tr}^*$ as

$$\boldsymbol{\theta}_r^* - \boldsymbol{\theta}_{tr}^* \approx \left[ \sum_{z^{tr} \in D_{tr}} \nabla_{\boldsymbol{\theta}}^2 \ell(h_{\boldsymbol{\theta}_r^*}, z^{tr}) \right]^{-1} \left( \sum_{z^{tr} \in D_{tr}} \nabla_{\boldsymbol{\theta}} \ell(h_{\boldsymbol{\theta}_r^*}, z^{tr}) \right). \tag{4}$$

We ignore the average operation in the original definition of the influence function for computation convenience because the size of $D_{tr}$ or $D_f$ is fixed. For GA, let $\varepsilon = -1$ in Eq. (3) and we have

$$\boldsymbol{\theta}_r^* - \boldsymbol{\theta}_f^* = \boldsymbol{\theta}_r^* - \boldsymbol{\theta}_{tr}^* - (\boldsymbol{\theta}_f^* - \boldsymbol{\theta}_{tr}^*) = \Delta\boldsymbol{\theta}_r - \Delta\boldsymbol{\theta}_f, \tag{5}$$

where $(-\Delta\boldsymbol{\theta}_r)$ represents the learning gap from $\boldsymbol{\theta}_r^*$ to $\boldsymbol{\theta}_{tr}^*$ while vector $\Delta\boldsymbol{\theta}_f$ represents how much the model unlearns (backtracked progress) between $\boldsymbol{\theta}_f^*$ and $\boldsymbol{\theta}_{tr}^*$. The details of $\Delta\boldsymbol{\theta}_r$ and $\Delta\boldsymbol{\theta}_f$ are given in Eq. (17) in Appendix. Ideally, when $\Delta\boldsymbol{\theta}_r$ and $\Delta\boldsymbol{\theta}_f$ are exactly the same vectors, GA can lead the model to the optimal retrained model since we have $\boldsymbol{\theta}_r^* = \boldsymbol{\theta}_f^*$. However, this condition is hard to satisfy in practice. Thus, GA cannot always help MU. We summarize it in Theorem 1. The proof and the error analysis are given in Appendix C.1 and C.2.

**Theorem 1.** *Given the approximation in Eq. (5), GA achieves exact MU if and only if*

$$\sum_{z^f \in D_f} \nabla_{\boldsymbol{\theta}} \ell(h_{\boldsymbol{\theta}_r^*}, z^f) \approx -\boldsymbol{H}(\boldsymbol{\theta}_r^*, \boldsymbol{\theta}_{tr}^*) \cdot \sum_{z^f \in D_f} \nabla_{\boldsymbol{\theta}} \ell(h_{\boldsymbol{\theta}_{tr}^*}, z^f),$$

$\boldsymbol{H}(\boldsymbol{\theta}_r^*, \boldsymbol{\theta}_{tr}^*) = \left[ \sum_{z^{tr} \in D_{tr}} \nabla_{\boldsymbol{\theta}}^2 \ell(h_{\boldsymbol{\theta}_r^*}, z^{tr}) \right] \left[ \sum_{z^r \in D_r} \nabla_{\boldsymbol{\theta}}^2 \ell(h_{\boldsymbol{\theta}_{tr}^*}, z^r) \right]^{-1}$. *Otherwise, there exist* $\boldsymbol{\theta}_r^*, \boldsymbol{\theta}_{tr}^*$ *such that GA cannot help MU, i.e.,* $\|\boldsymbol{\theta}_r^* - \boldsymbol{\theta}_f^*\| > \|\boldsymbol{\theta}_r^* - \boldsymbol{\theta}_{tr}^*\|$.

## 3.3 Label Smoothing Improves MU

Practically, we cannot guarantee that GA always helps MU as shown in Theorem 1. To alleviate the possible undesired effect of GA, we propose to use label smoothing as a plug-in module. Consider the cross-entropy loss as an example. For GLS, the loss is calculated as

$$\ell(h_{\boldsymbol{\theta}}, z^{\text{GLS},\alpha}) = \left(1 + \frac{1-K}{K}\alpha\right) \cdot \ell(h_{\boldsymbol{\theta}}, (x, y)) + \frac{\alpha}{K} \sum_{y' \in \mathcal{Y} \backslash y} \ell(h_{\boldsymbol{\theta}}, (x, y')), \tag{6}$$

where $\ell(h_{\boldsymbol{\theta}}, (x, y)) := \ell(h_{\boldsymbol{\theta}}, z)$ and $\ell(h_{\boldsymbol{\theta}}, (x, y'))$ to denote the loss of an example when its label is replaced with $y'$. Intuitively, Term $\sum_{y' \in \mathcal{Y} \backslash y} \ell(h_{\boldsymbol{\theta}}, (x, y'))$ in Eq. (6) leads to a state where the model makes *wrong predictions on data in the forgetting dataset* with equally low confidence (Wei et al., 2021; Lukasik et al., 2020).

With smoothed label given in Eq. (6), we show that there exists a vector $\Delta\boldsymbol{\theta}_n$ such that Eq. (5) can be written as

$$\boldsymbol{\theta}_r^* - \boldsymbol{\theta}_{f,\text{LS}}^* \approx \Delta\boldsymbol{\theta}_r - \Delta\boldsymbol{\theta}_f + \frac{1-K}{K}\alpha \cdot (\Delta\boldsymbol{\theta}_n - \Delta\boldsymbol{\theta}_f), \tag{7}$$

We leave the detailed form of $\Delta\boldsymbol{\theta}_n$ to Eq. (34). But intuitively, $\Delta\boldsymbol{\theta}_n$ captures the gradient influence of the smoothed non-target label on the weight. We show the effect of NLS ($\alpha < 0$) in Theorem 2 below and its proof is given in Appendix C.3.

**Theorem 2.** *Given the approximation in Eq. (5) and* $\langle \Delta\boldsymbol{\theta}_r - \Delta\boldsymbol{\theta}_f, \Delta\boldsymbol{\theta}_n - \Delta\boldsymbol{\theta}_f \rangle \leq 0$*, there exists an* $\alpha < 0$ *such that NLS improves GA in unlearning, i.e.,* $\|\boldsymbol{\theta}_r^* - \boldsymbol{\theta}_{f,\text{NLS}}^*\| < \|\boldsymbol{\theta}_r^* - \boldsymbol{\theta}_f^*\|$*, where* $\boldsymbol{\theta}_{f,\text{NLS}}^*$ *is the optimal parameters unlearned using GA and NLS, and* $\langle \cdot, \cdot \rangle$ *the inner product of two vectors.*

Now we explain the above theorem intuitively. Vector $\Delta\boldsymbol{\theta}_f - \Delta\boldsymbol{\theta}_r$ is the resultant of Newton's direction of learning and unlearning. Vector $\Delta\boldsymbol{\theta}_f - \Delta\boldsymbol{\theta}_n$ is the resultant of Newton's direction of learning non-target labels and unlearning the target label. When the condition $\langle \Delta\boldsymbol{\theta}_r - \Delta\boldsymbol{\theta}_f, \Delta\boldsymbol{\theta}_n - \Delta\boldsymbol{\theta}_f \rangle \leq 0$ holds, $\Delta\boldsymbol{\theta}_n - \Delta\boldsymbol{\theta}_f$ captures the effects of the smoothing term in the unlearning process. If we assume that the exact MU model is able to fully unlearn an example, vector $\Delta\boldsymbol{\theta}_n$ contributes a direction that pushes the model closer to the exact MU state by leading the model to give the wrong prediction. The illustration of $\langle \Delta\boldsymbol{\theta}_r - \Delta\boldsymbol{\theta}_f, \Delta\boldsymbol{\theta}_n - \Delta\boldsymbol{\theta}_f \rangle$ is shown in Figure 6 in the Appendix.

The effect of the smoothed term in gradient ascent (GA) with NLS is equivalent to performing a gradient descent optimization with traditionally defined (positive) LS. The gradient of the smoothed term is exactly the same as $\alpha/K \cdot \sum_{y' \in \mathcal{Y} \backslash y} \nabla \ell(h_{\boldsymbol{\theta}}, (x, y'))$ in both cases.

### 3.4 LABEL SMOOTHING HELPS LOCAL DIFFERENTIAL PRIVACY

When $\alpha < 0$, the smoothing term will incur a positive effect in the gradient ascent (GA) step. Label smoothing can also be viewed through the lens of privacy protection. This interpretation stems from the fact that label smoothing reduces the likelihood of a specific label, thereby allowing it to better blend in with other candidate labels. Particularly, we consider a local differential privacy (LDP) guarantee for labels as follows.

**Definition 1** (Label-LDP). *A privacy protection mechanism $\mathcal{M}$ satisfies $\epsilon$-Label-LDP, if for any labels $y, y', y^{pred} \in \mathcal{Y}$, $\frac{\mathbb{P}(\mathcal{M}(y)=y^{pred})}{\mathbb{P}(\mathcal{M}(y')=y^{pred})} \le e^\epsilon$.*

The operational meaning of $\mathcal{M}$ is to guarantee any two labels $y$ and $y'$ in the label space, after privatization, have a similar likelihood to become any $y^{\text{pred}}$ in the label space. That is, the prediction on the forgetting dataset should be similar no matter what the ground-truth label is. The similarity is measured by the privacy budget $\epsilon \in [0, +\infty)$. Smaller $\epsilon$ implies stronger indistinguishability between $y$ and $y'$, and hence, stricter privacy.

Recall $R_{tr}(\boldsymbol{\theta}) = \sum_{z^{tr} \in D_{tr}} \ell(h_{\boldsymbol{\theta}}, z^{tr})$. Denote by $R_f^{\text{NLS}}(\boldsymbol{\theta}; \alpha) = \sum_{z^{\text{LS},\alpha} \in D_f} \ell(h_{\boldsymbol{\theta}}, z^{\text{LS},\alpha}), \alpha < 0$ the empirical risk of forgetting data with NLS. After MU with label smoothing on $D_f$ by GA, the resulting model can be seen as minimizing the risk $\gamma_1 \cdot R_{tr}(\boldsymbol{\theta}) - \gamma_2 \cdot R_f^{\text{NLS}}(\boldsymbol{\theta}; \alpha)$, which is a weighted combination of the risk from two phases: 1) machine learning on $D_{tr}$ with weight $\gamma_1 > 0$ and 2) machine unlearning on $D_f$ with weight $\gamma_2 > 0$. By analyzing the risk, we have the following theorem to show NLS in MU induces $\epsilon$-Label-LDP for the forgetting data.

**Theorem 3.** *Suppose $\gamma_1 - \gamma_2(1 + \frac{1-K}{K}\alpha) > 0$. MU using GA+NLS achieves $\epsilon$-Label-LDP on $D_f$ where*

$$\epsilon = \left| \log\left( \frac{K}{\alpha}\left(1 - \frac{\gamma_1}{\gamma_2}\right) + 1 - K \right) \right|, \ \alpha < 0.$$

Intuitively, when $\alpha$ is more negative, the privacy of the labels in the forgetting dataset is better. When $\alpha \to (1 - \gamma_1/\gamma_2)$, we have $\epsilon \to 0$, indicating the best label-LDP result, which is the goal of MU. The theorem also warns that $\alpha$ cannot be arbitrarily negative.

## 4 UGRADSL: A PLUG-AND-PLAY AND GRADIENT-MIXED MU METHOD

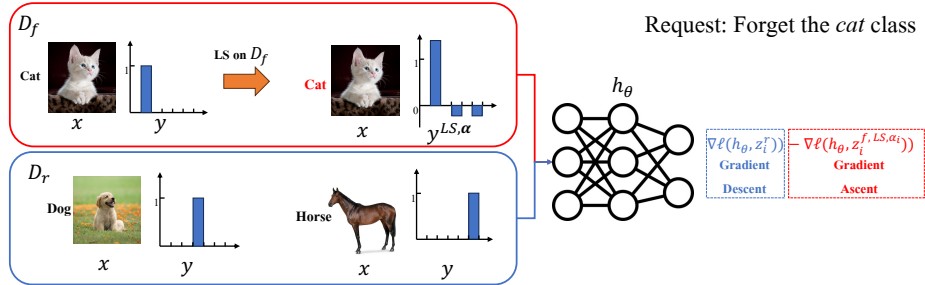

Figure 2: The framework of UGradSL. When there is an unlearning request, we can split the $D_{tr}$ into $D_f$ and $D_r$. We first apply label smoothing on $z_i^f = \{x, y\} \in D_f$ to get $z_i^{\text{LS},\alpha_i} = \{x, y^{\text{LS},\alpha_i}\}$, where the smooth rate can be pre-defined or self-adaptive. In the backpropagation process, we apply gradient descent on the data $z_i^r \in D_r$ and gradient ascent on the smoothed forgetting data $D_f$, which is the mix-gradient way.

Given the effect of label smoothing on MU and LDP, we propose our method here. Compared with retraining, Fine-Tune (FT) and GA are much more efficient as illustrated in Section 5 with comparable or better MU performance. FT and GA focus on different perspectives of MU. FT is to transfer the knowledge of the model from $D_{tr}$ to $D_r$ using gradient descent (GD) while GA is to remove the knowledge of $D_f$ from the model.

Due to the flexibility of label smoothing, our method is suitable for the gradient-based methods including FT and GA, making our method a plug-and-play algorithm. **UGradSL is based on GA while UGradSL+ is on FT. Compared with UGradSL, UGradSL+ will lead to a more comprehensive result but with a larger computation cost.**

How to choose the smooth rate $\boldsymbol{\alpha}$ is worth discussing. Normally, the $\alpha_i \in \boldsymbol{\alpha}$ for every data point $z_i^f \in D_f$ can be set identically. To gain better performance, we improve UGradSL and UGradSL+ by taking every data point into consideration and assigning $\alpha_i$ individually and adaptively based on the distance $d(h_{\boldsymbol{\theta}_f}(z_i^r), h_{\boldsymbol{\theta}_f}(z_j^f)) \in [0,1]$ for each $(z_i^r, z_i^f)$ pair. The intuition is that if an instance $z_i^f$ resides in a dense neighborhood of $D_r$, its inherent deniability is higher and therefore the requirement for "forgetting" is lower and should be reflected through a smaller $\alpha_i$. The algorithm is presented in Algorithm 1 and the framework is illustrated in Figure 2. We leave the details of the implementation, complexity analysis and the additional classification results in Appendix D.

Assuming the amount of retained data is significantly larger than the amount of data to be forgotten ($|D_r| > |D_f|$), $D_f$ will be iterated several times when $D_r$ is fully iterated once. We calculate the loss using a gradient-mixed method as:

$$
L(h_{\boldsymbol{\theta}}, B_f^{\text{NLS},\boldsymbol{\alpha}}, B_r, p) = p \cdot \sum_{z^r \in B_r} \ell(h_{\boldsymbol{\theta}}, z^r) - (1-p) \cdot \sum_{z_i^{f,\text{NLS},\alpha_i} \in B_f^{\text{NLS},\alpha}} \ell(h_{\boldsymbol{\theta}}, z_i^{f,\text{NLS},\alpha_i}) \tag{8}
$$

where $p \in [0,1]$ is used to balance GD and GA and the minus sign between two elements on the RHS stands for the GA. $\boldsymbol{\alpha}$ is the vector for the smoothing rate of every data point $z_i^f$. $h_{\boldsymbol{\theta}}$ is updated according to $L$ in Eq. (8). UGradSL is similar to UGradSL+ and the dataset used is given in bracket in Algorithm 1. The difference between UGradSL and UGradSL+ is the convergence standard. UGradSL is based on the convergence of $D_f$ while UGradSL+ is based on $D_r$. It should be noted that the Hessian matrix in Theorem 1 is only used in the theoretical proof. In the practical calculation, **there is no need to calculate the Hessian matrix**. Thus, our method does not incur substantially more computation but improves the MU performance on a large scale. We present empirical evidence in Section 5. Compared with applying the label smoothing evenly, the improved version takes the similarity of the data points between $D_r$ and $D_f$ into consideration and provides self-adaptive smoothed labels for individual $z_i^f$ as well as protects label-LDP.

## 5 EXPERIMENTS AND RESULTS

### 5.1 EXPERIMENT SETUP

**Dataset and Model Selection** We validate our method using various datasets in different scales and modalities, including CIFAR-10, CIFAR-100 (Krizhevsky et al., 2009), SVHN (Netzer et al., 2011), CelebA (Liu et al., 2015), Tiny-ImageNet, ImageNet (Deng et al., 2009) and 20 Newsgroups (Lang, 1995) datasets. For the vision and language dataset, we use ResNet-18 (He et al., 2016) and Bert (Devlin et al., 2018) as the backbone model, respectively. Due to the page limit, the details of the training parameter and the additional results of different models including VGG-16 (Simonyan & Zisserman, 2014) and vision transformer (ViT) (Dosovitskiy et al., 2020) are given in the Appendix E.5.

**Baseline Methods** We compare the proposed methods with a series of baselines, including retrain, fine-tuning (FT) (Warnecke et al., 2021; Golatkar et al., 2020), gradient ascent (GA) (Graves et al., 2021; Thudi et al., 2021), unlearning based on the influence function (IU) (Izzo et al., 2021; Koh & Liang, 2017), boundary unlearning (BU) (Chen et al., 2023), $\ell_1$-sparse (Jia et al., 2023), random label (RL) (Hayase et al., 2020), SCRUB (Kurmanji et al., 2023), SalUN (Fan et al., 2023), EU-$k$, CF-$k$ (Goel et al., 2022), GLI (Choi et al., 2024) and PABI (Koloskova et al., 2025). The implementation details of these baselines are given in Appendix E.1.

**Evaluation Metrics** The evaluation metrics we use follow (Jia et al., 2023), where we jointly consider unlearning accuracy (UA), membership inference attack (MIA), remaining accuracy (RA), testing accuracy (TA), and run-time efficiency (RTE). UA is the ratio of incorrect prediction on $D_f$, showing the MU performance. TA is the accuracy used to evaluate the performance on the whole testing set $D_{te}$, except for the class-wise forgetting because the task is to forget the specific class.

---

**Algorithm 1** UGradSL+: A plug-and-play, efficient, gradient-based MU method using LS. UGradSL can be specified by imposing the dataset replacement in brackets. If $\boldsymbol{\alpha}$ is not given, the algorithm turns to the self-adaptive version.

---

**Require:** An almost-converged model $h_{\hat{\boldsymbol{\theta}}_{tr}}$ trained with $D_{tr}$. The retained dataset $D_r$. The forgetting dataset $D_f$. Unlearning epochs $E$. GA ratio $p$. Distance threshold $\beta$. The optional smoothing ratio $\boldsymbol{\alpha}$.

**Ensure:** The unlearned model $h_{\boldsymbol{\theta}_f}$.

 1: Set the current epoch index as $t_c \leftarrow 1$
 2: **while** $t_c < E$ **do**
 3:     **while** $D_r(D_f)$ is not fully iterated **do**
 4:         Sample a batch $B_r$ in $D_r$
 5:         Sample a batch $B_f$ from $D_f$ where $|B_f| = |B_r|$
 6:         **if** $\boldsymbol{\alpha}$ is not given **then**              ▷ The improved and self-adaptive version
 7:             Extract the feature of $z_i^r$ and $z_j^f$ using $h_{\boldsymbol{\theta}_f}$.
 8:             Calculate the distance $d(h_{\boldsymbol{\theta}_f}(z_i^r), h_{\boldsymbol{\theta}_f}(z_j^f))$ for each $(z_i^r, z_j^f)$ pair where $z_i^r \in B_r$ and $z_j^f \in B_f$.
 9:             For each $z_j^f$, count the number $c_j^f$ of $z_i^r$ whose $d(h_{\boldsymbol{\theta}_f}(z_i^r), h_{\boldsymbol{\theta}_f}(z_j^f)) < \beta$
10:             Calculate the smooth rate $\alpha_j = c_j^f/|B_f|$ for each $z_j^f \in B_f$
11:         **end if**
12:         Update the model using $B_r$, $B_f$, $p$ and $\alpha_i$ according to Eq. (8)
13:     **end while**
14:     $t_c \leftarrow t_c + 1$
15: **end while**

---

RA is the accuracy on $D_r$. To evaluate the effectiveness of "forgetting", we resort to the MIA metrics described in (Jia et al., 2023; Fan et al., 2023), i.e. accuracy of an attack model against target model $\theta_u$, such that the score is reported as true negative rate (TNR) on the forget set. Formally, this is a global MIA score (Yeom et al., 2018), which we rewrite as $\text{MIA}_{\text{Score}} = 1 - \Pr(x_f|\theta_\star)$, where $x_f \in D_f$ are the forget samples and $\theta_\star$ is the model under test. Overall, we use *Avg. Gap* to quantify the mean performance gap between each unlearning method and the retrained model across all individual metrics above. A lower value indicates better performance.

**Unlearning Paradigm** We mainly consider three unlearning paradigms, including *class-wise forgetting*, *random forgetting*, and *group forgetting*. Class-wise forgetting is to unlearn the whole specific class where we remove one class in $D_r$ and the corresponding class in $D_{te}$ completely. Random forgetting across all classes is to unlearn data points belonging to all classes. As a special case of random forgetting, *group forgetting* means that the model is trained to unlearn the group or sub-class of the corresponding super-classes. A more detailed description is given in Appendix E.2.

## 5.2 EXPERIMENT RESULTS

### 5.2.1 CLASS-WISE FORGETTING

We select the class randomly and run class-wise forgetting on five datasets. We report the results of CIFAR-100 / ImageNet and CIFAR-10 in Table 1 and 4, respectively. The results of 20 News-Group and SVHN are given in Appendix E.3. As we can see, UGradSL and UGradSL+ can boost the performance of GA and FT, respectively without an increment in RTE or drop in TA and RA, leading to comprehensive satisfaction in the main metrics, despite randomness in $D_f$, showing the robustness and flexibility of our methods in MU regardless of the size of the dataset and the data modality. Moreover, in terms of *Avg. Gap*, the proposed method shows its similarity to the retrained model.

### 5.2.2 RANDOM FORGETTING

We select data randomly from every class as $D_f$, making sure all the classes are selected and the size of $D_f$ is 10% of the $D_{tr}$. We report the results of CIFAR-100 and TinyImageNet in Table 2.

Table 1: Results of class-wise forgetting in CIFAR-100 and ImageNet. The best comprehensive metrics are **bold**.

| Method | CIFAR-100 UA | MIA$_{Score}$ | RA | TA | Avg. Gap (↓) | RTE (↓, min) | ImageNet UA | MIA$_{Score}$ | RA | TA | Avg. Gap (↓) | RTE (↓, hr) |
|---|---|---|---|---|---|---|---|---|---|---|---|---|
| Retrain | $100.00_{\pm0.00}$ | $100.00_{\pm0.00}$ | $99.96_{\pm0.01}$ | $71.10_{\pm0.12}$ | - | 26.95 | $100.00_{\pm0.00}$ | $100.00_{\pm0.00}$ | $71.62_{\pm0.12}$ | $69.57_{\pm0.07}$ | - | 26.18 |
| FT | $0.67_{\pm0.38}$ | $27.20_{\pm1.34}$ | $99.96_{\pm0.01}$ | $71.46_{\pm0.09}$ | 43.12 | 1.74 | $52.42_{\pm15.81}$ | $55.86_{\pm18.02}$ | $70.66_{\pm2.54}$ | $69.25_{\pm0.78}$ | 23.25 | 2.87 |
| GA | $99.00_{\pm0.57}$ | $99.07_{\pm0.50}$ | $77.83_{\pm2.07}$ | $53.73_{\pm0.96}$ | 10.36 | 0.06 | $81.23_{\pm0.69}$ | $83.52_{\pm2.08}$ | $66.00_{\pm0.03}$ | $64.72_{\pm0.02}$ | 11.43 | 0.01 |
| IU | $2.07_{\pm1.65}$ | $33.20_{\pm8.83}$ | $99.96_{\pm0.01}$ | $71.39_{\pm0.19}$ | 41.26 | 1.24 | $33.54_{\pm19.46}$ | $49.83_{\pm21.57}$ | $66.25_{\pm1.99}$ | $66.28_{\pm1.19}$ | 31.32 | 1.51 |
| BE | $99.07_{\pm0.34}$ | $99.00_{\pm0.49}$ | $70.81_{\pm2.69}$ | $49.85_{\pm1.32}$ | 13.08 | 0.55 | $98.62_{\pm0.58}$ | $0.15_{\pm0.11}$ | $53.13_{\pm0.27}$ | $56.72_{\pm0.31}$ | 33.14 | 0.24 |
| BS | $98.87_{\pm0.57}$ | $98.73_{\pm0.68}$ | $71.16_{\pm2.60}$ | $50.03_{\pm1.36}$ | 13.06 | 0.77 | $98.85_{\pm0.50}$ | $0.13_{\pm0.12}$ | $53.35_{\pm0.16}$ | $56.93_{\pm0.03}$ | 32.98 | 0.37 |
| $\ell_1$-sparse | $98.97_{\pm1.03}$ | $100.00_{\pm0.00}$ | $86.99_{\pm0.76}$ | $79.08_{\pm0.75}$ | 5.50 | 0.15 | $100.00_{\pm0.00}$ | $100.00_{\pm0.00}$ | $39.01_{\pm1.03}$ | $44.62_{\pm0.91}$ | 14.39 | 0.16 |
| RL | $99.80_{\pm0.35}$ | $100.00_{\pm0.00}$ | $99.97_{\pm0.62}$ | $77.31_{\pm0.35}$ | 1.61 | 1.10 | $100.00_{\pm0.00}$ | $100.00_{\pm0.00}$ | $62.06_{\pm4.19}$ | $62.93_{\pm0.45}$ | 4.05 | 1.17 |
| EU-$k$ | $100.00_{\pm0.00}$ | $0.00_{\pm0.00}$ | $63.79_{\pm1.10}$ | $43.90_{\pm0.73}$ | 40.84 | 4.50 | $100.00_{\pm0.00}$ | $0.00_{\pm0.00}$ | $32.99_{\pm0.07}$ | $37.19_{\pm0.15}$ | 42.75 | 0.62 |
| CF-$k$ | $100.00_{\pm0.00}$ | $0.00_{\pm0.00}$ | $94.88_{\pm0.46}$ | $61.32_{\pm1.17}$ | 28.72 | 3.01 | $99.79_{\pm0.36}$ | $0.00_{\pm0.00}$ | $66.84_{\pm0.03}$ | $68.35_{\pm0.28}$ | 26.55 | 1.25 |
| SCRUB | $30.07_{\pm49.48}$ | $66.60_{\pm29.19}$ | $99.98_{\pm0.01}$ | $77.97_{\pm0.56}$ | 27.56 | 1.07 | $56.59_{\pm2.17}$ | $75.59_{\pm1.19}$ | $66.98_{\pm0.11}$ | $68.24_{\pm0.07}$ | 18.45 | 0.21 |
| SalUN | $99.90_{\pm0.01}$ | $99.96_{\pm0.00}$ | $99.98_{\pm0.00}$ | $75.02_{\pm0.10}$ | 1.02 | 2.15 | $100.00_{\pm0.00}$ | $100.00_{\pm0.00}$ | $63.00_{\pm5.03}$ | $62.72_{\pm0.31}$ | 3.87 | 1.95 |
| GLI | $39.78_{\pm6.74}$ | $69.63_{\pm7.44}$ | $95.57_{\pm2.11}$ | $69.63_{\pm0.60}$ | 24.11 | 1.03 | $53.38_{\pm2.96}$ | $73.31_{\pm3.22}$ | $73.01_{\pm0.11}$ | $63.23_{\pm0.06}$ | 20.26 | 3.79 |
| PABI | $100.00_{\pm0.00}$ | $100.00_{\pm0.00}$ | $98.94_{\pm0.16}$ | $73.41_{\pm0.09}$ | 0.83 | 20.09 | - | - | - | - | - | - |
| UGradSL | $66.59_{\pm0.90}$ | $90.96_{\pm5.05}$ | $95.45_{\pm1.42}$ | $70.34_{\pm1.78}$ | 11.93 | 0.07 | $100.00_{\pm0.00}$ | $100.00_{\pm0.00}$ | $76.91_{\pm1.82}$ | $65.94_{\pm1.35}$ | **2.23** | **0.01** |
| UGradSL+ | $100.00_{\pm0.00}$ | $100.00_{\pm0.00}$ | $98.44_{\pm0.62}$ | $72.12_{\pm0.70}$ | **0.64** | 3.37 | $100.00_{\pm0.00}$ | $100.00_{\pm0.00}$ | $78.16_{\pm0.07}$ | $66.84_{\pm0.06}$ | 2.32 | 4.19 |

Table 2: Results of random forgetting in CIFAR-100 and Tiny-ImageNet. The best comprehensive metrics are **bold**.

| Method | CIFAR-100 UA | MIA$_{Score}$ | RA | TA | Avg. Gap (↓) | RTE (↓, min) | Tiny-ImageNet UA | MIA$_{Score}$ | RA | TA | Avg. Gap (↓) | RTE (↓, min) |
|---|---|---|---|---|---|---|---|---|---|---|---|---|
| Retrain | $29.47_{\pm1.59}$ | $53.50_{\pm1.19}$ | $99.98_{\pm0.01}$ | $70.51_{\pm1.17}$ | - | 25.01 | $49.35_{\pm0.38}$ | $58.44_{\pm0.89}$ | $83.80_{\pm0.29}$ | $59.66_{\pm0.44}$ | - | 235.68 |
| FT | $2.55_{\pm0.03}$ | $10.59_{\pm0.27}$ | $99.95_{\pm0.01}$ | $75.95_{\pm0.05}$ | 18.83 | 1.95 | $29.23_{\pm0.29}$ | $37.02_{\pm0.33}$ | $82.51_{\pm0.20}$ | $60.96_{\pm0.23}$ | 11.03 | 18.61 |
| GA | $2.58_{\pm0.06}$ | $5.95_{\pm0.17}$ | $97.45_{\pm0.02}$ | $76.09_{\pm0.01}$ | 20.64 | 0.29 | $19.34_{\pm1.67}$ | $25.19_{\pm0.68}$ | $81.51_{\pm1.56}$ | $59.66_{\pm0.61}$ | 16.39 | 8.65 |
| IU | $15.71_{\pm5.19}$ | $18.69_{\pm4.12}$ | $84.65_{\pm5.29}$ | $62.20_{\pm4.17}$ | 18.05 | 1.20 | $60.61_{\pm0.01}$ | $83.67_{\pm0.15}$ | $16.36_{\pm0.37}$ | $23.44_{\pm0.29}$ | 35.04 | 7.30 |
| BE | $0.01_{\pm0.00}$ | $1.45_{\pm0.02}$ | $99.97_{\pm0.18}$ | $78.26_{\pm0.00}$ | 22.32 | **0.24** | $17.65_{\pm0.31}$ | $24.48_{\pm0.42}$ | $82.85_{\pm0.20}$ | $58.16_{\pm0.08}$ | 17.03 | **3.53** |
| BS | $2.20_{\pm1.21}$ | $10.73_{\pm0.37}$ | $98.22_{\pm1.26}$ | $70.23_{\pm1.67}$ | 18.02 | 0.34 | $25.45_{\pm0.15}$ | $25.45_{\pm0.15}$ | $81.23_{\pm0.74}$ | $56.75_{\pm0.80}$ | 17.09 | 5.63 |
| $\ell_1$-sparse | $8.19_{\pm0.38}$ | $19.11_{\pm0.52}$ | $88.39_{\pm0.31}$ | $80.26_{\pm0.16}$ | 19.25 | 1.00 | $35.73_{\pm0.35}$ | $41.98_{\pm0.73}$ | $78.19_{\pm0.05}$ | $61.44_{\pm0.12}$ | 9.37 | 23.40 |
| RL | $4.06_{\pm0.37}$ | $50.12_{\pm3.48}$ | $99.92_{\pm0.01}$ | $71.30_{\pm0.12}$ | 7.41 | 1.20 | $40.52_{\pm0.15}$ | $59.01_{\pm0.76}$ | $77.58_{\pm0.06}$ | $60.18_{\pm0.19}$ | 4.04 | 27.08 |
| EU-$k$ | $1.73_{\pm0.06}$ | $3.33_{\pm0.07}$ | $98.44_{\pm0.05}$ | $59.92_{\pm0.43}$ | 22.51 | 1.96 | $33.55_{\pm0.35}$ | $22.19_{\pm1.75}$ | $81.41_{\pm0.27}$ | $58.08_{\pm0.21}$ | 14.01 | 20.02 |
| CF-$k$ | $0.07_{\pm0.02}$ | $0.47_{\pm0.16}$ | $99.98_{\pm0.01}$ | $67.86_{\pm0.12}$ | 21.27 | 0.88 | $19.31_{\pm0.38}$ | $23.22_{\pm2.28}$ | $81.59_{\pm0.37}$ | $58.15_{\pm0.19}$ | 17.25 | 13.18 |
| SCRUB | $0.09_{\pm0.59}$ | $4.01_{\pm1.25}$ | $99.97_{\pm0.34}$ | $77.45_{\pm0.26}$ | 21.46 | 1.06 | $20.11_{\pm1.15}$ | $25.35_{\pm7.53}$ | $80.91_{\pm0.77}$ | $60.11_{\pm0.99}$ | 16.42 | 25.79 |
| SalUN | $35.23_{\pm0.32}$ | $89.39_{\pm0.46}$ | $99.53_{\pm0.04}$ | $64.26_{\pm0.58}$ | 12.10 | 3.33 | $40.39_{\pm0.15}$ | $52.32_{\pm10.67}$ | $77.60_{\pm0.11}$ | $60.30_{\pm0.31}$ | 5.48 | 34.42 |
| GLI | $2.88_{\pm1.51}$ | $9.33_{\pm2.36}$ | $97.16_{\pm1.48}$ | $72.04_{\pm0.30}$ | 18.78 | 0.63 | $38.40_{\pm1.74}$ | $67.87_{\pm2.20}$ | $98.37_{\pm0.27}$ | $61.53_{\pm0.30}$ | 9.21 | 22.92 |
| PABI | $28.33_{\pm0.74}$ | $39.31_{\pm0.88}$ | $99.14_{\pm0.01}$ | $72.00_{\pm0.20}$ | 4.42 | 19.10 | $99.90_{\pm0.03}$ | $66.46_{\pm56.83}$ | $0.50_{\pm0.01}$ | $0.00_{\pm0.00}$ | 50.38 | 54.58 |
| UGradSL | $18.36_{\pm0.17}$ | $40.71_{\pm0.13}$ | $98.38_{\pm0.03}$ | $68.23_{\pm0.16}$ | 6.95 | 0.55 | $40.73_{\pm0.71}$ | $37.58_{\pm0.21}$ | $67.30_{\pm0.04}$ | $50.38_{\pm0.77}$ | 13.82 | 9.47 |
| UGradSL+ | $21.69_{\pm0.59}$ | $49.47_{\pm1.25}$ | $99.87_{\pm0.34}$ | $73.60_{\pm0.26}$ | **3.75** | 3.52 | $53.06_{\pm1.27}$ | $59.46_{\pm1.01}$ | $81.38_{\pm0.75}$ | $52.52_{\pm0.84}$ | **3.57** | 25.93 |

Compared with class-wise forgetting, it is harder to improve the MU performance and still keep the RA and TA close to the retrained model. Benefit from the mix-gradient design, the proposed method can make a good balance between forgetting $D_f$ and retaining the knowledge in $D_r$. The rest of the experiments are given in Appendix E.4.

### 5.2.3 GROUP FORGETTING

Although group forgetting can be seen as part of random forgetting, we want to highlight its use case here due to its practical impacts on *e.g.*, facial attributes classification. The identities can be regarded as the subgroup in the attributes.

Table 3: Results of Group Forgetting on CIFAR-20 and CelebA. For CIFAR-20, the model is trained to classify 20 super-classes, with $D_f$ representing one of five subclasses within a single super-class. In the CelebA dataset, the model performs binary classification to determine whether a person is smiling, with $D_f$ selected based on specific identities. The best comprehensive metrics are **bold**.

| Method | CIFAR-20 UA | MIA$_{Score}$ | RA | TA | Avg. Gap (↓) | RTE (↓, min) | CelebA UA | MIA$_{Score}$ | RA | TA | Avg. Gap (↓) | RTE (↓, min) |
|---|---|---|---|---|---|---|---|---|---|---|---|---|
| Retrain | $13.33_{\pm1.64}$ | $28.47_{\pm0.75}$ | $99.94_{\pm0.01}$ | $81.23_{\pm0.13}$ | - | 27.35 | $6.74_{\pm0.26}$ | $9.77_{\pm1.49}$ | $94.38_{\pm0.49}$ | $91.78_{\pm0.33}$ | - | 258.69 |
| FT | $1.00_{\pm0.43}$ | $2.73_{\pm0.52}$ | $99.37_{\pm0.08}$ | $79.02_{\pm0.03}$ | 10.21 | 7.47 | $5.36_{\pm0.17}$ | $5.87_{\pm0.11}$ | $93.91_{\pm0.04}$ | $93.18_{\pm0.03}$ | 1.79 | 25.94 |
| GA | $87.93_{\pm2.92}$ | $88.93_{\pm2.33}$ | $81.46_{\pm0.77}$ | $64.07_{\pm0.95}$ | 42.68 | **0.11** | $6.00_{\pm0.16}$ | $5.76_{\pm0.14}$ | $92.86_{\pm0.13}$ | $92.52_{\pm0.08}$ | 1.75 | 1.20 |
| IU | $0.00_{\pm0.00}$ | $2.07_{\pm1.29}$ | $99.95_{\pm0.01}$ | $80.92_{\pm0.34}$ | 10.01 | 1.10 | $5.90_{\pm0.11}$ | $4.91_{\pm0.30}$ | $93.05_{\pm0.01}$ | $92.62_{\pm0.01}$ | 1.97 | 219.77 |
| BE | $89.07_{\pm1.39}$ | $91.73_{\pm1.75}$ | $76.36_{\pm0.92}$ | $60.17_{\pm0.92}$ | 45.91 | 0.33 | $11.50_{\pm0.80}$ | $48.41_{\pm8.86}$ | $88.37_{\pm0.81}$ | $88.07_{\pm0.81}$ | 13.28 | 48.91 |
| BS | $88.60_{\pm1.13}$ | $90.67_{\pm1.14}$ | $76.70_{\pm1.08}$ | $60.41_{\pm1.17}$ | 45.38 | 0.29 | $8.95_{\pm5.11}$ | $27.35_{\pm30.20}$ | $91.00_{\pm5.22}$ | $90.63_{\pm5.65}$ | 6.08 | 50.99 |
| $\ell_1$-sparse | $0.13_{\pm0.09}$ | $2.27_{\pm0.57}$ | $99.57_{\pm0.04}$ | $80.44_{\pm0.08}$ | 10.14 | 0.38 | $9.46_{\pm1.82}$ | $36.91_{\pm30.96}$ | $90.52_{\pm1.75}$ | $90.35_{\pm1.77}$ | 8.79 | 37.49 |
| RL | $56.93_{\pm3.24}$ | $98.60_{\pm0.29}$ | $99.92_{\pm0.01}$ | $80.28_{\pm0.05}$ | 28.67 | 0.37 | $8.31_{\pm0.43}$ | $28.55_{\pm16.74}$ | $91.85_{\pm0.51}$ | $91.62_{\pm0.42}$ | 5.76 | 40.09 |
| EU-$k$ | $8.00_{\pm4.57}$ | $16.33_{\pm7.18}$ | $97.07_{\pm0.18}$ | $69.67_{\pm0.35}$ | 7.98 | 0.87 | $7.20_{\pm0.19}$ | $18.77_{\pm3.69}$ | $92.55_{\pm0.30}$ | $91.04_{\pm0.67}$ | 3.01 | 1.98 |
| CF-$k$ | $0.00_{\pm0.00}$ | $0.80_{\pm0.40}$ | $99.98_{\pm0.01}$ | $77.46_{\pm0.03}$ | 11.20 | 1.21 | $5.46_{\pm0.32}$ | $17.26_{\pm0.08}$ | $94.45_{\pm0.04}$ | $92.72_{\pm0.04}$ | 2.45 | 1.60 |
| SCRUB | $0.00_{\pm0.00}$ | $1.13_{\pm0.34}$ | $99.93_{\pm0.01}$ | $81.05_{\pm0.20}$ | 10.21 | 0.30 | $8.78_{\pm0.77}$ | $13.37_{\pm5.22}$ | $91.21_{\pm0.86}$ | $90.65_{\pm0.86}$ | 2.49 | 70.13 |
| SalUN | $52.93_{\pm2.21}$ | $99.80_{\pm0.35}$ | $99.55_{\pm0.00}$ | $76.48_{\pm0.26}$ | 29.02 | 2.88 | $6.53_{\pm0.28}$ | $25.57_{\pm8.22}$ | $92.97_{\pm0.03}$ | $92.27_{\pm0.07}$ | 4.48 | 83.43 |
| GLI | $22.22_{\pm2.47}$ | $34.26_{\pm4.88}$ | $91.12_{\pm0.66}$ | $75.13_{\pm0.73}$ | 7.40 | 1.01 | $6.43_{\pm0.43}$ | $5.70_{\pm0.74}$ | $92.59_{\pm0.08}$ | $92.36_{\pm0.11}$ | 1.69 | **0.52** |
| PABI | $82.96_{\pm9.83}$ | $90.83_{\pm3.98}$ | $99.29_{\pm0.47}$ | $81.47_{\pm0.02}$ | 33.22 | 19.96 | $3.71_{\pm0.19}$ | $45.16_{\pm1.97}$ | $99.83_{\pm0.01}$ | $99.57_{\pm0.00}$ | 12.92 | 38.87 |
| UGradSL | $22.87_{\pm0.90}$ | $38.93_{\pm1.57}$ | $97.20_{\pm0.19}$ | $75.84_{\pm0.16}$ | **7.03** | 0.13 | $6.29_{\pm1.41}$ | $5.73_{\pm3.50}$ | $93.44_{\pm0.14}$ | $92.80_{\pm0.27}$ | 1.61 | 2.17 |
| UGradSL+ | $78.44_{\pm1.19}$ | $88.67_{\pm0.35}$ | $97.93_{\pm0.71}$ | $79.77_{\pm0.58}$ | 32.20 | 8.12 | $6.88_{\pm0.63}$ | $8.88_{\pm0.52}$ | $93.39_{\pm0.41}$ | $92.99_{\pm0.12}$ | **0.81** | 16.78 |

**CIFAR-10 and CIFAR-100** share the same image dataset while CIFAR-100 is labeled with 100 fine-grained classes and 20 coarse (super) classes (Krizhevsky et al., 2009; Chundawat et al., 2023). We train a model to classify 20 super classes using CIFAR-100 training set. The setting of the *group forgetting within one coarse class* is to remove one fine-grained class from one super class in CIFAR-100 datasets. For example, there are five fine-grained fishes in the *Fish* coarse class and we want to remove one fine-grained fish from the model. Different from class-wise forgetting, we do not modify the testing set. We report the group forgetting in Table 3.

**CelebA** We select CelebA dataset as another real-world case and show the results in Table 3. We train a binary classification model to classify whether the person is smile or not. There are 8192 identities in the training set and we select 1% of the identities (82 identities) as $D_f$. Both smiling and non-smiling images are in $D_f$. This experiment has significant practical meaning, since the biometric, such as identity and fingerprint, needs more privacy protection (Minaee et al., 2023). Compared with baseline methods, our method can forget the identity information better without forgetting too much remaining information in the dataset. This paradigm provides a practical usage of MU and our methods provide a faster and more reliable way to improve the MU performance.

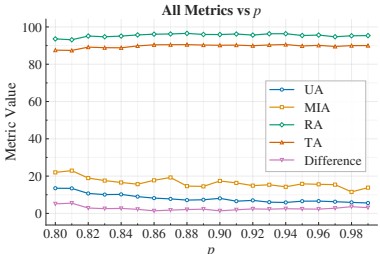 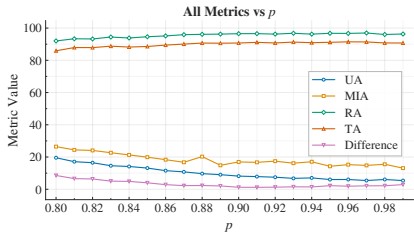

Figure 3: Ablation study on the GA ratio $p$ for random forgetting (10%) on CIFAR-10. Our methods (left: UGradSL, right: UGradSL+) remain relatively stable across a wide range of $p$ under multiple evaluation metrics.

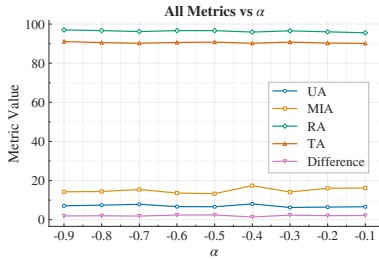 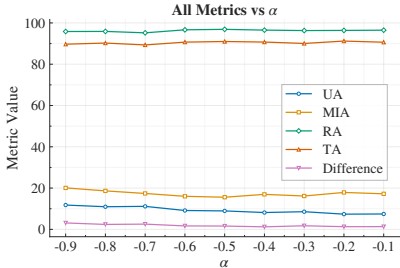

Figure 4: Ablation study on the smoothing rate $\alpha$ for random forgetting (10%) on CIFAR-10. Our methods (left: UGradSL, right: UGradSL+) remain relatively stable across a wide range of $\alpha$ under multiple evaluation metrics.

## 5.3 Ablation Studies

To evaluate the robustness of our method, we conduct ablation studies on the forgetting set size, the GA ratio $p$, and the smoothing rate $\alpha$. For the forgetting set size, we do the experiments on CIFAR-10 and CIFAR-100. The results in Tables 17 and 18 show that our method is consistently robust to the size of $D_f$ and always outperforms the other baselines. For the trade-off parameter $p$, Table 19 indicates that pure GA is unstable, and its performance becomes more stable as more GA steps are combined with our method. In practice, we therefore choose $p > 0.9$ to stabilize performance while still leveraging the effect of GA for unlearning. We further study random forgetting (10%) on CIFAR-10, fixing $\alpha = -0.4$ for both UGradSL and UGradSL+, and sweep $p$ in $[0.8, 1.0]$ with a step size of 0.01. The resulting UA, MIA, RA, TA, and Avg. Gap curves in Figure 3 show that our methods are relatively stable with respect to $p$. For the smoothing rate $\alpha$, we fix $p = 0.9$ and vary $\alpha$ from $-0.9$ to $-0.1$ with a step size of 0.1. The results in Figure 4 demonstrate that both

UGradSL and UGradSL+ remain stable. Overall, these ablations show that our methods are robust to the choice of hyperparameters.

Table 4: Results of class-wise forgetting and random forgetting on CIFAR-10 with additional (add.) MIA. The best comprehensive metrics are **bold**. Avg. Gap is calculated with additional MIA.

| | Class-wise | | | | | | | Random | | | | | | |
|---|---|---|---|---|---|---|---|---|---|---|---|---|---|---|
| | UA | MIA$_{Score}$ | RA | TA | Add. MIA | Avg. Gap (↓) | RTE (↓, min) | UA | MIA$_{Score}$ | RA | TA | Add. MIA | Avg. Gap (↓) | RTE (↓, min) |
| Retrain | $100.00_{\pm0.00}$ | $100.00_{\pm0.00}$ | $98.19_{\pm3.14}$ | $94.50_{\pm0.34}$ | $99.23_{\pm0.08}$ | - | 24.62 | $8.07_{\pm0.47}$ | $17.41_{\pm0.69}$ | $100.00_{\pm0.01}$ | $91.61_{\pm0.24}$ | $50.69_{\pm0.73}$ | - | 24.66 |
| FT | $22.71_{\pm5.31}$ | $79.21_{\pm8.60}$ | $99.82_{\pm0.09}$ | $94.13_{\pm0.14}$ | $99.09_{\pm0.07}$ | 20.04 | 2.02 | $1.10_{\pm0.19}$ | $4.06_{\pm0.41}$ | $99.83_{\pm0.03}$ | $93.70_{\pm0.10}$ | $54.05_{\pm0.31}$ | 5.19 | 1.58 |
| GA | $25.19_{\pm11.38}$ | $73.48_{\pm9.68}$ | $96.84_{\pm0.58}$ | $73.10_{\pm1.62}$ | $99.43_{\pm0.09}$ | 24.86 | **0.08** | $0.56_{\pm0.01}$ | $1.19_{\pm0.05}$ | $99.48_{\pm0.02}$ | $94.55_{\pm0.05}$ | $55.04_{\pm0.66}$ | 6.31 | **0.31** |
| IU | $83.92_{\pm1.16}$ | $92.59_{\pm1.41}$ | $98.77_{\pm0.12}$ | $92.64_{\pm0.23}$ | $99.71_{\pm0.07}$ | 5.28 | 1.18 | $17.51_{\pm2.19}$ | $21.39_{\pm1.70}$ | $83.28_{\pm2.44}$ | $78.15_{\pm2.85}$ | $53.98_{\pm0.55}$ | 9.37 | 1.18 |
| BE | $64.93_{\pm0.01}$ | $98.19_{\pm0.00}$ | $99.47_{\pm0.00}$ | $94.00_{\pm0.11}$ | $99.60_{\pm0.02}$ | 7.81 | 0.20 | $0.00_{\pm0.00}$ | $0.26_{\pm0.02}$ | $100.00_{\pm0.00}$ | $95.35_{\pm0.18}$ | $55.41_{\pm0.49}$ | 6.74 | 3.17 |
| BS | $93.69_{\pm4.32}$ | $99.82_{\pm0.04}$ | $97.69_{\pm1.29}$ | $92.89_{\pm1.26}$ | $99.56_{\pm0.10}$ | 1.79 | 0.29 | $0.48_{\pm0.07}$ | $1.16_{\pm0.04}$ | $99.47_{\pm0.01}$ | $94.58_{\pm0.03}$ | $55.88_{\pm0.72}$ | 6.51 | 1.41 |
| $\ell_1$-sparse | $100.00_{\pm0.00}$ | $100.00_{\pm0.00}$ | $97.86_{\pm1.29}$ | $96.11_{\pm1.26}$ | $99.02_{\pm0.15}$ | 0.43 | 1.00 | $2.80_{\pm0.37}$ | $18.59_{\pm3.48}$ | $99.97_{\pm0.01}$ | $94.08_{\pm0.12}$ | $55.01_{\pm0.49}$ | 2.65 | 1.98 |
| RL | $99.99_{\pm0.01}$ | $100.00_{\pm0.00}$ | $95.50_{\pm0.11}$ | | $99.08_{\pm0.07}$ | 0.59 | 1.04 | $2.27_{\pm0.29}$ | $14.37_{\pm1.01}$ | $99.98_{\pm0.01}$ | $94.14_{\pm0.18}$ | $52.17_{\pm0.87}$ | 2.57 | 1.98 |
| EU-$k$ | $100.00_{\pm0.00}$ | $100.00_{\pm0.00}$ | $100.00_{\pm0.00}$ | $75.04_{\pm1.10}$ | $99.89_{\pm0.18}$ | 4.39 | 1.45 | $0.00_{\pm0.00}$ | $0.50_{\pm0.30}$ | $99.99_{\pm0.01}$ | $77.21_{\pm1.21}$ | $61.88_{\pm1.33}$ | 10.12 | 1.58 |
| CF-$k$ | $100.00_{\pm0.00}$ | $100.00_{\pm0.00}$ | $100.00_{\pm0.00}$ | $78.95_{\pm0.53}$ | $100.00_{\pm0.00}$ | 3.63 | 1.32 | $0.00_{\pm0.00}$ | $0.00_{\pm0.00}$ | $100.00_{\pm0.00}$ | $80.98_{\pm0.27}$ | $69.91_{\pm1.33}$ | 11.07 | 1.47 |
| SCRUB | $100.00_{\pm0.00}$ | $100.00_{\pm0.00}$ | $99.93_{\pm0.01}$ | $95.22_{\pm0.07}$ | $100.00_{\pm0.00}$ | 0.65 | 1.09 | $0.70_{\pm0.59}$ | $3.88_{\pm1.25}$ | $99.59_{\pm0.34}$ | $94.22_{\pm0.26}$ | $55.33_{\pm0.59}$ | 5.71 | 4.05 |
| SalUN | $90.74_{\pm13.91}$ | $100.00_{\pm0.00}$ | $98.20_{\pm0.34}$ | $80.49_{\pm1.21}$ | $98.63_{\pm0.59}$ | 4.78 | 2.22 | $46.95_{\pm0.15}$ | $86.33_{\pm1.29}$ | $97.75_{\pm0.42}$ | $77.22_{\pm0.77}$ | $69.95_{\pm0.12}$ | 28.74 | 2.42 |
| UGradSL | $94.99_{\pm4.35}$ | $97.95_{\pm1.78}$ | $95.47_{\pm4.08}$ | $86.78_{\pm5.68}$ | $99.94_{\pm0.01}$ | 3.64 | 0.22 | $5.87_{\pm0.51}$ | $13.33_{\pm0.70}$ | $98.82_{\pm0.28}$ | $92.17_{\pm0.23}$ | $53.54_{\pm0.97}$ | **2.17** | 0.45 |
| UGradSL+ | $100.00_{\pm0.00}$ | $100.00_{\pm0.00}$ | $99.26_{\pm0.01}$ | $94.29_{\pm0.07}$ | $100.00_{\pm0.00}$ | **0.41** | 3.07 | $6.03_{\pm0.17}$ | $10.65_{\pm0.13}$ | $99.79_{\pm0.03}$ | $93.64_{\pm0.16}$ | $52.29_{\pm0.85}$ | 2.53 | 3.07 |

## 5.4 DISCUSSION

**Influence Function in Deep Learning** Influence functions were originally proposed for the convex function. As given in Section 3.2, we apply the influence function to the converged model, which can be regarded as a local convex model. A plot of loss landscape of the retrained model $\theta_r$ on CIFAR-10 dataset is given in Figure 7 in Appendix.

**MIA as a Proxy for "Forgetfulness".** Given a model $\theta_\star$, we can evaluate the degree of its generalization by running a membership inference attack on the model. In the context of the current work, generalization is equivalent to the degree of "forgetfulness" that the forgetting algorithm achieves. Given the distribution of model response observations $A_f = \mathcal{A}(\theta_\star, \mathcal{D}_f)$ and $A_{te} = \mathcal{A}(\theta_\star, \mathcal{D}_{te})$, where $\mathcal{A}$ is an adversary and $A = A_f \cup A_{te}$ is the observation visible to $\mathcal{A}$, one can get the degree of generalization by analyzing the observations. In the context of MU, the most straightforward way is to get the accuracy of $\mathcal{A}$ on the seen and unseen samples ($\mathcal{D}_{te}$ and $\mathcal{D}_f$ respectively). This could be done by computing the $(TP + TN)/(|D_f| + |D_{te}|)$, where the true positive (TP) predictions correspond to "seen" samples, and true negative (TN) predictions are "unseen" samples. We conducted the experiments on CIFAR-10 both for class-wise and random forgetting. The results are given in Table 4, where Avg. Gap is calculated with additional MIA. We assume that the distribution of $D_{tr}$ and $D_{te}$ should be the same. For class-wise forgetting, the additional MIA is almost 1 because $D_f$ is a separate single class and the distribution of $D_f$ and $D_{te}$ without the corresponding class are totally different. For random forgetting, the additional MIA is almost 0.5 because $D_f$ is randomly selected from $D_{tr}$ and the distribution of $D_f$ and $D_{te}$ should be the same. The plots of loss distribution for random and class-wise forgetting are given in Figure 8 in the Appendix. In Table 4, the proposed methods still outperform the other baseline methods, showing the robustness to the other MIA auditing methods and the generalization capability in privacy preservation.

**Difference between UGradSL and UGradSL+** Although two methods are similar in the mathematical formulation, there exists fundamental difference in their design and behavior. Compared with UGradSL, UGradSL+ can be more stable and less sensitive due to its origin from FT. As shown in the experiment results in the tables, UGradSL+ can always perform as top-tier methods. However, the RTE of UGradSL+ would be higher. We present more analysis in Appendix E.11.

The study of the Streisand effect (Jansen & Martin, 2015) and gradient analysis are given in Appendix E.6 and E.10, respectively.

## 6 CONCLUSIONS

We have proposed UGradSL, a plug-and-play, efficient, gradient-based MU method using smoothed labels. Theoretical proofs and extensive numerical experiments have demonstrated the effectiveness of the proposed method.

ETHIC STATEMENT

This work does not involve potential malicious or unintended uses, fairness considerations, privacy considerations, security considerations, crowd sourcing, or research with human subjects.

REPRODUCIBILITY STATEMENT

We provide details to reproduce our results in Appendix D and E. We also provide pseudo-code in Algorithm 1 and will release the code upon acceptance.

ACKNOWLEDGMENTS

Z. Di and Y. Liu are partially supported by the National Science Foundation (NSF) under grants IIS-2143895 and IIS-2040800. The work of J. Jia and S. Liu is supported in part by the Open Philanthropy Research Award.

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

**Roadmap** The appendix is composed as follows. Section A presents all the notations and their meaning we use in this paper. Section B presents the rest of the Related Work. Section C gives the proof of our theoretical analysis. Section D gives a more detailed explanation of the proposed algorithm. Section E shows the additional experiment results with more details that are not given in the main paper due to the page limit.

## A  NOTATION TABLE

The notations used in the paper is summarized in the Table 5.

Table 5: Notation used in this paper

| Notations | Description |
|---|---|
| $K$ | The number of classes in the dataset |
| $\mathcal{D}, \mathcal{X}, \mathcal{Y}$ | The general dataset distribution, the feature space and the label space |
| $D$ | The dataset $D \in \mathcal{D}$ |
| $D_{tr}, D_r, D_f$ | The training set, remaining set and forgetting set |
| $\Theta_{\mathcal{M}}$ | The distribution of models learned using mechanism $\mathcal{M}$ |
| $\boldsymbol{\theta}$ | The model weight |
| $\boldsymbol{\theta}^*$ | The optimal model weight |
| $\boldsymbol{\theta}^*_{f,\text{LS}}$ | The optimal model weight trained with $D_f$ whose label is smoothed |
| $\|\boldsymbol{\theta}\|$ | The 2-norm of the model weight |
| $n$ | The size of the dataset |
| $\varepsilon$ | The up-weighted weight of data point $z$ in influence function |
| $\mathcal{I}(z)$ | Influence function of data point $z$ |
| $h_{\boldsymbol{\theta}}$ | A function $h$ parameterized by $\boldsymbol{\theta}$ |
| $\ell(h_{\boldsymbol{\theta}}, z_i)$ | Loss of $h_{\boldsymbol{\theta}}(x_i)$ and $y_i$ |
| $R_{tr}(\boldsymbol{\theta})$ | The empirical risk of training set when the model weight is $\boldsymbol{\theta}$ |
| $R_f(\boldsymbol{\theta})$ | The empirical risk of forgetting set when the model weight is $\boldsymbol{\theta}$ |
| $R_r(\boldsymbol{\theta})$ | The empirical risk of remaining set when the model weight is $\boldsymbol{\theta}$ |
| $H_{\boldsymbol{\theta}}$ | The Hessian matrix w.r.t. $\boldsymbol{\theta}$ |
| $\nabla_{\boldsymbol{\theta}}$ | The gradient w.r.t. $\boldsymbol{\theta}$ |
| $B$ | Data batch |
| $B^{\text{LS},\alpha}$ | The smoothed batch using $\alpha$ |
| $z_i = (x_i, y_i)$ | A data point $z_i$ whose feature is $x_i$ and label is $y_i$ |
| $\boldsymbol{y}_i$ | The one-hot encoded vector form of $y_i$ |
| $\boldsymbol{y}_i^{\text{GLS},\alpha}$ | The smoothed one-hot encoded vector form of $y_i$ where the smooth rate is $\alpha$ |
| $\alpha$ | Smooth rate in general label smoothing |
| $h_{\boldsymbol{\theta}}(x)$ | The extracted feature of $x$ from the model parameterized by $\boldsymbol{\theta}$ |
| $\gamma_1, \gamma_2$ | The weight of machine learning and machine unlearning on ERM |

## B  RELATED WORK

**Label Smoothing** (LS) or positive label smoothing (PLS) (Szegedy et al., 2016) is a commonly used regularization method to improve the model performance. Standard training with one-hot labels will lead to overfitting easily. Empirical studies have shown the effectiveness of LS in noisy label (Szegedy et al., 2016; Pereyra et al., 2017; Vaswani et al., 2017; Chorowski & Jaitly, 2016). In addition, LS shows its capability to reduce overfitting, improve generalization, etc. LS can also improve the model calibration (Müller et al., 2019). However, most work on LS is PLS. (Wei et al., 2021) first proposes the concept of negative label smoothing and shows there is a wider feasible domain for the smoothing rate when the rate is negative, expanding the usage of LS.

**Influence Function** is a classic statistical method to track the impact of one training sample. (Koh & Liang, 2017) uses a second-order optimization approximation to evaluate the impact of a training sample. Additionally, it can also be used to identify the importance of the training groups (Basu et al., 2020; Koh et al., 2019). The influence function is widely used in many machine-learning

tasks, such as data bias solution (Brunet et al., 2019; Kong et al., 2021), fairness (Sattigeri et al., 2022; Wang et al., 2022a), security (Liu et al., 2022a), transfer learning (Jain et al., 2022), out-of-distribution generalization (Ye et al., 2021), etc. The approach also plays an important role as the algorithm backbone in the MU tasks (Jia et al., 2023; Warnecke et al., 2021; Izzo et al., 2021).

**Differential Privacy** (DP) is a mathematical framework designed to quantify and mitigate privacy risks in machine learning models. It ensures that the inclusion or exclusion of a single data point in a dataset does not significantly affect the model's output, thus protecting individual data points from being inferred by adversaries (Dwork et al., 2006). In machine learning, DP mechanisms such as noise addition and gradient clipping are employed during the training process to provide formal privacy guarantees while maintaining model utility (Abadi et al., 2016). These techniques help balance the trade-off between data privacy and model performance, making DP a cornerstone of privacy-preserving machine learning (Shokri & Shmatikov, 2015; McMahan et al., 2017).

A multitude of **privacy risk assessment** tools have been proposed to gauge the degree of leakage associated with the training data. Specifically targeted at the training data, model attacks are often used as a proxy metric for privacy leakage in pretrained models. For example, model inversion attacks are designed to extract aggregate information about specific sub-classes rather than individual samples (Fredrikson et al., 2015). Data extraction attacks aim to reverse engineer individual samples used during training (Carlini et al., 2021), while property inference attacks focus on inferring properties of the training data (Ganju et al., 2018).

More relevant to the current work are **Membership Inference Attacks** (MIA), which predict whether a particular sample was used to train the model. First introduced by (Homer et al., 2008), membership attack algorithms were later formalized in the context of DP, enabling privacy attacks and defenses for machine learning models (Rahman et al., 2018). (Shokri et al., 2017) introduced MIA based on the assumption of adversarial queries to the target model. By training a reference attack model (shadow model) based on the model inference response, this type of MIA has proven to be powerful in scenarios such as white-box (Leino & Fredrikson, 2020; Nasr et al., 2019; Sablay-rolles et al., 2019), black-box (Chen et al., 2021a; Hisamoto et al., 2020; Song & Mittal, 2021), and label-only (Choquette-Choo et al., 2021; Li & Zhang, 2021) access. However, most MIA mechanisms often require training a large number of shadow models with diverse subsets of queries, making them prohibitively expensive. As a result, some recent works have focused on developing cheaper MIA mechanisms (Steinke et al., 2023).

**Basics of Influence Function** Given a dataset $D = \{z_i : (x_i, y_i)\}_{i=1}^{n}$ and a function $h$ parameterized by $\boldsymbol{\theta}$ which maps from the input feature space $\mathcal{X}$ to the output space $\mathcal{Y}$. Recall the standard empirical risk minimization is written as:

$$\boldsymbol{\theta}^* = \arg\min_{\boldsymbol{\theta}} \frac{1}{n} \sum_{z \in D} \ell\left(h_{\boldsymbol{\theta}}, z\right). \tag{9}$$

To find the impact of a training point $\hat{z}$, we up-weight its weight by an infinitesimal amount $\varepsilon$[1]. The new model parameter $\boldsymbol{\theta}_{\{\hat{z}\}}^{\varepsilon}$ can be obtained from

$$\boldsymbol{\theta}_{\{z\}}^{\epsilon_I} = \arg\min_{\boldsymbol{\theta}} \frac{1}{n} \sum_{z \in D} \ell\left(h_{\boldsymbol{\theta}}, z\right) + \varepsilon \cdot \ell\left(h_{\boldsymbol{\theta}}, \hat{z}\right) \tag{10}$$

When $\varepsilon = -\frac{1}{n}$, it is indicating removing $\hat{z}$. According to (Koh & Liang, 2017), $\boldsymbol{\theta}_{\{\hat{z}\}}^{\varepsilon}$ can be approximated by using the first-order Taylor series expansion as

$$\boldsymbol{\theta}_{\{\hat{z}\}}^{\varepsilon} \approx \boldsymbol{\theta}^* - \varepsilon \cdot H_{\boldsymbol{\theta}^*}^{-1} \cdot \nabla_{\boldsymbol{\theta}} \ell\left(h_{\boldsymbol{\theta}^*}, \hat{z}\right), \tag{11}$$

where $H_{\boldsymbol{\theta}^*}$ is the Hessian with respect to (w.r.t.) $\boldsymbol{\theta}^*$. The change of $\boldsymbol{\theta}$ due to changing the weight can be given using the influence function $\mathcal{I}(\hat{z})$ as

$$\Delta\boldsymbol{\theta} = \boldsymbol{\theta}_{\{\hat{z}\}}^{\varepsilon} - \boldsymbol{\theta}^* = \mathcal{I}(\hat{z}) = \frac{d\boldsymbol{\theta}_{\{\hat{z}\}}^{\varepsilon}}{d\varepsilon}\bigg|_{\varepsilon=0} = -H_{\boldsymbol{\theta}^*}^{-1} \cdot \nabla_{\boldsymbol{\theta}} \ell\left(h_{\boldsymbol{\theta}^*}, \hat{z}\right).$$

---

[1]To distinguish it from the $\epsilon$ in differential privacy, we use $\varepsilon$ here.

# C PROOFS

## C.1 PROOF FOR THEOREM 1

*Proof.* For $p(x)$, the Taylor expansion at $x = a$ is

$$p(x) = p(a) + \frac{p'(a)}{1}(x - a) + o \tag{12}$$

Here, $p(\boldsymbol{\theta}) = \nabla R_{tr}(\boldsymbol{\theta}) + \varepsilon \sum_{D_f} \nabla \ell(h_{\boldsymbol{\theta}}, z_i^f)$ so we have

$$p(\boldsymbol{\theta}) = \nabla R_{tr}(a) + \varepsilon \sum_{z^f \in D_f} \nabla \ell(h_a, z^f) + \left[ \nabla^2 R_{tr}(a) + \varepsilon \sum_{z^f \in D_f} \nabla^2 \ell(h_a, z^f) \right](\boldsymbol{\theta} - a) + o \tag{13}$$

For Eq. (2), we expand $p(\boldsymbol{\theta}_f^*)$ at $\boldsymbol{\theta} = \boldsymbol{\theta}_{tr}^*$ as

$$
\begin{aligned}
p(\boldsymbol{\theta}_f^*) &= \nabla R_{tr}(\boldsymbol{\theta}_{tr}^*) + \varepsilon \sum_{z^f \in D_f} \nabla \ell(h_{\boldsymbol{\theta}_{tr}^*}, z^f) \\
&+ \left[ \nabla^2 R_{tr}(\boldsymbol{\theta}_{tr}^*) + \varepsilon \sum_{z^f \in D_f} \nabla^2 \ell(h_{\boldsymbol{\theta}_{tr}^*}, z^f) \right](\boldsymbol{\theta}_f^* - \boldsymbol{\theta}_{tr}^*) + o = 0
\end{aligned} \tag{14}
$$

Since we have $\nabla R_{tr}(\boldsymbol{\theta}_{tr}^*) = 0$ and ignore $o$, we can get the approximation as

$$\boldsymbol{\theta}_f^* - \boldsymbol{\theta}_{tr}^* \approx - \left[ \sum_{z^{tr} \in D_{tr}} \nabla^2 \ell(h_{\boldsymbol{\theta}_{tr}^*}, z^{tr}) + \varepsilon \sum_{z^f \in D_f} \nabla^2 \ell(h_{\boldsymbol{\theta}_{tr}^*}, z^f) \right]^{-1} \left[ \varepsilon \sum_{z^f \in D_f} \nabla \ell(h_{\boldsymbol{\theta}_{tr}^*}, z^f) \right] \tag{15}$$

Similarly, we can expand $q(\boldsymbol{\theta}_{tr}^*) = \nabla R_{tr}(\boldsymbol{\theta}_{tr}^*)$ at $\boldsymbol{\theta} = \boldsymbol{\theta}_r^*$ as

$$
\begin{aligned}
q(\boldsymbol{\theta}_{tr}^*) &= \sum_{z^{tr} \in D_{tr}} \nabla \ell(h_{\boldsymbol{\theta}_r^*}, z^{tr}) + \sum_{z^{tr} \in D_{tr}} \nabla^2 \ell(h_{\boldsymbol{\theta}_r^*}, z^{tr})(\boldsymbol{\theta}_{tr}^* - \boldsymbol{\theta}_r^*) \approx 0 \\
\boldsymbol{\theta}_r^* - \boldsymbol{\theta}_{tr}^* &\approx \left[ \sum_{z^{tr} \in D_{tr}} \nabla^2 \ell(h_{\boldsymbol{\theta}_r^*}, z^{tr}) \right]^{-1} \sum_{z^{tr} \in D_{tr}} \nabla \ell(h_{\boldsymbol{\theta}_r^*}, z^{tr})
\end{aligned} \tag{16}
$$

Because of gradient ascent, $\varepsilon = -1$ and we have

$$
\begin{aligned}
\boldsymbol{\theta}_r^* - \boldsymbol{\theta}_f^* = \boldsymbol{\theta}_r^* - \boldsymbol{\theta}_{tr}^* - (\boldsymbol{\theta}_{tr}^* - \boldsymbol{\theta}_f^*) &= \underbrace{\left( \sum_{z^{tr} \in D_{tr}} \nabla^2 \ell(h_{\boldsymbol{\theta}_r^*}, z^{tr}) \right)^{-1} \sum_{z^{tr} \in D_{tr}} \nabla \ell(h_{\boldsymbol{\theta}_r^*}, z^{tr})}_{\Delta \boldsymbol{\theta}_r} \\
&- \underbrace{\left( \sum_{z^r \in D_r} \nabla^2 \ell(h_{\boldsymbol{\theta}_{tr}^*}, z^r) \right)^{-1} \sum_{z^f \in D_f} \nabla \ell(h_{\boldsymbol{\theta}_{tr}^*}, z^f)}_{\Delta \boldsymbol{\theta}_f}
\end{aligned} \tag{17}
$$

Thus, $\|\boldsymbol{\theta}_r^* - \boldsymbol{\theta}_f^*\| = 0$ if and only if $\Delta \boldsymbol{\theta}_f = \Delta \boldsymbol{\theta}_r$, where

$$\sum_{z^{tr} \in D_{tr}} \nabla \ell(h_{\boldsymbol{\theta}_r^*}, z^{tr}) = \underbrace{\left[ \sum_{z^{tr} \in D_{tr}} \nabla^2 \ell(h_{\boldsymbol{\theta}_r^*}, z^{tr}) \right]\left[ \sum_{z^r \in D_r} \nabla^2 \ell(h_{\boldsymbol{\theta}_{tr}^*}, z^r) \right]^{-1}}_{H(\boldsymbol{\theta}_r^*, \boldsymbol{\theta}_{tr}^*)} \sum_{z^f \in D_f} \nabla \ell(h_{\boldsymbol{\theta}_{tr}^*}, z^f) \tag{18}$$

$\square$

## C.2 Error Analysis in Theorem 1

If we do not ignore the Lagrange remainder in Eq. 14 and 16 and denote them as $e_r$ and $e_f$, Eq. 14 and 16 become

$$p(\boldsymbol{\theta}_f^*) = \nabla R_{tr}(\boldsymbol{\theta}_{tr}^*) + \varepsilon \sum_{z^f \in D_f} \nabla \ell(h_{\boldsymbol{\theta}_{tr}^*}, z^f)$$
$$+ \left[ \nabla^2 R_{tr}(\boldsymbol{\theta}_{tr}^*) + \varepsilon \sum_{z^f \in D_f} \nabla^2 \ell(h_{\boldsymbol{\theta}_{tr}^*}, z^f) \right] (\boldsymbol{\theta}_f^* - \boldsymbol{\theta}_{tr}^*) + e_r = 0 \tag{19}$$

$$q(\boldsymbol{\theta}_{tr}^*) = \sum_{z^{tr} \in D_{tr}} \nabla \ell(h_{\boldsymbol{\theta}_r^*}, z^{tr}) + \sum_{z^{tr} \in D_{tr}} \nabla^2 \ell(h_{\boldsymbol{\theta}_r^*}, z^{tr})(\boldsymbol{\theta}_{tr}^* - \boldsymbol{\theta}_r^*) + e_f = 0 \tag{20}$$

, respectively. Thus,

$$\boldsymbol{\theta}_r^* - \boldsymbol{\theta}_f^* = (\boldsymbol{\theta}_r^* - \boldsymbol{\theta}_{tr}^*) - (\boldsymbol{\theta}_f^* - \boldsymbol{\theta}_{tr}^*) \tag{21}$$
$$= (\Delta\boldsymbol{\theta}_r + e_r) - (\Delta\boldsymbol{\theta}_f + e_f) = (\Delta\boldsymbol{\theta}_r - \Delta\boldsymbol{\theta}_f) + (e_r - e_f). \tag{22}$$

We now bound the error of using the linearized difference $\Delta\boldsymbol{\theta}_r - \Delta\boldsymbol{\theta}_f$ to approximate $\boldsymbol{\theta}_r^* - \boldsymbol{\theta}_f^*$.

$$\boldsymbol{\theta}_r^* - \boldsymbol{\theta}_f^* - (\Delta\boldsymbol{\theta}_r - \Delta\boldsymbol{\theta}_f) = e_r - e_f, \tag{23}$$

and hence

$$\left\| \boldsymbol{\theta}_r^* - \boldsymbol{\theta}_f^* - (\Delta\boldsymbol{\theta}_r - \Delta\boldsymbol{\theta}_f) \right\| = \|e_r - e_f\| \le \|e_r\| + \|e_f\|. \tag{24}$$

Assume that $q(\boldsymbol{\theta}) = \nabla R_{tr}(\boldsymbol{\theta})$ and $p(\boldsymbol{\theta}) = \nabla R_{tr}(\boldsymbol{\theta}) - \nabla R_f(\boldsymbol{\theta})$ have Lipschitz-continuous Hessians with constants $L_q$ and $L_p$, respectively, i.e.,

$$\|\nabla^2 q(\boldsymbol{\theta}_1) - \nabla^2 q(\boldsymbol{\theta}_2)\| \le L_q \|\boldsymbol{\theta}_1 - \boldsymbol{\theta}_2\|, \tag{25}$$
$$\|\nabla^2 p(\boldsymbol{\theta}_1) - \nabla^2 p(\boldsymbol{\theta}_2)\| \le L_p \|\boldsymbol{\theta}_1 - \boldsymbol{\theta}_2\|. \tag{26}$$

Then standard Taylor bounds imply

$$\|r_q\| \le \frac{L_q}{2} \|\boldsymbol{\theta}_{tr}^* - \boldsymbol{\theta}_r^*\|^2, \tag{27}$$

$$\|r_p\| \le \frac{L_p}{2} \|\boldsymbol{\theta}_f^* - \boldsymbol{\theta}_{tr}^*\|^2. \tag{28}$$

Using $e_r = -H_r^{-1} r_q$ and $e_f = -H_f^{-1} r_p$, we obtain

$$\|e_r\| \le \|H_r^{-1}\| \|r_q\| \le \frac{L_q}{2} \|H_r^{-1}\| \|\boldsymbol{\theta}_r^* - \boldsymbol{\theta}_{tr}^*\|^2, \tag{29}$$

$$\|e_f\| \le \|H_f^{-1}\| \|r_p\| \le \frac{L_p}{2} \|H_f^{-1}\| \|\boldsymbol{\theta}_{tr}^* - \boldsymbol{\theta}_{tr}^*\|^2. \tag{30}$$

Therefore, the approximation error satisfies

$$\left\| \boldsymbol{\theta}_r^* - \boldsymbol{\theta}_f^* - (\Delta\boldsymbol{\theta}_r - \Delta\boldsymbol{\theta}_f) \right\| \le \frac{L_q}{2} \|H_r^{-1}\| \|\boldsymbol{\theta}_r^* - \boldsymbol{\theta}_{tr}^*\|^2 + \frac{L_p}{2} \|H_f^{-1}\| \|\boldsymbol{\theta}_f^* - \boldsymbol{\theta}_{tr}^*\|^2. \tag{31}$$

## C.3 Proof for Theorem 2

*Proof.* Recall the loss calculation in label smoothing and we have

$$\ell(h_{\boldsymbol{\theta}}, z^{\mathrm{GLS},\alpha}) = (1 + \frac{1-K}{K}\alpha)\ell(h_{\boldsymbol{\theta}}, (x,y)) + \frac{\alpha}{K} \sum_{y' \in \mathcal{Y}\backslash y} \ell(h_{\boldsymbol{\theta}}, (x,y'))), \tag{32}$$

where we use notations $\ell(h_{\boldsymbol{\theta}}, (x,y)) := \ell(h_{\boldsymbol{\theta}}, z)$ to specify the loss of an example $z = \{x,y\}$ existing in the dataset and $\ell(h_{\boldsymbol{\theta}}, (x,y'))$ to denote the loss of an example when its label is replaced

with $y'$. $\nabla_{\boldsymbol{\theta}}\ell(h_{\boldsymbol{\theta}},(x,y))$ is the gradient of the target label and $\sum_{y'\in\mathcal{Y}\backslash y}\nabla_{\boldsymbol{\theta}}\ell(h_{\boldsymbol{\theta}},(x,y'))$ is the sum of the gradient of non-target labels.

With label smoothing in Eq. (32), Eq. (17) becomes

$$
\begin{aligned}
\boldsymbol{\theta}_r^* - \boldsymbol{\theta}_{f,\mathrm{LS}}^* &\approx \Delta\boldsymbol{\theta}_r + (1 + \frac{1-K}{K}\alpha) \cdot (-\Delta\boldsymbol{\theta}_f) + \frac{1-K}{K}\alpha \cdot \Delta\boldsymbol{\theta}_n \\
&= \Delta\boldsymbol{\theta}_r - \Delta\boldsymbol{\theta}_f + \frac{1-K}{K}\alpha \cdot (\Delta\boldsymbol{\theta}_n - \Delta\boldsymbol{\theta}_f)
\end{aligned}
\tag{33}
$$

where

$$
\begin{aligned}
\Delta\boldsymbol{\theta}_r &:= \left[ \sum_{z^{tr}\in D_{tr}} \nabla_{\boldsymbol{\theta}}^2 \ell(h_{\boldsymbol{\theta}_r^*}, z^{tr}) \right]^{-1} \sum_{z^{tr}\in D_{tr}} \nabla_{\boldsymbol{\theta}}\ell(h_{\boldsymbol{\theta}_r^*}, z^{tr}) \\
\Delta\boldsymbol{\theta}_f &:= \left[ \sum_{z^r\in D_r} \nabla_{\boldsymbol{\theta}}^2 \ell(h_{\boldsymbol{\theta}_{tr}^*}, z^r) \right]^{-1} \sum_{z^f\in D_f} \nabla_{\boldsymbol{\theta}}\ell(h_{\boldsymbol{\theta}_{tr}^*}, z^f)
\end{aligned}
$$

as given in Eq. ( 17). So we have

$$
\boldsymbol{\theta}_r^* - \boldsymbol{\theta}_{f,\mathrm{LS}}^* \approx \Delta\boldsymbol{\theta}_r - \Delta\boldsymbol{\theta}_f + \frac{1-K}{K}\alpha \cdot (\Delta\boldsymbol{\theta}_n - \Delta\boldsymbol{\theta}_f)
\tag{34}
$$

where

$$
\Delta\boldsymbol{\theta}_n := \frac{1}{K-1} \left[ \sum_{z^r\in D_r} \nabla_{\boldsymbol{\theta}}^2 \ell(h_{\boldsymbol{\theta}_{tr}^*}, z^r) \right]^{-1} \sum_{z^f\in D_f} \nabla_{\boldsymbol{\theta}} \sum_{y'\in\mathcal{Y}\backslash y^f} \ell(h_{\boldsymbol{\theta}_{tr}^*}, (x^f, y'))
$$

When we have

$$
\langle \Delta\boldsymbol{\theta}_r - \Delta\boldsymbol{\theta}_f, \Delta\boldsymbol{\theta}_n - \Delta\boldsymbol{\theta}_f \rangle \le 0,
\tag{35}
$$

$\alpha < 0$ can help with MU, making

$$
\|\boldsymbol{\theta}_r^* - \boldsymbol{\theta}_{f,\mathrm{NLS}}^*\| \le \|\boldsymbol{\theta}_r^* - \boldsymbol{\theta}_f^*\|
\tag{36}
$$

$\square$

### C.4  PROOF FOR THEOREM 3

*Proof.* When the optimization is gradient ascent (GA) with negative label smoothing (NLS), Eq. (6) can be written as

$$
\ell(h_{\boldsymbol{\theta}}, z^{\mathrm{NLS},\alpha}) = -\left(1 + \frac{1-K}{K}\alpha\right) \cdot \ell(h_{\boldsymbol{\theta}},(x,y)) - \frac{\alpha}{K} \sum_{y'\in\mathcal{Y}\backslash y} \ell(h_{\boldsymbol{\theta}},(x,y')), \alpha < 0,
\tag{37}
$$

Recall $R_{tr}(\boldsymbol{\theta}) = \sum_{z^{tr}\in D_{tr}} \ell(h_{\boldsymbol{\theta}}, z^{tr})$. Denote by $R_f^{\mathrm{NLS}}(\boldsymbol{\theta};\alpha) = \sum_{z^{\mathrm{LS},\alpha}\in D_f} \ell(h_{\boldsymbol{\theta}}, z^{\mathrm{NLS},\alpha}), \alpha < 0$ the empirical risk of forgetting data with NLS. After MU with label smoothing on $D_f$ by gradient ascent, the resulting model can be seen as minimizing the risk $\gamma_1 \cdot R_{tr}(\boldsymbol{\theta}) - \gamma_2 \cdot R_f^{\mathrm{NLS}}(\boldsymbol{\theta};\alpha)$, which is a weighted combination of the risk from two phases: 1) machine learning on $D_{tr}$ with weight $\gamma_1 > 0$ and 2) machine unlearning on $D_f$ with weight $\gamma_2 > 0$. Consider an example $(x,y)$ in the forgetting dataset. The loss of this example is:

$$
\begin{aligned}
\gamma_1 \ell(h_{\boldsymbol{\theta}},(x,y)) - \gamma_2 \ell(h_{\boldsymbol{\theta}}, z^{\mathrm{GLS},\alpha}) &= \left[ \gamma_1 - \gamma_2 \left(1 + \frac{1-K}{K}\alpha\right) \right] \cdot \ell(h_{\boldsymbol{\theta}},(x,y)) \\
&\quad - \frac{\alpha}{K}\gamma_2 \sum_{y'\in\mathcal{Y}\backslash y} \ell(h_{\boldsymbol{\theta}},(x,y')).
\end{aligned}
$$

When $\left[\gamma_1 - \gamma_2\left(1 + \frac{1-K}{K}\alpha\right)\right] > 0$, the optimal solution by minimizing this loss is

$$\mathbb{P}(\mathcal{M}(y) = y^{\text{pred}}) = \begin{cases} \frac{\gamma_1 - \gamma_2\left(1+\frac{1-K}{K}\alpha\right)}{\left(\gamma_1 - \gamma_2\left(1+\frac{1-K}{K}\alpha\right)\right) - \frac{K-1}{K}\alpha\gamma_2}, & \text{if } y^{\text{pred}} = y, \\ \frac{-\frac{\alpha}{K}\cdot\gamma_2}{\left(\gamma_1 - \gamma_2\left(1+\frac{1-K}{K}\alpha\right)\right) - \frac{K-1}{K}\alpha\gamma_2}, & \text{if } y^{\text{pred}} \neq y. \end{cases}$$

Accordingly, for another label $y'$, we have

$$\mathbb{P}(\mathcal{M}(y') = y^{\text{pred}}) = \begin{cases} \frac{\gamma_1 - \gamma_2\left(1+\frac{1-K}{K}\alpha\right)}{\left(\gamma_1 - \gamma_2\left(1+\frac{1-K}{K}\alpha\right)\right) - \frac{K-1}{K}\alpha\gamma_2}, & \text{if } y^{\text{pred}} = y', \\ \frac{-\frac{\alpha}{K}\cdot\gamma_2}{\left(\gamma_1 - \gamma_2\left(1+\frac{1-K}{K}\alpha\right)\right) - \frac{K-1}{K}\alpha\gamma_2}, & \text{if } y^{\text{pred}} \neq y'. \end{cases}$$

Then the quotient of two probabilities can be upper bounded by:

$$\log\left(\frac{\mathbb{P}(\mathcal{M}(y) = y^{\text{pred}})}{\mathbb{P}(\mathcal{M}(y') = y^{\text{pred}})}\right) \leq \left|\log\left(\frac{\gamma_1 - \gamma_2\left(1 + \frac{1-K}{K}\alpha\right)}{-\frac{\alpha}{K}\cdot\gamma_2}\right)\right| = \left|\log\left(\frac{K}{\alpha}\left(1 - \frac{\gamma_1}{\gamma_2}\right) + 1 - K\right)\right| = \epsilon.$$

$\square$

# D  THE DETAILS OF ALGORITHM

## D.1  ALGORITHM DETAILS

We provide a more detailed explanation of UGradSL and UGradSL+ in Algorithm 1 here. For UGradSL+, we first sample a batch $B_r = \{z_i^r : (x_i^r, y_i^r)\}_{i=1}^{n_{B_r}}$ from $D_r$ (Line 3-4). Additionally, we sample a batch $B_f = \{z_i^f : (x_i^f, y_i^f)\}_{i=1}^{n_{B_f}}$ from $D_f$ where $n_{B_r} = n_{B_f}$ (Line 5). We compute the distance $d(z_i^r, z_i^f) \in [0, 1]$ for each $(z_i^r, z_i^f)$ pair where $z_i^r \in B_r$ and $z_i^f \in B_f$ (Line 6). For each $z_i^f$, we count the number of $z_i^r$ whose $d(z_i^r, z_i^f) < \beta$, where $\beta$ is the distance threshold. This count is denoted by $c_i^f$ (Line 7). Then we get the smooth rate by normalizing the count as $\alpha_i = c_i^f / |B_f|$, where $\alpha_i \in [0, 1]$ (Line 8). GA with NLS is to decrease the model confidence of $D_f$. The larger the absolute value of $\alpha_i$, the lower confidence will be given. Our intuition is that a smaller $d(z_i^r, z_i^f)$ means $z_i^r$ is more similar to $D_r$ and the confidence of $z_i^f$ should not be decreased too much. The distances we use is cosine distance. UGradSL is similar and the difference is the dataset replacement. For each epoch, UGradSL+ is terminated after completing the iterations on $D_r$, while UGradSL is terminated after completing the iterations on $D_f$.

## D.2  ALGORITHM EXPLANATION

In the self-adaptive version of UGradSL+, the label smoothing rate for each forgetting sample is computed dynamically from its proximity to the retained data in feature space. For each iteration, the algorithm samples a batch of retained examples $B_r$ and a batch of forgetting examples $B_f$ with equal size, extracts their features $\{z_i^r\}$ and $\{z_j^f\}$, and computes the **feature distance** $d(z_i^r, z_j^f)$ for every retained-forgetting pair. Then, for each forgetting feature $z_j^f$, it counts how many retained features fall within a distance threshold $\beta$, denoted as $c_j^f$. This count is normalized by the batch size $|B_f|$ to obtain the adaptive smoothing rate $\alpha_j = c_j^f / |B_f|$. As a result, forgetting samples that are close to many retained samples (i.e., highly entangled in representation space) receive a higher smoothing rate and are updated more conservatively, while those that are far from retained data get a lower smoothing rate (possibly zero) and can be pushed away more aggressively during unlearning.

## D.3  ADDITIONAL RESULTS

As mentioned in Section 4, to avoid the smooth rate selection, we propose a self-adaptive smooth rate version. We compare the performance with and without self-adaptive smooth on CIFAR-10 and SVHN. The forgetting scenario is random forgetting. The results are given in Table 10.

### D.4 COMPLEXITY ANALYSIS

Compared with the fixed $\alpha$, the additional computation from the adaptive version is the distance calculation. The code we compute the distance is given below. All computations are implemented as **batched GPU tensor operations without any explicit Python loops**. We assume the feature from $D_r$ and $D_f$ are both in $\mathbb{R}^{n \times d}$, where $n$ is the batch size and $d$ is the feature dimension.

For the FLOP count,

- The two normalization operations cost approximately $6nd$ FLOPs in total, since normalizing a single $n \times d$ tensor requires about $3nd$ FLOPs (square, sum, and division).

- Computing the cosine similarity matrix costs about $2n^2 d$ FLOPs, as each of the $n^2$ entries is a dot product between two $d$-dimensional vectors.

- Converting similarity to distance and applying the threshold require about $2n^2$ and $n^2$ FLOPs, respectively.

- The density computation costs about $n^2$ FLOPs for forming the mask and $n$ FLOPs for the length normalization.

Overall, the total FLOP count is $6nd + 2n^2 d + 4n^2 + n$, which is dominated by the $O(n^2 d)$ cosine-similarity term. For our typical setting $n = 64$ and $d = 512$, this corresponds to roughly $4.4 \times 10^6$ FLOPs. Compared with the FP32 peak throughput of an A6000 GPU (38.71 TFLOPS), this overhead is negligible relative to the usual forward/backward passes.

For memory usage, the additional GPU tensors have the following shapes:

- Each features: $n \times d$

- Each normalized features: $n \times d$

- The cosine similarity, cosine distance and the filtered mask: $n \times n$

- The density: $n$

Assuming FP32 (4 bytes) for all tensors, the peak extra memory is at most $4(4nd + 3n^2 + n)$ bytes, which is $561{,}408$ bytes ($\approx 0.5$ MiB) for $n = 64$ and $d = 512$. This is negligible compared with the model parameters, so the memory overhead can also be safely ignored.

```python
forget_norm = F.normalize(forget_feature, p=2, dim=1)
retain_norm = F.normalize(retain_feature, p=2, dim=1)
# cosine similarity (batch x batch)
cos_sim = forget_norm @ retain_norm.T
# convert to distance in [0,1]
cos_dist = (1 - cos_sim) / 2  # shape: [batch, batch]
# --- threshold and count ---
threshold = 0.2  # example threshold in [0,1]
# boolean matrix: True = close
close_mask = cos_dist < threshold  # [batch, batch]
density = close_mask.sum(dim=0) / len(forget_feature)
```

### D.5 ABLATION STUDY

By default, we adopt cosine distance because it naturally lies in $[0, 1]$, and we set the threshold $\beta$ to the median of all pairwise distances. We conduct an ablation study on different distance metrics and thresholds for random forgetting on CIFAR-10, where the forgetting set size is 10% of the training data. For comparison, we also evaluate Euclidean distance. The results for cosine and Euclidean

Table 6: The ablation studies of threshold $\beta$ and different distance functions of UGradSL for the random forgetting on CIFAR-10 and the size of forgetting set is 10% of the training set. The first row is the results for retraining for reference.

| $\beta$ | Distance | UA | MIA$_{Score}$ | RA | TA | Avg. Gap ($\downarrow$) |
|---|---|---|---|---|---|---|
| - | - | 8.07 | 17.41 | 100.00 | 91.61 | - |
| Median | Cosine | $6.04_{\pm0.11}$ | $13.75_{\pm0.32}$ | $99.11_{\pm0.01}$ | $92.07_{\pm0.02}$ | 1.76 |
|  | Euclidean | $8.59_{\pm1.85}$ | $17.30_{\pm0.98}$ | $94.39_{\pm1.15}$ | $88.97_{\pm1.12}$ | 2.22 |
| 0.1 | Cosine | $6.50_{\pm0.14}$ | $14.76_{\pm1.52}$ | $95.64_{\pm0.23}$ | $89.91_{\pm0.17}$ | 2.57 |
|  | Euclidean | $6.68_{\pm0.88}$ | $14.69_{\pm1.66}$ | $95.34_{\pm0.79}$ | $89.90_{\pm0.69}$ | 2.62 |
| 0.2 | Cosine | $7.01_{\pm0.67}$ | $15.86_{\pm0.86}$ | $95.18_{\pm0.44}$ | $89.69_{\pm0.19}$ | 2.34 |
|  | Euclidean | $6.82_{\pm0.44}$ | $15.81_{\pm0.70}$ | $95.58_{\pm0.73}$ | $90.02_{\pm0.57}$ | 2.21 |
| 0.3 | Cosine | $7.01_{\pm0.98}$ | $15.13_{\pm1.26}$ | $95.24_{\pm0.99}$ | $89.76_{\pm0.41}$ | 2.49 |
|  | Euclidean | $7.32_{\pm1.06}$ | $16.45_{\pm2.08}$ | $94.68_{\pm0.89}$ | $89.16_{\pm0.33}$ | 2.37 |
| 0.4 | Cosine | $7.91_{\pm0.26}$ | $15.69_{\pm1.11}$ | $94.69_{\pm0.51}$ | $89.07_{\pm0.29}$ | 2.43 |
|  | Euclidean | $6.24_{\pm0.21}$ | $14.16_{\pm0.12}$ | $95.75_{\pm0.40}$ | $90.13_{\pm0.13}$ | 2.70 |
| 0.5 | Cosine | $7.61_{\pm0.66}$ | $16.50_{\pm1.68}$ | $95.03_{\pm0.36}$ | $89.69_{\pm0.72}$ | 2.07 |
|  | Euclidean | $8.27_{\pm1.33}$ | $16.44_{\pm1.83}$ | $94.67_{\pm1.33}$ | $89.03_{\pm1.28}$ | 2.27 |
| 0.6 | Cosine | $8.76_{\pm0.28}$ | $16.53_{\pm1.88}$ | $94.31_{\pm0.61}$ | $88.54_{\pm0.50}$ | 2.58 |
|  | Euclidean | $8.67_{\pm0.28}$ | $17.01_{\pm2.43}$ | $94.34_{\pm0.16}$ | $88.93_{\pm0.30}$ | 2.34 |
| 0.7 | Cosine | $9.88_{\pm1.05}$ | $18.33_{\pm2.82}$ | $93.55_{\pm0.92}$ | $88.08_{\pm0.42}$ | 3.18 |
|  | Euclidean | $9.61_{\pm0.86}$ | $17.93_{\pm2.33}$ | $94.11_{\pm0.49}$ | $88.69_{\pm0.19}$ | 2.72 |
| 0.8 | Cosine | $9.61_{\pm1.12}$ | $16.91_{\pm1.51}$ | $93.68_{\pm1.20}$ | $88.48_{\pm0.76}$ | 2.87 |
|  | Euclidean | $9.75_{\pm0.17}$ | $16.79_{\pm0.52}$ | $93.87_{\pm0.02}$ | $88.34_{\pm0.39}$ | 2.93 |
| 0.9 | Cosine | $9.19_{\pm0.66}$ | $17.84_{\pm0.72}$ | $94.19_{\pm0.50}$ | $88.51_{\pm0.84}$ | 2.63 |
|  | Euclidean | $9.76_{\pm0.49}$ | $18.61_{\pm0.65}$ | $93.90_{\pm0.39}$ | $88.47_{\pm0.35}$ | 3.03 |
| 1.0 | Cosine | $9.39_{\pm0.07}$ | $16.94_{\pm0.26}$ | $94.26_{\pm0.33}$ | $88.74_{\pm0.22}$ | 2.60 |
|  | Euclidean | $10.41_{\pm0.24}$ | $19.16_{\pm1.08}$ | $93.50_{\pm0.63}$ | $88.21_{\pm0.34}$ | 3.50 |

Table 7: The ablation studies of threshold $\beta$ and different distance functions of UGradSL+ for the random forgetting on CIFAR-10 and the size of forgetting set is 10% of the training set. The first row is the results for retraining for reference.

| $\beta$ | Distance | UA | MIA$_{Score}$ | RA | TA | Avg. Gap ($\downarrow$) |
|---|---|---|---|---|---|---|
| - | - | 8.07 | 17.41 | 100.00 | 91.61 | - |
| Median | Cosine | $7.54_{\pm0.43}$ | $13.57_{\pm0.12}$ | $99.67_{\pm0.00}$ | $92.97_{\pm0.17}$ | 1.52 |
|  | Euclidean | $11.21_{\pm0.21}$ | $21.02_{\pm2.23}$ | $94.35_{\pm0.22}$ | $88.58_{\pm0.26}$ | 3.86 |
| 0.1 | Cosine | $7.79_{\pm0.52}$ | $17.04_{\pm0.61}$ | $95.84_{\pm0.27}$ | $90.10_{\pm0.47}$ | 1.58 |
|  | Euclidean | $7.30_{\pm0.62}$ | $16.42_{\pm0.66}$ | $96.16_{\pm0.94}$ | $90.46_{\pm0.91}$ | 1.69 |
| 0.2 | Cosine | $8.38_{\pm0.19}$ | $17.46_{\pm1.09}$ | $95.38_{\pm0.34}$ | $89.56_{\pm0.53}$ | 1.76 |
|  | Euclidean | $7.80_{\pm0.76}$ | $16.55_{\pm1.91}$ | $95.75_{\pm1.04}$ | $89.80_{\pm0.50}$ | 1.80 |
| 0.3 | Cosine | $8.27_{\pm0.65}$ | $18.19_{\pm0.29}$ | $95.94_{\pm0.84}$ | $90.18_{\pm0.62}$ | 1.62 |
|  | Euclidean | $7.68_{\pm0.65}$ | $17.28_{\pm0.52}$ | $95.85_{\pm0.75}$ | $90.25_{\pm0.55}$ | 1.51 |
| 0.4 | Cosine | $8.49_{\pm0.28}$ | $17.92_{\pm0.52}$ | $95.85_{\pm0.20}$ | $90.09_{\pm0.03}$ | 1.66 |
|  | Euclidean | $8.38_{\pm0.60}$ | $17.86_{\pm0.89}$ | $95.60_{\pm0.78}$ | $90.06_{\pm0.57}$ | 1.68 |
| 0.5 | Cosine | $9.23_{\pm0.89}$ | $16.81_{\pm1.66}$ | $95.46_{\pm0.62}$ | $89.79_{\pm0.86}$ | 2.03 |
|  | Euclidean | $8.98_{\pm0.69}$ | $16.77_{\pm1.62}$ | $95.39_{\pm1.01}$ | $89.34_{\pm1.17}$ | 2.11 |
| 0.6 | Cosine | $9.95_{\pm0.64}$ | $19.90_{\pm0.95}$ | $95.47_{\pm0.12}$ | $89.82_{\pm0.30}$ | 2.67 |
|  | Euclidean | $10.00_{\pm0.10}$ | $19.00_{\pm1.92}$ | $95.15_{\pm0.26}$ | $89.53_{\pm0.28}$ | 2.61 |
| 0.7 | Cosine | $11.81_{\pm0.74}$ | $20.67_{\pm2.62}$ | $94.25_{\pm0.76}$ | $88.78_{\pm1.02}$ | 3.90 |
|  | Euclidean | $11.25_{\pm0.59}$ | $21.54_{\pm1.12}$ | $94.69_{\pm0.71}$ | $89.05_{\pm0.71}$ | 3.79 |
| 0.8 | Cosine | $13.06_{\pm0.53}$ | $18.81_{\pm0.81}$ | $92.89_{\pm0.69}$ | $87.29_{\pm0.75}$ | 4.45 |
|  | Euclidean | $12.07_{\pm0.45}$ | $19.23_{\pm2.00}$ | $93.81_{\pm0.95}$ | $88.34_{\pm1.00}$ | 3.82 |
| 0.9 | Cosine | $11.75_{\pm0.09}$ | $21.02_{\pm1.43}$ | $94.34_{\pm0.38}$ | $88.81_{\pm0.31}$ | 3.94 |
|  | Euclidean | $12.01_{\pm1.12}$ | $21.49_{\pm1.17}$ | $94.26_{\pm1.08}$ | $88.74_{\pm0.88}$ | 4.16 |
| 1.0 | Cosine | $11.48_{\pm0.06}$ | $20.59_{\pm2.63}$ | $94.19_{\pm0.56}$ | $88.82_{\pm0.38}$ | 3.80 |
|  | Euclidean | $11.79_{\pm0.37}$ | $17.35_{\pm0.85}$ | $94.37_{\pm0.34}$ | $88.67_{\pm0.56}$ | 3.09 |

distance under different $\beta$ values for UGradSL and UGradSL+ are reported in Table 6 and Table 7, respectively.

# E  EXPERIMENTS

## E.1  ADDITIONAL EXPERIMENTAL SETTINGS

The datasets and model configurations for the original model training and retraining are given in Table 8. We run all the experiments using PyTorch 1.12 on NVIDIA A5000 GPUs and AMD EPYC 7513 32-Core Processor.

Table 8: The hyperparameters used in the original training and retraining for different models and datasets.

| Settings | CIFAR-10 | | | SVHN | CIFAR-100 | ImageNet | 20 Newsgroups |
| | ResNet-18 | VGG-16 | ViT | ResNet-18 | ResNet-18 | ResNet-18 | Bert |
|---|---|---|---|---|---|---|---|
| Batch Size | 256 | 256 | 256 | 256 | 256 | 1024 | 128 |
| Learning rate | $1e^{-2}$ | $1e^{-4}$ | $1e^{-6}$ | $1e^{-2}$ | $1e^{-2}$ | $1e^{-2}$ | $1e^{-4}$ |
| Epochs | 160 | 160 | 160 | 160 | 160 | 90 | 60 |

The settings of the baseline methods are:

- Fine-tuning (FT): FT is to fine-tune the original model $\boldsymbol{\theta}_o$ trained from $D_{tr}$ using $D_r$. We fix the epoch of FT for 10 epochs for all the datasets except ImageNet. We fine-tune ImageNet for 5 epochs. The learning rate is the same as the original training.

- Fisher forgetting (FF): FF is to perturb the $\boldsymbol{\theta}_o$ by adding the Gaussian noise, which with a zero mean and a covariance corresponds to the 4th root of the Fisher Information Matrix with respect to (w.r.t.) $\boldsymbol{\theta}_o$ on $D_r$ (Golatkar et al., 2020). We perform a greedy search for hyperparameter tuning between $1e^{-9}$ and $1e^{-6}$.

- Influence unlearning (IU): IU uses influence function (Koh & Liang, 2017) to estimate the change from $\boldsymbol{\theta}_o$ to $\boldsymbol{\theta}_u$ when one training sample is removed.

- Boundary unlearning[2] (BU): BU unlearns the data by assigning pseudo label and manipulating the decision boundary. It contains boundary shrink and boundary expansion, two types of unlearning methods. The hyper-parameters are the default values in the paper.

- $\ell_1$-sparse[3]: $\ell_1$-sparse improves machine unlearning by integrating the $\ell_1$ norm-based sparse penalty to the loss function. The learning rate is $1e^{-3}$ and we search $\gamma$ in $[1e^{-5}, 1e^{-1}]$ as given in (Jia et al., 2023).

- SCRUB: SCRUB casts the unlearning problem into a teacher-student framework. We follow the settings exactly the same in the original repo[4] where $\gamma = 0.99$ and $\alpha = 0.001$.

- Random Labeling (RL): Unlike FT, RL is to train the model with the random label rather than the fixed label. The settings are the same as for FT.

- SalUN[5]: SalUN takes the weight saliency into consideration. We search $\gamma$ from $[0.5, 0.9]$.

## E.2  DATASET SPLIT OF DIFFERENT FORGETTING PARADIGMS

We also provide the details of the dataset split for different forgetting paradigms. For classwise forgetting, we remove the whole class from $D_{tr}$ and $D_{te}$. In CIFAR-10 and CIFAR-100, the size of $D_f$ is 500 and 5000, respectively. For the other datasets, the size of $D_f$ ranges from the smallest class size to the largest class size because we remove the whole class completely. The selected class to be forgotten is totally random. For random forgetting, we randomly select $10\%$ data from $D_{tr}$ as $D_f$. We make sure the distribution of $D_f$ is the same as $D_{tr}$. For CIFAR-20 in group forgetting, each fine-grained class is in the same size which is 500. The coarse class is 2500.

---

[2]https://github.com/TY-LEE-KR/Boundary-Unlearning-Code
[3]https://github.com/OPTML-Group/Unlearn-Sparse
[4]https://github.com/meghdadk/SCRUB/tree/main
[5]https://github.com/OPTML-Group/Unlearn-Saliency

Table 9: The experiment results of class-wise forgetting in 20 Newsgroups and SVHN datasets.

| 20 Newsgroups | UA | MIA$_{Score}$ | RA | TA | Avg. Gap (↓) | RTE (↓, min) |
|---|---|---|---|---|---|---|
| Retrain | $100.00_{\pm0.00}$ | $100.00_{\pm0.00}$ | $98.31_{\pm2.56}$ | $81.95_{\pm1.69}$ | - | 26.25 |
| FT | $4.14_{\pm2.11}$ | $9.23_{\pm3.40}$ | $98.83_{\pm0.86}$ | $82.63_{\pm0.73}$ | 46.96 | 1.77 |
| GA | $17.12_{\pm9.48}$ | $62.03_{\pm5.84}$ | $99.99_{\pm0.01}$ | $85.41_{\pm0.37}$ | 31.50 | **0.37** |
| IU | $0.00_{\pm0.00}$ | $0.25_{\pm0.12}$ | $100.00_{\pm0.00}$ | $85.58_{\pm0.20}$ | 51.27 | 1.52 |
| BS | $78.33_{\pm3.47}$ | $92.63_{\pm2.19}$ | $97.28_{\pm0.99}$ | $90.93_{\pm0.81}$ | 9.76 | 1.42 |
| UGradSL | $100.00_{\pm0.00}$ | $100.00_{\pm0.00}$ | $96.31_{\pm4.02}$ | $78.54_{\pm5.10}$ | 1.35 | 0.39 |
| UGradSL+ | $100.00_{\pm0.00}$ | $100.00_{\pm0.00}$ | $99.76_{\pm0.23}$ | $84.21_{\pm0.41}$ | **0.93** | 2.13 |
| **SVHN** | **UA** | **MIA$_{Score}$** | **RA** | **TA** | **Avg. Gap (↓)** | **RTE (↓, min)** |
| Retrain | $100.00_{\pm0.00}$ | $100.00_{\pm0.00}$ | $100.00_{\pm0.01}$ | $95.94_{\pm0.11}$ | - | 37.05 |
| FT | $6.49_{\pm1.49}$ | $99.98_{\pm0.04}$ | $100.00_{\pm0.01}$ | $96.08_{\pm0.01}$ | 23.42 | 2.42 |
| GA | $87.49_{\pm1.94}$ | $99.85_{\pm0.09}$ | $99.52_{\pm0.03}$ | $95.27_{\pm0.21}$ | 3.45 | **0.15** |
| IU | $93.55_{\pm2.78}$ | $100.00_{\pm0.00}$ | $99.54_{\pm0.03}$ | $95.64_{\pm0.31}$ | 1.80 | 0.23 |
| BE | $85.56_{\pm3.07}$ | $99.98_{\pm0.02}$ | $99.55_{\pm0.01}$ | $95.53_{\pm0.07}$ | 3.83 | 3.17 |
| BS | $96.62_{\pm1.14}$ | $99.95_{\pm0.09}$ | $99.99_{\pm0.00}$ | $95.39_{\pm0.18}$ | 1.00 | 3.91 |
| $\ell_1$-sparse | $99.78_{\pm0.31}$ | $100.00_{\pm0.00}$ | $98.63_{\pm0.01}$ | $97.36_{\pm0.18}$ | 0.75 | 2.91 |
| RL | $99.99_{\pm0.01}$ | $100.00_{\pm0.00}$ | $100.00_{\pm0.00}$ | $95.44_{\pm0.13}$ | 0.13 | 3.53 |
| EU-$k$ | $100.00_{\pm0.00}$ | $100.00_{\pm0.00}$ | $99.61_{\pm0.08}$ | $65.56_{\pm2.38}$ | 7.69 | 4.93 |
| CF-$k$ | $0.09_{\pm0.03}$ | $2.18_{\pm2.21}$ | $99.34_{\pm0.02}$ | $69.87_{\pm4.13}$ | 56.12 | 5.02 |
| SCRUB | $99.99_{\pm0.02}$ | $100.00_{\pm0.00}$ | $100.00_{\pm0.00}$ | $95.79_{\pm0.26}$ | **0.04** | 4.97 |
| SalUN | $99.74_{\pm0.39}$ | $100.00_{\pm0.00}$ | $99.53_{\pm0.02}$ | $95.00_{\pm1.50}$ | 0.42 | 4.77 |
| UGradSL | $90.71_{\pm4.08}$ | $99.90_{\pm0.16}$ | $99.54_{\pm0.04}$ | $95.64_{\pm0.25}$ | 2.54 | 0.23 |
| UGradSL+ | $100.00_{\pm0.00}$ | $100.00_{\pm0.00}$ | $99.82_{\pm0.62}$ | $94.35_{\pm0.70}$ | 0.44 | 4.56 |

### E.3 ADDITIONAL CLASS-WISE FORGETTING RESULTS

We present the performance of class-wise forgetting in 20 Newsgroups and SVHN datasets in Table 9. The observation is similar in CIFAR-100 and ImageNet given in Table 1. UGradSL and UGradSL+ can improve the MU performance with acceptable time increment, showing the generalization of the proposed method in different modalities and different dataset sizes.

### E.4 ADDITIONAL RANDOM FORGETTING RESULTS

We present the performance of random forgetting in CIFAR-10 and SVHN datasets in Table 10. The observation is similar in CIFAR-100 and Tiny ImageNet given in Table 2.

### E.5 MU WITH THE OTHER CLASSIFIER

To validate the generalization of the proposed method, we also try the other classification models. We test vision transformer (ViT) and VGG-16 on the task of class-wise forgetting and random forgetting using CIFAR-10, respectively. The results are given in Tables 11 and 12. The observation is similar in Tables 1 and 2, respectively.

Table 11: The experiment results of class-wise forgetting in CIFAR-10 using ViT.

| CIFAR-10 | UA | MIA$_{Score}$ | RA | TA | Avg. Gap (↓) | RTE (↓, min) |
|---|---|---|---|---|---|---|
| Retrain | $100.00_{\pm0.00}$ | $100.00_{\pm0.00}$ | $61.41_{\pm0.81}$ | $58.94_{\pm1.09}$ | - | 189.08 |
| FT | $3.97_{\pm0.87}$ | $7.60_{\pm1.76}$ | $98.29_{\pm0.05}$ | $80.44_{\pm0.22}$ | 61.70 | 2.99 |
| GA | $33.77_{\pm6.36}$ | $40.47_{\pm6.63}$ | $89.47_{\pm4.21}$ | $71.65_{\pm2.79}$ | 41.63 | 0.32 |
| IU | $1.74_{\pm0.09}$ | $2.16_{\pm0.61}$ | $73.96_{\pm0.01}$ | $68.88_{\pm0.00}$ | 54.65 | 0.24 |
| BE | $85.56_{\pm3.07}$ | $99.98_{\pm0.02}$ | $99.55_{\pm0.01}$ | $95.53_{\pm0.07}$ | 22.30 | 3.17 |
| UGradSL | $68.11_{\pm11.03}$ | $73.84_{\pm9.58}$ | $84.11_{\pm2.70}$ | $68.33_{\pm1.69}$ | 22.54 | **0.22** |
| UGradSL+ | $99.99_{\pm0.01}$ | $99.99_{\pm0.02}$ | $94.46_{\pm1.06}$ | $77.26_{\pm1.19}$ | **12.85** | 5.86 |

Table 10: The experiment results of random forgetting in CIFAR-10 and SVHN.

| CIFAR-10 | UA | MIA$_{Score}$ | RA | TA | Avg. Gap ($\downarrow$) | RTE ($\downarrow$, min) |
|---|---|---|---|---|---|---|
| Retrain | $8.07_{\pm 0.47}$ | $17.41_{\pm 0.69}$ | $100.00_{\pm 0.01}$ | $91.61_{\pm 0.24}$ | - | 24.66 |
| FT | $1.10_{\pm 0.19}$ | $4.06_{\pm 0.41}$ | $99.83_{\pm 0.03}$ | $93.70_{\pm 0.10}$ | 5.65 | 1.58 |
| GA | $0.56_{\pm 0.01}$ | $1.19_{\pm 0.05}$ | $99.48_{\pm 0.02}$ | $94.55_{\pm 0.05}$ | 6.80 | **0.31** |
| IU | $17.51_{\pm 2.19}$ | $21.39_{\pm 1.70}$ | $83.28_{\pm 2.44}$ | $78.13_{\pm 2.85}$ | 10.91 | 1.18 |
| BE | $0.00_{\pm 0.00}$ | $0.26_{\pm 0.02}$ | $100.00_{\pm 0.00}$ | $95.35_{\pm 0.18}$ | 7.24 | 3.17 |
| BS | $0.48_{\pm 0.07}$ | $1.16_{\pm 0.04}$ | $99.47_{\pm 0.01}$ | $94.58_{\pm 0.03}$ | 6.84 | 1.41 |
| $\ell_1$-sparse | $1.21_{\pm 0.38}$ | $4.33_{\pm 0.52}$ | $97.39_{\pm 0.31}$ | $95.49_{\pm 0.18}$ | 6.61 | 1.82 |
| SCRUB | $0.70_{\pm 0.59}$ | $3.88_{\pm 1.25}$ | $99.59_{\pm 0.34}$ | $94.22_{\pm 0.26}$ | 5.98 | 4.05 |
| Random Label | $2.80_{\pm 0.37}$ | $18.59_{\pm 3.48}$ | $99.97_{\pm 0.01}$ | $94.08_{\pm 0.12}$ | 2.24 | 1.98 |
| UGradSL | $5.87_{\pm 0.51}$ | $13.33_{\pm 0.70}$ | $98.82_{\pm 0.28}$ | $92.17_{\pm 0.23}$ | 2.01 | 0.45 |
| UGradSL+ | $6.03_{\pm 0.17}$ | $10.65_{\pm 0.13}$ | $99.79_{\pm 0.03}$ | $93.64_{\pm 0.16}$ | 2.76 | 3.07 |
| UGradSL (Adp) | $6.04_{\pm 0.11}$ | $13.75_{\pm 0.32}$ | $99.11_{\pm 0.01}$ | $92.07_{\pm 0.02}$ | 1.76 | 1.35 |
| UGradSL+ (Adp) | $7.54_{\pm 0.43}$ | $13.57_{\pm 0.12}$ | $99.67_{\pm 0.00}$ | $92.97_{\pm 0.17}$ | **1.52** | 9.23 |
| **SVHN** | **UA** | **MIA$_{Score}$** | **RA** | **TA** | **Avg. Gap ($\downarrow$)** | **RTE ($\downarrow$, min)** |
| Retrain | $4.95_{\pm 0.03}$ | $15.59_{\pm 0.93}$ | $99.99_{\pm 0.01}$ | $95.61_{\pm 0.22}$ | - | 35.65 |
| FT | $0.45_{\pm 0.14}$ | $2.30_{\pm 0.04}$ | $99.99_{\pm 0.00}$ | $95.78_{\pm 0.01}$ | 4.49 | 2.76 |
| GA | $0.58_{\pm 0.04}$ | $1.13_{\pm 0.02}$ | $99.56_{\pm 0.01}$ | $95.62_{\pm 0.01}$ | 4.82 | 0.31 |
| FF | $0.45_{\pm 0.09}$ | $1.30_{\pm 0.12}$ | $99.55_{\pm 0.01}$ | $95.49_{\pm 0.03}$ | 4.84 | 6.02 |
| BE | $0.00_{\pm 0.02}$ | $0.02_{\pm 0.17}$ | $100.00_{\pm 0.01}$ | $96.14_{\pm 0.02}$ | 5.27 | 1.03 |
| BS | $0.45_{\pm 0.14}$ | $1.13_{\pm 0.05}$ | $99.57_{\pm 0.03}$ | $95.66_{\pm 0.01}$ | 4.86 | 4.24 |
| $\ell_1$-sparse | $3.73_{\pm 0.78}$ | $8.44_{\pm 0.34}$ | $97.84_{\pm 0.28}$ | $96.18_{\pm 0.33}$ | 2.77 | **0.07** |
| SCRUB | $0.35_{\pm 0.20}$ | $4.96_{\pm 0.93}$ | $99.94_{\pm 0.02}$ | $95.36_{\pm 0.23}$ | 3.88 | 3.24 |
| RL | $8.00_{\pm 0.64}$ | $29.40_{\pm 11.92}$ | $98.72_{\pm 0.45}$ | $94.04_{\pm 1.10}$ | 4.93 | 1.79 |
| UGradSL | $3.29_{\pm 2.53}$ | $14.32_{\pm 4.56}$ | $99.89_{\pm 0.02}$ | $94.38_{\pm 0.28}$ | 1.07 | 0.57 |
| UGradSL+ | $5.77_{\pm 2.93}$ | $15.95_{\pm 2.26}$ | $100.00_{\pm 0.00}$ | $95.12_{\pm 0.50}$ | 0.42 | 4.44 |
| UGradSL (Adp) | $3.97_{\pm 0.29}$ | $14.63_{\pm 2.15}$ | $99.89_{\pm 0.01}$ | $94.40_{\pm 0.12}$ | 0.81 | 2.20 |
| UGradSL+ (Adp) | $5.07_{\pm 0.34}$ | $15.89_{\pm 1.03}$ | $100.00_{\pm 0.00}$ | $95.21_{\pm 0.44}$ | **0.21** | 14.33 |

Table 12: The experiment results of random forgetting across all classes in CIFAR-10 using VGG-16

| CIFAR-10 | UA | MIA$_{Score}$ | RA | TA | Avg. Gap ($\downarrow$) | RTE ($\downarrow$, min) |
|---|---|---|---|---|---|---|
| Retrain | $11.41_{\pm 0.41}$ | $11.97_{\pm 0.50}$ | $74.65_{\pm 0.23}$ | $66.13_{\pm 0.16}$ | - | 9.48 |
| FT | $1.32_{\pm 0.13}$ | $3.48_{\pm 0.13}$ | $74.24_{\pm 0.04}$ | $67.04_{\pm 0.10}$ | 4.98 | 0.60 |
| GA | $1.35_{\pm 0.08}$ | $2.18_{\pm 0.66}$ | $73.95_{\pm 0.01}$ | $66.88_{\pm 0.01}$ | 5.33 | **0.14** |
| IU | $1.74_{\pm 0.09}$ | $2.16_{\pm 0.61}$ | $73.96_{\pm 0.01}$ | $68.88_{\pm 0.00}$ | 5.73 | 0.24 |
| FF | $1.35_{\pm 0.09}$ | $2.21_{\pm 0.58}$ | $73.95_{\pm 0.02}$ | $66.87_{\pm 0.04}$ | 5.32 | 1.02 |
| BE | $0.01_{\pm 0.01}$ | $0.23_{\pm 0.05}$ | $99.98_{\pm 0.00}$ | $94.04_{\pm 0.21}$ | 19.10 | 1.09 |
| BS | $0.01_{\pm 0.01}$ | $0.22_{\pm 0.03}$ | $99.98_{\pm 0.01}$ | $94.00_{\pm 0.14}$ | 19.09 | 3.17 |
| $\ell_1$-sparse | $1.27_{\pm 1.13}$ | $3.60_{\pm 2.41}$ | $98.97_{\pm 1.13}$ | $92.18_{\pm 1.46}$ | 17.22 | 0.08 |
| SCRUB | $61.16_{\pm 50.89}$ | $44.65_{\pm 43.31}$ | $39.26_{\pm 50.57}$ | $36.95_{\pm 46.68}$ | 36.75 | 0.91 |
| UGradSL | $13.45_{\pm 0.63}$ | $11.77_{\pm 0.54}$ | $65.05_{\pm 0.48}$ | $58.52_{\pm 0.38}$ | **4.86** | 0.19 |
| UGradSL+ | $12.41_{\pm 0.32}$ | $14.96_{\pm 0.52}$ | $65.90_{\pm 0.52}$ | $58.58_{\pm 0.35}$ | 5.13 | 1.08 |

## E.6 STREISAND EFFECT

From the perspective of security, it is important to make the predicted distributions are almost the same from the forgetting set $D_f$ and the testing set $D_{te}$, which is called Streisand effect. We investigate this effect in the *random forgetting* on CIFAR-10 by plotting confusion matrix as shown in Figure 5. It can be found that our method will not lead to the extra hint of $D_f$.

Table 13: Ablation studies of GA ratio $p$ for random forgetting on CIFAR-10. The forgetting set size is 10% training set. The method is UGradSL. We fix $\alpha$ as -0.4. The first row is the retraining results for reference.

| $p$ | UA | $\text{MIA}_{\text{Score}}$ | RA | TA | Avg. Gap ($\downarrow$) |
|------|------|------|------|------|------|
| - | 8.07 | 17.41 | 100.00 | 91.61 | - |
| 0.80 | $12.47_{\pm1.01}$ | $20.24_{\pm2.12}$ | $94.11_{\pm0.71}$ | $88.21_{\pm0.57}$ | 4.13 |
| 0.81 | $11.57_{\pm1.69}$ | $19.68_{\pm3.31}$ | $94.39_{\pm1.41}$ | $88.56_{\pm1.04}$ | 3.61 |
| 0.82 | $10.61_{\pm0.08}$ | $17.85_{\pm0.97}$ | $94.92_{\pm0.18}$ | $88.94_{\pm0.37}$ | 2.68 |
| 0.83 | $9.64_{\pm0.52}$ | $16.49_{\pm0.96}$ | $95.54_{\pm0.71}$ | $89.63_{\pm0.71}$ | 2.23 |
| 0.84 | $9.33_{\pm1.04}$ | $15.84_{\pm1.17}$ | $95.43_{\pm0.95}$ | $89.48_{\pm0.71}$ | 2.38 |
| 0.85 | $8.27_{\pm0.82}$ | $14.74_{\pm1.36}$ | $96.05_{\pm0.41}$ | $90.08_{\pm0.40}$ | 2.09 |
| 0.86 | $7.96_{\pm0.42}$ | $15.45_{\pm2.00}$ | $96.10_{\pm0.44}$ | $90.22_{\pm0.18}$ | **1.84** |
| 0.87 | $7.51_{\pm0.26}$ | $15.26_{\pm3.47}$ | $96.05_{\pm0.18}$ | $90.20_{\pm0.51}$ | 2.02 |
| 0.88 | $6.87_{\pm0.37}$ | $13.18_{\pm1.26}$ | $96.43_{\pm0.35}$ | $90.29_{\pm0.64}$ | 2.58 |
| 0.89 | $6.91_{\pm0.56}$ | $14.44_{\pm2.46}$ | $96.38_{\pm0.73}$ | $90.47_{\pm0.36}$ | 2.22 |
| 0.90 | $6.92_{\pm1.08}$ | $13.60_{\pm3.42}$ | $96.00_{\pm0.50}$ | $90.26_{\pm0.14}$ | 2.58 |
| 0.91 | $6.44_{\pm1.30}$ | $14.16_{\pm2.27}$ | $95.93_{\pm1.18}$ | $90.17_{\pm0.72}$ | 2.60 |
| 0.92 | $6.50_{\pm0.69}$ | $14.35_{\pm0.72}$ | $95.64_{\pm0.50}$ | $90.06_{\pm0.12}$ | 2.64 |
| 0.93 | $5.88_{\pm0.82}$ | $14.84_{\pm1.26}$ | $96.03_{\pm0.84}$ | $90.31_{\pm0.54}$ | 2.51 |
| 0.94 | $5.65_{\pm0.30}$ | $13.55_{\pm0.78}$ | $96.25_{\pm0.44}$ | $90.54_{\pm0.10}$ | 2.77 |
| 0.95 | $6.13_{\pm1.29}$ | $13.14_{\pm2.43}$ | $95.73_{\pm1.03}$ | $89.88_{\pm0.75}$ | 3.05 |
| 0.96 | $6.07_{\pm0.91}$ | $14.28_{\pm2.15}$ | $95.64_{\pm0.79}$ | $90.15_{\pm0.36}$ | 2.74 |
| 0.97 | $5.83_{\pm1.25}$ | $14.07_{\pm1.98}$ | $95.20_{\pm0.98}$ | $89.67_{\pm0.59}$ | 3.08 |
| 0.98 | $5.73_{\pm0.84}$ | $13.19_{\pm1.99}$ | $95.43_{\pm0.98}$ | $89.82_{\pm0.38}$ | 3.23 |
| 0.99 | $5.83_{\pm1.05}$ | $12.98_{\pm1.37}$ | $94.99_{\pm0.79}$ | $89.46_{\pm0.59}$ | 3.46 |

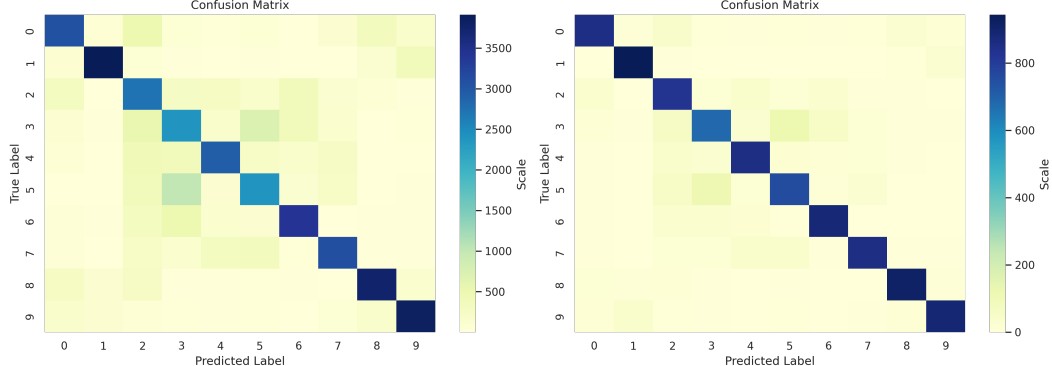

Figure 5: The confusion matrix of testing set and forgetting set $D_f$ using our method on CIFAR-10 with random forgetting across all the classes. There is no big difference between the prediction distribution. Our method will not make $D_f$ more distinguishable.

### E.7 ABLATION STUDY: FORGETTING SET SIZE

Since the size of the forgetting set can affect unlearning performance, we further evaluate the robustness of our method under varying forgetting ratios. In addition to the 10% random forgetting results reported in Table 2 and Table 4, we consider forgetting set sizes of 20%, 30%, 40%, and 50% of the training data on CIFAR-10 and CIFAR-100. The results are summarized in Tables 17 and 18.

### E.8 ABLATION STUDY: GA RATIO $p$

In addition to an overview of the performance fluctuation in Figure 3. We provide the specific value of the ablation study regarding GA ratio $p$. We test the performance on random forgetting on CIFAR-10. The forgetting set size is 10% of the training set. The results of UGradSL and UGradSL+ are given in Tables 13 and 14.

Table 14: Ablation studies of GA ratio $p$ for random forgetting on CIFAR-10. The forgetting set size is 10% training set. The method is UGradSL+. We fix $\alpha$ as -0.4. The first row is the retraining results for reference.

| $p$ | UA | $\text{MIA}_{\text{Score}}$ | RA | TA | Avg. Gap ($\downarrow$) |
|---|---|---|---|---|---|
| - | 8.07 | 17.41 | 100.00 | 91.61 | - |
| 0.80 | $18.84_{\pm0.71}$ | $26.78_{\pm1.78}$ | $91.95_{\pm0.86}$ | $86.05_{\pm1.21}$ | 8.44 |
| 0.81 | $17.00_{\pm0.47}$ | $24.55_{\pm0.76}$ | $93.21_{\pm0.21}$ | $87.50_{\pm0.49}$ | 6.74 |
| 0.82 | $16.45_{\pm0.33}$ | $24.22_{\pm0.46}$ | $93.60_{\pm0.49}$ | $87.64_{\pm0.25}$ | 6.39 |
| 0.83 | $14.45_{\pm0.73}$ | $21.73_{\pm0.82}$ | $94.66_{\pm0.38}$ | $88.59_{\pm0.19}$ | 4.76 |
| 0.84 | $13.44_{\pm0.77}$ | $20.92_{\pm1.15}$ | $94.67_{\pm0.71}$ | $88.67_{\pm0.44}$ | 4.29 |
| 0.85 | $12.57_{\pm0.65}$ | $19.18_{\pm0.64}$ | $95.25_{\pm1.02}$ | $89.33_{\pm1.09}$ | 3.32 |
| 0.86 | $11.42_{\pm0.14}$ | $18.34_{\pm0.61}$ | $95.56_{\pm0.46}$ | $89.49_{\pm0.27}$ | 2.71 |
| 0.87 | $10.90_{\pm0.72}$ | $17.22_{\pm0.51}$ | $95.79_{\pm0.39}$ | $89.77_{\pm0.81}$ | 2.27 |
| 0.88 | $10.13_{\pm0.42}$ | $17.85_{\pm2.11}$ | $95.97_{\pm0.17}$ | $90.03_{\pm0.60}$ | 2.03 |
| 0.89 | $8.98_{\pm0.29}$ | $14.94_{\pm0.09}$ | $96.20_{\pm0.27}$ | $90.23_{\pm0.33}$ | 2.14 |
| 0.90 | $8.41_{\pm0.33}$ | $16.87_{\pm1.17}$ | $96.53_{\pm0.03}$ | $90.64_{\pm0.09}$ | 1.33 |
| 0.91 | $8.01_{\pm0.30}$ | $17.33_{\pm1.17}$ | $96.50_{\pm0.36}$ | $90.68_{\pm0.40}$ | **1.14** |
| 0.92 | $7.74_{\pm0.33}$ | $15.62_{\pm1.80}$ | $96.28_{\pm0.26}$ | $90.48_{\pm0.46}$ | 1.75 |
| 0.93 | $6.67_{\pm0.12}$ | $15.93_{\pm0.22}$ | $96.86_{\pm0.10}$ | $90.96_{\pm0.34}$ | 1.67 |
| 0.94 | $6.79_{\pm0.71}$ | $16.47_{\pm0.52}$ | $96.42_{\pm0.83}$ | $90.74_{\pm0.45}$ | 1.67 |
| 0.95 | $6.03_{\pm0.26}$ | $14.82_{\pm1.39}$ | $96.76_{\pm0.41}$ | $90.94_{\pm0.35}$ | 2.14 |
| 0.96 | $5.78_{\pm0.24}$ | $14.79_{\pm1.14}$ | $96.90_{\pm0.19}$ | $91.30_{\pm0.16}$ | 2.08 |
| 0.97 | $5.98_{\pm0.49}$ | $14.96_{\pm0.34}$ | $96.56_{\pm0.45}$ | $90.81_{\pm0.53}$ | 2.20 |
| 0.98 | $6.46_{\pm0.74}$ | $15.15_{\pm1.76}$ | $95.52_{\pm0.67}$ | $90.15_{\pm0.89}$ | 2.45 |
| 0.99 | $5.67_{\pm0.27}$ | $14.40_{\pm1.18}$ | $96.17_{\pm0.46}$ | $90.61_{\pm0.24}$ | 2.56 |

Table 15: The ablation study of smoothing rate $\alpha$ for random forgetting on CIFAR-10. The forgetting set size is 10% training set. The method we use is UGradSL. We fix $p$ as 0.9.

| $\alpha$ | UA | $\text{MIA}_{\text{Score}}$ | RA | TA | Avg. Gap ($\downarrow$) |
|---|---|---|---|---|---|
| - | 8.07 | 17.41 | 100.00 | 91.61 | - |
| $-0.9$ | $8.17_{\pm1.74}$ | $14.96_{\pm2.47}$ | $95.81_{\pm1.97}$ | $89.97_{\pm1.46}$ | **2.10** |
| $-0.8$ | $6.98_{\pm0.47}$ | $13.41_{\pm0.99}$ | $96.67_{\pm0.88}$ | $90.75_{\pm0.22}$ | 2.32 |
| $-0.7$ | $7.23_{\pm0.56}$ | $14.33_{\pm1.23}$ | $96.28_{\pm0.11}$ | $90.47_{\pm0.26}$ | 2.20 |
| $-0.6$ | $6.69_{\pm0.22}$ | $12.93_{\pm0.67}$ | $96.46_{\pm0.39}$ | $90.64_{\pm0.04}$ | 2.59 |
| $-0.5$ | $6.56_{\pm0.29}$ | $13.00_{\pm0.50}$ | $96.58_{\pm0.23}$ | $90.66_{\pm0.20}$ | 2.57 |
| $-0.4$ | $6.92_{\pm1.08}$ | $13.60_{\pm3.42}$ | $96.00_{\pm0.50}$ | $90.26_{\pm0.14}$ | 2.58 |
| $-0.3$ | $6.32_{\pm0.43}$ | $13.63_{\pm0.67}$ | $96.18_{\pm0.41}$ | $90.52_{\pm0.27}$ | 2.61 |
| $-0.2$ | $6.95_{\pm0.54}$ | $13.98_{\pm1.99}$ | $95.41_{\pm0.68}$ | $89.65_{\pm0.56}$ | 2.77 |
| $-0.1$ | $7.13_{\pm1.44}$ | $14.47_{\pm1.91}$ | $95.08_{\pm1.55}$ | $89.57_{\pm1.04}$ | 2.71 |

## E.9 Smoothing Ratio $\alpha$

Similar to $p$, we report the detailed results regarding the smoothing rate $\alpha$. The results of UGradSL and UGradSL+ are given in Tables 15 and 16.

## E.10 Gradient Analysis

As mentioned in Section 3.3, $\langle \Delta\boldsymbol{\theta}_r - \Delta\boldsymbol{\theta}_f, \Delta\boldsymbol{\theta}_n - \Delta\boldsymbol{\theta}_f \rangle \leq 0$ is always practically valid. We check the results on CelebA dataset (ResNet-18), ImageNet (ViT), CIFAR-100 (VGG-16) and CIFAR-10 (ResNet-18). The distribution of $\langle \Delta\boldsymbol{\theta}_r - \Delta\boldsymbol{\theta}_f, \Delta\boldsymbol{\theta}_n - \Delta\boldsymbol{\theta}_f \rangle$ is shown in Figure 6, which aligns with our assumption.

## E.11 The difference between UGradSL and UGradSL+

Although UGradSL and UGradSL+ look similar, the intuition of these two method is totally different because of the difference between FT and GA. We conducted experiments to illustrate the difference between GA and FT as well as UGradSL and UGradSL+. The results are given in Table 19. The dataset and forgetting paradigm is CIFAR-10 random forgetting. It can be found that the difference becomes much larger when the number of epochs is over 8. When the number of epochs is 10, the model becomes unusable because TA is less than 10%. We also report the performance of UGradSL

Table 16: The ablation study of smoothing rate $\alpha$ for random forgetting on CIFAR-10. The forgetting set size is 10% training set. The method we use is UGradSL+. We fix $p$ as 0.9.

| $\alpha$ | UA | $MIA_{Score}$ | RA | TA | Avg. Gap ($\downarrow$) |
|---|---|---|---|---|---|
| - | 8.07 | 17.41 | 100.00 | 91.61 | - |
| $-0.9$ | $11.59_{\pm0.40}$ | $19.41_{\pm0.59}$ | $95.77_{\pm0.58}$ | $89.47_{\pm0.56}$ | 2.97 |
| $-0.8$ | $10.68_{\pm0.27}$ | $18.41_{\pm0.48}$ | $95.94_{\pm0.24}$ | $89.91_{\pm0.36}$ | 2.35 |
| $-0.7$ | $10.12_{\pm1.01}$ | $16.88_{\pm0.69}$ | $96.07_{\pm0.89}$ | $90.08_{\pm0.78}$ | 2.01 |
| $-0.6$ | $8.98_{\pm0.15}$ | $16.29_{\pm0.87}$ | $96.64_{\pm0.19}$ | $90.75_{\pm0.05}$ | 1.56 |
| $-0.5$ | $9.07_{\pm0.21}$ | $15.83_{\pm0.25}$ | $96.65_{\pm0.34}$ | $90.43_{\pm0.50}$ | 1.78 |
| $-0.4$ | $8.41_{\pm0.33}$ | $16.87_{\pm1.17}$ | $96.53_{\pm0.03}$ | $90.64_{\pm0.09}$ | **1.33** |
| $-0.3$ | $8.59_{\pm0.07}$ | $16.86_{\pm2.15}$ | $96.24_{\pm0.38}$ | $90.20_{\pm0.15}$ | 1.56 |
| $-0.2$ | $7.55_{\pm0.18}$ | $16.68_{\pm1.60}$ | $96.43_{\pm0.14}$ | $90.86_{\pm0.33}$ | 1.39 |
| $-0.1$ | $7.57_{\pm0.18}$ | $17.32_{\pm0.23}$ | $96.15_{\pm0.38}$ | $90.34_{\pm0.27}$ | 1.43 |

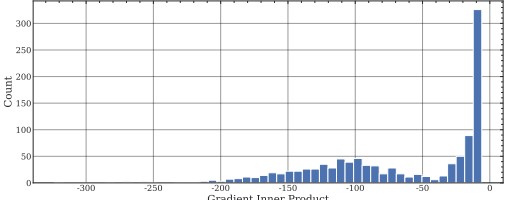

(a) The group forgetting on CelebA using ResNet-18

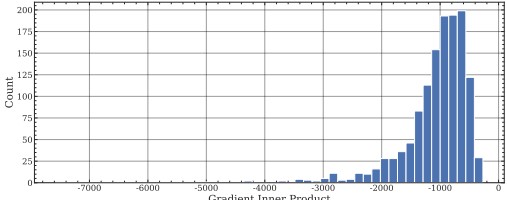

(b) The class-wise forgetting on ImageNet using ViT

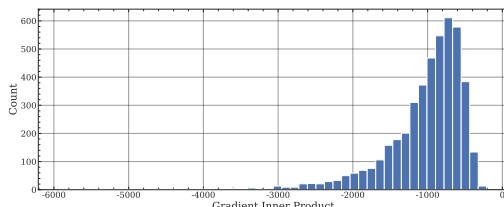

(c) The random forgetting on CIFAR-100 using VGG-16

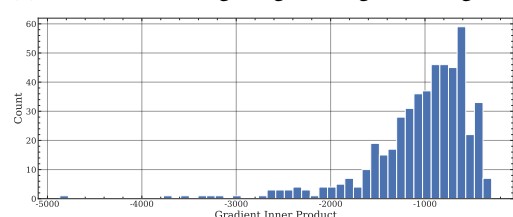

(d) The class-wise forgetting on CIFAR-10 using ResNet-18

Figure 6: The distribution of $\langle \Delta\boldsymbol{\theta}_r - \Delta\boldsymbol{\theta}_f, \Delta\boldsymbol{\theta}_n - \Delta\boldsymbol{\theta}_f \rangle$ using multiple models on multiple datasets.

and UGradSL+ in different epochs. For UGradSL, when the epochs are over 14, the model cannot be used at all. For UGradSL+, the algorithm is much more stable, showing the very good adaptive capability.

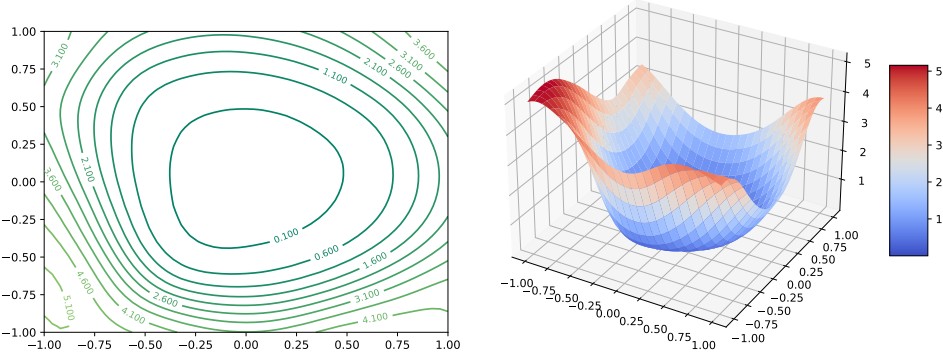

Figure 7: The loss landscape of $\boldsymbol{\theta}_r$ on CIFAR-10 and the model is ResNet-18.

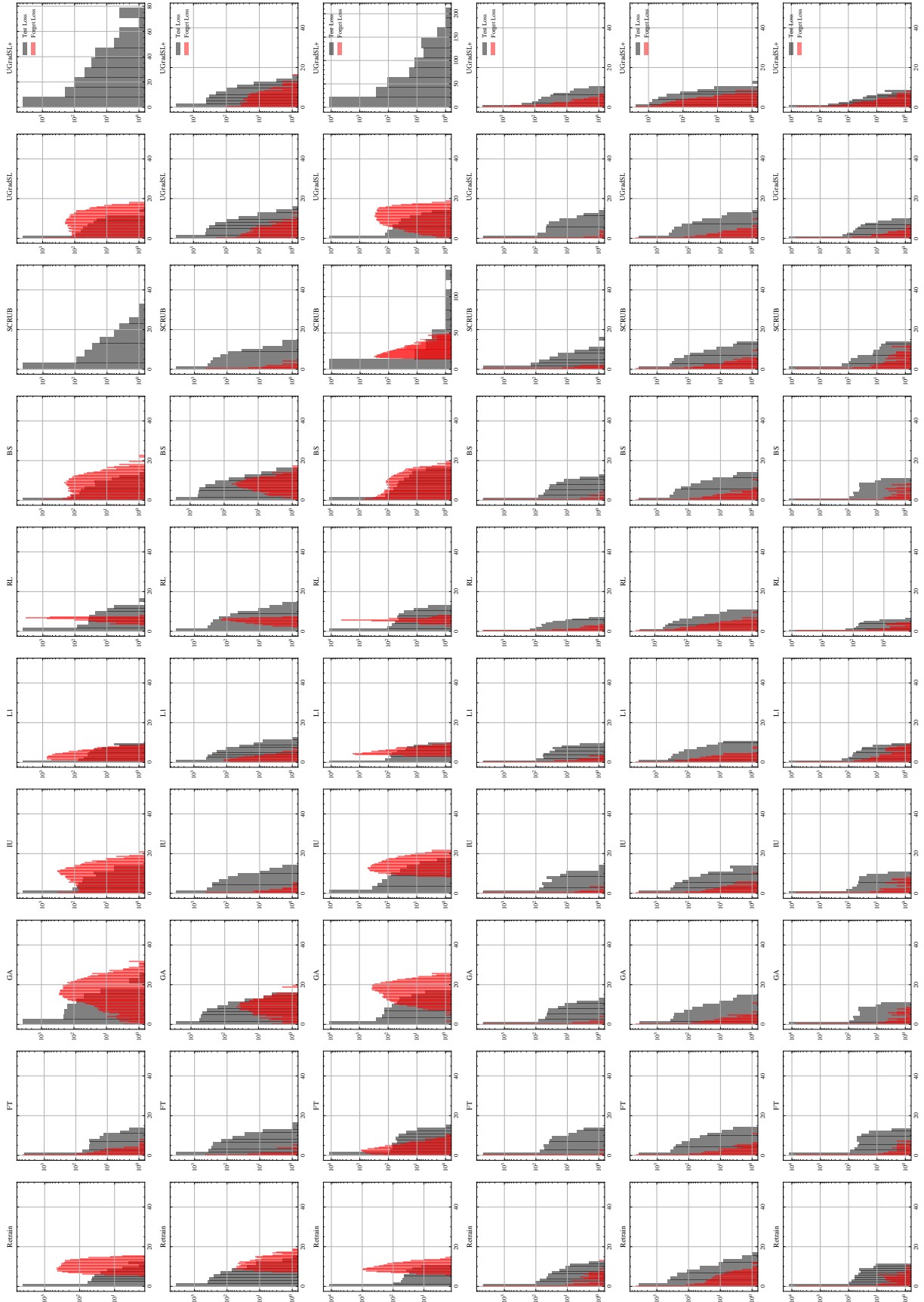

Figure 8: The distributions of the cross-entropy losses for the forget and test instances from the unlearned models. The y-axis is in log scale for better visualization. From the first to the last figure, they are random forgetting on CIFAR-10, CIFAR-100, SVHN and class-wise forgetting on CIFAR-10, CIFAR-100, SVHN.

Table 17: MU Performance across different forgetting data amounts on ResNet-18, pre-trained on CIFAR-10 dataset, for random data forgetting.

| Method | UA | MIA$_{\text{Score}}$ | RA | TA | Avg. Gap (↓) | RTE (↓, min) |
|---|---|---|---|---|---|---|
| **Random Set Size (10%)** | | | | | | |
| Retrain | 8.07 | 17.41 | 100.00 | 91.61 | - | 24.66 |
| FT | $1.10_{\pm0.19}$ | $4.06_{\pm0.40}$ | $98.83_{\pm0.03}$ | $93.70_{\pm0.10}$ | 5.90 | 1.58 |
| RL | $6.39_{\pm1.09}$ | $0.00_{\pm0.00}$ | $99.50_{\pm0.10}$ | $99.04_{\pm0.08}$ | 6.76 | 1.92 |
| GA | $0.56_{\pm0.01}$ | $1.19_{\pm0.05}$ | $99.48_{\pm0.02}$ | $94.55_{\pm0.05}$ | 6.80 | **0.31** |
| IU | $17.51_{\pm2.19}$ | $21.39_{\pm1.70}$ | $98.00_{\pm0.38}$ | $98.11_{\pm0.38}$ | 5.48 | 1.18 |
| BE | $0.00_{\pm0.00}$ | $0.26_{\pm0.02}$ | $100.00_{\pm0.00}$ | $95.35_{\pm0.18}$ | 7.24 | 1.37 |
| BS | $0.37_{\pm0.10}$ | $1.10_{\pm0.43}$ | $99.93_{\pm0.01}$ | $98.97_{\pm0.02}$ | 7.86 | 1.21 |
| $\ell_1$-sparse | $2.80_{\pm0.37}$ | $19.59_{\pm3.48}$ | $99.07_{\pm0.04}$ | $98.00_{\pm0.12}$ | 3.69 | 1.98 |
| SalUn | $46.95_{\pm0.15}$ | $86.33_{\pm2.58}$ | $97.75_{\pm0.42}$ | $97.22_{\pm0.77}$ | 28.92 | 2.42 |
| UGradSL | $5.87_{\pm0.50}$ | $13.33_{\pm0.20}$ | $98.82_{\pm0.28}$ | $92.17_{\pm0.20}$ | **2.01** | 0.45 |
| UGradSL+ | $6.03_{\pm0.17}$ | $10.65_{\pm0.13}$ | $99.79_{\pm0.03}$ | $93.64_{\pm0.16}$ | 2.76 | 3.07 |
| **Random Set Size (20%)** | | | | | | |
| Retrain | 5.31 | 13.30 | 100.00 | 94.10 | - | 38.74 |
| FT | $0.76_{\pm4.55}$ | $2.69_{\pm10.61}$ | $99.89_{\pm0.11}$ | $93.97_{\pm0.13}$ | 3.85 | 2.17 |
| RL | $6.47_{\pm1.16}$ | $28.62_{\pm15.32}$ | $99.60_{\pm0.40}$ | $92.39_{\pm1.71}$ | 4.65 | 2.65 |
| GA | $0.67_{\pm4.64}$ | $1.44_{\pm11.86}$ | $99.48_{\pm0.52}$ | $94.42_{\pm0.32}$ | 4.33 | 0.26 |
| IU | $2.91_{\pm2.40}$ | $5.53_{\pm7.77}$ | $97.30_{\pm2.70}$ | $90.64_{\pm3.46}$ | 4.08 | 3.29 |
| BE | $0.57_{\pm4.74}$ | $1.64_{\pm11.66}$ | $99.44_{\pm0.56}$ | $94.32_{\pm0.22}$ | 4.29 | 0.53 |
| BS | $0.62_{\pm4.69}$ | $1.62_{\pm11.68}$ | $99.46_{\pm0.54}$ | $94.20_{\pm0.10}$ | 4.25 | 0.86 |
| $\ell_1$-sparse | $3.92_{\pm1.39}$ | $8.94_{\pm4.36}$ | $98.09_{\pm1.91}$ | $91.92_{\pm2.18}$ | 2.46 | 2.20 |
| SalUn | $3.73_{\pm1.58}$ | $13.18_{\pm0.12}$ | $98.61_{\pm1.39}$ | $92.75_{\pm1.35}$ | 1.11 | 2.66 |
| UGradSL | $6.07_{\pm0.70}$ | $13.82_{\pm1.03}$ | $95.71_{\pm0.17}$ | $90.19_{\pm0.23}$ | 2.37 | **0.24** |
| UGradSL+ | $6.39_{\pm0.19}$ | $12.34_{\pm1.79}$ | $97.08_{\pm0.44}$ | $90.91_{\pm0.95}$ | **2.04** | 0.31 |
| **Random Set Size (30%)** | | | | | | |
| Retrain | 6.64 | 14.60 | 100.00 | 92.78 | - | 33.65 |
| FT | $0.56_{\pm6.08}$ | $1.66_{\pm12.94}$ | $99.83_{\pm0.17}$ | $94.22_{\pm1.44}$ | 5.16 | 1.98 |
| RL | $6.89_{\pm0.25}$ | $31.09_{\pm16.49}$ | $99.36_{\pm0.64}$ | $91.35_{\pm1.43}$ | 4.70 | 2.63 |
| GA | $0.65_{\pm5.99}$ | $1.50_{\pm13.10}$ | $99.46_{\pm0.54}$ | $94.44_{\pm1.66}$ | 5.32 | 2.40 |
| IU | $3.95_{\pm2.69}$ | $7.26_{\pm7.34}$ | $96.22_{\pm3.78}$ | $89.61_{\pm3.17}$ | 4.24 | 3.32 |
| BE | $0.63_{\pm6.01}$ | $3.35_{\pm11.25}$ | $99.39_{\pm0.61}$ | $94.19_{\pm1.41}$ | 4.82 | 0.81 |
| BS | $0.63_{\pm6.01}$ | $2.88_{\pm11.72}$ | $99.39_{\pm0.61}$ | $94.15_{\pm1.37}$ | 4.93 | 1.28 |
| $\ell_1$-sparse | $4.70_{\pm1.94}$ | $9.97_{\pm4.63}$ | $97.63_{\pm2.37}$ | $91.19_{\pm1.59}$ | 2.63 | 1.99 |
| SalUn | $6.22_{\pm0.42}$ | $14.11_{\pm0.49}$ | $95.91_{\pm4.09}$ | $90.72_{\pm2.06}$ | 1.76 | 2.64 |
| UGradSL | $6.78_{\pm0.66}$ | $15.96_{\pm0.12}$ | $96.94_{\pm0.56}$ | $90.72_{\pm0.80}$ | 1.66 | **0.70** |
| UGradSL+ | $6.36_{\pm0.65}$ | $14.99_{\pm0.82}$ | $97.35_{\pm0.79}$ | $91.10_{\pm1.10}$ | **1.25** | 0.53 |
| **Random Set Size (40%)** | | | | | | |
| Retrain | 7.01 | 18.37 | 100.00 | 92.52 | - | 28.47 |
| FT | $0.77_{\pm6.24}$ | $2.88_{\pm15.49}$ | $99.96_{\pm0.04}$ | $94.27_{\pm1.75}$ | 5.88 | 1.62 |
| RL | $5.02_{\pm1.99}$ | $37.76_{\pm19.39}$ | $99.61_{\pm0.39}$ | $92.14_{\pm0.38}$ | 5.54 | 2.68 |
| GA | $0.67_{\pm6.34}$ | $1.57_{\pm16.80}$ | $99.47_{\pm0.53}$ | $94.38_{\pm1.86}$ | 6.38 | **0.53** |
| IU | $7.89_{\pm0.88}$ | $10.99_{\pm7.38}$ | $92.21_{\pm7.79}$ | $86.15_{\pm6.37}$ | 5.60 | 3.27 |
| BE | $0.86_{\pm6.15}$ | $15.72_{\pm2.65}$ | $99.27_{\pm0.73}$ | $93.46_{\pm0.94}$ | 2.62 | 1.04 |
| BS | $1.18_{\pm5.83}$ | $13.97_{\pm4.40}$ | $98.94_{\pm1.06}$ | $93.01_{\pm0.49}$ | 2.95 | 1.72 |
| $\ell_1$-sparse | $2.84_{\pm4.17}$ | $7.09_{\pm11.28}$ | $98.75_{\pm1.25}$ | $92.20_{\pm0.32}$ | 4.26 | 1.63 |
| SalUn | $6.86_{\pm0.15}$ | $15.15_{\pm3.22}$ | $95.01_{\pm4.99}$ | $89.76_{\pm2.76}$ | 2.78 | 2.67 |
| UGradSL | $5.81_{\pm0.11}$ | $14.98_{\pm2.65}$ | $97.31_{\pm1.06}$ | $90.73_{\pm0.48}$ | **2.27** | 0.62 |
| UGradSL+ | $5.82_{\pm0.37}$ | $14.53_{\pm1.83}$ | $97.11_{\pm0.40}$ | $90.74_{\pm0.38}$ | 2.42 | 0.63 |
| **Random Set Size (50%)** | | | | | | |
| Retrain | 7.91 | 19.29 | 100.00 | 91.72 | - | 23.90 |
| FT | $0.44_{\pm7.47}$ | $2.15_{\pm17.14}$ | $99.96_{\pm0.04}$ | $94.23_{\pm2.51}$ | 6.79 | 1.31 |
| RL | $7.61_{\pm0.30}$ | $37.36_{\pm18.07}$ | $99.67_{\pm0.33}$ | $92.83_{\pm1.11}$ | 4.95 | 2.65 |
| GA | $0.40_{\pm7.51}$ | $1.22_{\pm18.07}$ | $99.61_{\pm0.39}$ | $94.34_{\pm2.62}$ | 7.15 | **0.66** |
| IU | $3.97_{\pm3.94}$ | $7.29_{\pm12.00}$ | $96.21_{\pm3.79}$ | $90.00_{\pm1.72}$ | 5.36 | 3.25 |
| BE | $3.08_{\pm4.83}$ | $24.87_{\pm5.58}$ | $96.84_{\pm3.16}$ | $90.41_{\pm1.31}$ | 3.72 | 1.31 |
| BS | $9.76_{\pm1.85}$ | $32.15_{\pm12.86}$ | $90.19_{\pm9.81}$ | $83.71_{\pm8.01}$ | 8.13 | 2.12 |
| $\ell_1$-sparse | $1.44_{\pm6.47}$ | $4.76_{\pm14.53}$ | $99.52_{\pm0.48}$ | $93.13_{\pm1.41}$ | 5.72 | 1.31 |
| SalUn | $7.75_{\pm0.16}$ | $16.99_{\pm2.30}$ | $94.28_{\pm5.72}$ | $89.29_{\pm2.43}$ | 2.65 | 2.68 |
| UGradSL | $6.83_{\pm0.23}$ | $12.73_{\pm1.66}$ | $97.62_{\pm0.71}$ | $90.27_{\pm0.55}$ | 2.87 | 0.77 |
| UGradSL+ | $6.13_{\pm1.35}$ | $16.49_{\pm2.73}$ | $97.84_{\pm0.34}$ | $90.84_{\pm0.69}$ | **1.91** | 0.77 |

Table 18: MU Performance across different forgetting data amounts on ResNet-18, pre-trained on CIFAR-100 dataset, for random data forgetting.

| Method | Random Set Size (10%) | | | | | |
| | UA | $\mathrm{MIA_{Score}}$ | RA | TA | Avg. Gap ($\downarrow$) | RTE ($\downarrow$, min) |
|---|---|---|---|---|---|---|
| Retrain | 29.47 | 53.50 | 99.98 | 70.51 | - | 25.01 |
| FT | $2.55_{\pm0.03}$ | $10.59_{\pm0.27}$ | $99.95_{\pm0.01}$ | $75.95_{\pm0.05}$ | 18.83 | 1.95 |
| RL | $4.06_{\pm0.37}$ | $50.12_{\pm3.48}$ | $99.92_{\pm0.02}$ | $71.30_{\pm0.36}$ | 7.41 | 1.20 |
| GA | $2.58_{\pm0.06}$ | $5.95_{\pm0.17}$ | $97.45_{\pm0.02}$ | $76.09_{\pm0.01}$ | 20.64 | 0.29 |
| IU | $15.71_{\pm5.19}$ | $18.69_{\pm4.12}$ | $84.65_{\pm5.19}$ | $62.20_{\pm4.17}$ | 18.05 | 1.20 |
| BE | $0.01_{\pm0.00}$ | $1.45_{\pm0.02}$ | $98.22_{\pm1.26}$ | $78.26_{\pm0.00}$ | 22.76 | **0.24** |
| BS | $2.20_{\pm2.11}$ | $10.73_{\pm9.37}$ | $98.22_{\pm1.26}$ | $70.23_{\pm1.67}$ | 18.02 | 0.34 |
| $\ell_1$-sparse | $8.19_{\pm0.38}$ | $19.11_{\pm0.52}$ | $88.39_{\pm0.31}$ | $80.26_{\pm0.16}$ | 19.25 | 1.00 |
| SalUn | $35.23_{\pm0.32}$ | $89.39_{\pm0.46}$ | $99.53_{\pm0.04}$ | $64.26_{\pm0.58}$ | 12.10 | 3.33 |
| UGradSL | $18.36_{\pm0.17}$ | $40.71_{\pm0.13}$ | $98.38_{\pm0.03}$ | $68.23_{\pm0.16}$ | 6.95 | 0.55 |
| UGradSL+ | $21.69_{\pm0.59}$ | $49.47_{\pm1.25}$ | $99.87_{\pm0.34}$ | $73.60_{\pm0.26}$ | **3.75** | 3.52 |

| Method | Random Set Size (20%) | | | | | |
| | UA | $\mathrm{MIA_{Score}}$ | RA | TA | Avg. Gap ($\downarrow$) | RTE ($\downarrow$, min) |
|---|---|---|---|---|---|---|
| Retrain | 26.84 | 52.41 | 99.99 | 73.88 | - | 36.88 |
| FT | $2.70_{\pm24.14}$ | $11.63_{\pm40.78}$ | $99.95_{\pm0.04}$ | $75.51_{\pm1.63}$ | 16.65 | 2.05 |
| RL | $54.74_{\pm27.90}$ | $97.32_{\pm44.91}$ | $99.47_{\pm0.52}$ | $65.59_{\pm8.29}$ | 20.41 | 2.11 |
| GA | $6.79_{\pm20.05}$ | $13.22_{\pm39.19}$ | $94.11_{\pm5.88}$ | $71.39_{\pm2.49}$ | 16.90 | **0.26** |
| IU | $5.34_{\pm21.50}$ | $11.79_{\pm40.62}$ | $95.54_{\pm4.45}$ | $70.89_{\pm2.99}$ | 17.39 | 3.77 |
| BE | $2.51_{\pm24.33}$ | $6.70_{\pm45.71}$ | $97.38_{\pm2.61}$ | $75.07_{\pm1.19}$ | 18.46 | 0.49 |
| BS | $2.53_{\pm24.31}$ | $6.57_{\pm45.84}$ | $97.38_{\pm2.61}$ | $75.05_{\pm1.17}$ | 18.48 | 0.82 |
| $\ell_1$-sparse | $37.83_{\pm10.99}$ | $38.90_{\pm13.51}$ | $76.63_{\pm23.36}$ | $58.79_{\pm15.09}$ | 15.74 | 2.05 |
| SalUn | $25.83_{\pm1.01}$ | $64.69_{\pm12.28}$ | $96.01_{\pm3.98}$ | $65.87_{\pm8.01}$ | 6.32 | 2.12 |
| UGradSL | $30.10_{\pm1.03}$ | $47.39_{\pm1.17}$ | $93.49_{\pm0.24}$ | $64.99_{\pm0.04}$ | **5.92** | 0.83 |
| UGradSL+ | $27.29_{\pm0.99}$ | $35.92_{\pm0.94}$ | $93.36_{\pm0.03}$ | $66.59_{\pm0.37}$ | 7.71 | 0.59 |

| Method | Random Set Size (30%) | | | | | |
| | UA | $\mathrm{MIA_{Score}}$ | RA | TA | Avg. Gap ($\downarrow$) | RTE ($\downarrow$, min) |
|---|---|---|---|---|---|---|
| Retrain | 28.52 | 52.24 | 99.98 | 70.91 | - | 32.92 |
| FT | $2.65_{\pm25.87}$ | $11.18_{\pm41.06}$ | $99.94_{\pm0.04}$ | $75.17_{\pm4.26}$ | 17.81 | 1.44 |
| RL | $51.46_{\pm22.94}$ | $96.34_{\pm44.10}$ | $99.32_{\pm0.66}$ | $62.77_{\pm8.14}$ | 18.96 | 2.14 |
| GA | $2.40_{\pm26.12}$ | $5.70_{\pm46.54}$ | $97.39_{\pm2.59}$ | $75.33_{\pm4.42}$ | 19.92 | **0.40** |
| IU | $5.96_{\pm22.56}$ | $12.63_{\pm39.61}$ | $94.59_{\pm5.39}$ | $69.74_{\pm1.17}$ | 17.18 | 3.76 |
| BE | $2.44_{\pm26.08}$ | $6.53_{\pm45.71}$ | $97.37_{\pm2.61}$ | $74.77_{\pm3.86}$ | 19.56 | 0.76 |
| BS | $2.49_{\pm26.03}$ | $6.40_{\pm45.84}$ | $97.33_{\pm2.65}$ | $74.65_{\pm3.74}$ | 19.56 | 1.24 |
| $\ell_1$-sparse | $38.45_{\pm9.93}$ | $38.52_{\pm13.72}$ | $76.36_{\pm23.62}$ | $58.09_{\pm12.82}$ | 15.02 | 1.47 |
| SalUn | $27.34_{\pm1.18}$ | $62.99_{\pm10.75}$ | $94.50_{\pm5.48}$ | $63.10_{\pm7.81}$ | 6.31 | 2.16 |
| UGradSL | $30.10_{\pm0.12}$ | $47.39_{\pm2.08}$ | $93.49_{\pm0.74}$ | $64.99_{\pm1.53}$ | **4.71** | 0.83 |
| UGradSL+ | $24.89_{\pm0.24}$ | $44.60_{\pm0.94}$ | $94.90_{\pm0.88}$ | $66.16_{\pm0.78}$ | 5.28 | 0.79 |

| Method | Random Set Size (40%) | | | | | |
| | UA | $\mathrm{MIA_{Score}}$ | RA | TA | Avg. Gap ($\downarrow$) | RTE ($\downarrow$, min) |
|---|---|---|---|---|---|---|
| Retrain | 30.07 | 58.06 | 99.99 | 69.87 | - | 28.29 |
| FT | $2.66_{\pm27.41}$ | $11.05_{\pm47.01}$ | $99.95_{\pm0.04}$ | $75.35_{\pm5.48}$ | 19.99 | 1.51 |
| RL | $51.75_{\pm21.68}$ | $95.78_{\pm37.72}$ | $99.27_{\pm0.72}$ | $59.41_{\pm10.46}$ | 17.64 | 2.12 |
| GA | $2.46_{\pm27.61}$ | $5.91_{\pm52.15}$ | $97.39_{\pm2.60}$ | $75.40_{\pm5.53}$ | 21.97 | **0.51** |
| IU | $4.58_{\pm25.49}$ | $10.32_{\pm47.74}$ | $96.29_{\pm3.70}$ | $70.92_{\pm1.05}$ | 19.49 | 3.78 |
| BE | $2.54_{\pm27.53}$ | $7.44_{\pm50.62}$ | $97.35_{\pm2.64}$ | $74.56_{\pm4.69}$ | 21.37 | 1.00 |
| BS | $2.70_{\pm27.37}$ | $7.63_{\pm50.43}$ | $97.26_{\pm2.73}$ | $74.10_{\pm4.23}$ | 21.19 | 1.66 |
| $\ell_1$-sparse | $38.49_{\pm8.42}$ | $40.21_{\pm17.85}$ | $78.43_{\pm21.56}$ | $57.66_{\pm12.21}$ | 15.01 | 1.52 |
| SalUn | $25.54_{\pm4.53}$ | $60.08_{\pm2.02}$ | $94.64_{\pm5.35}$ | $62.52_{\pm7.35}$ | 4.81 | 2.14 |
| UGradSL | $30.07_{\pm1.58}$ | $49.23_{\pm1.07}$ | $95.30_{\pm0.34}$ | $64.52_{\pm0.28}$ | **4.72** | 1.08 |
| UGradSL+ | $30.42_{\pm0.77}$ | $45.94_{\pm1.41}$ | $93.98_{\pm0.50}$ | $63.21_{\pm0.35}$ | 6.29 | 0.77 |

| Method | Random Set Size (50%) | | | | | |
| | UA | $\mathrm{MIA_{Score}}$ | RA | TA | Avg. Gap ($\downarrow$) | RTE ($\downarrow$, min) |
|---|---|---|---|---|---|---|
| Retrain | 32.69 | 61.15 | 99.99 | 67.22 | - | 25.01 |
| FT | $2.71_{\pm29.98}$ | $10.71_{\pm50.44}$ | $99.96_{\pm0.03}$ | $75.11_{\pm7.89}$ | 22.08 | 1.25 |
| RL | $50.52_{\pm17.83}$ | $95.91_{\pm34.76}$ | $99.47_{\pm0.52}$ | $56.75_{\pm10.47}$ | 15.90 | 2.13 |
| GA | $2.61_{\pm30.08}$ | $5.92_{\pm55.23}$ | $97.49_{\pm2.50}$ | $75.27_{\pm8.05}$ | 23.97 | **0.66** |
| IU | $12.64_{\pm20.05}$ | $17.54_{\pm43.61}$ | $87.96_{\pm12.03}$ | $62.76_{\pm4.46}$ | 20.04 | 3.80 |
| BE | $2.76_{\pm29.93}$ | $8.85_{\pm52.30}$ | $97.39_{\pm2.60}$ | $74.05_{\pm6.83}$ | 22.92 | 1.26 |
| BS | $2.99_{\pm29.70}$ | $8.76_{\pm52.39}$ | $97.24_{\pm2.75}$ | $73.38_{\pm6.16}$ | 22.75 | 2.08 |
| $\ell_1$-sparse | $39.86_{\pm7.17}$ | $40.43_{\pm20.72}$ | $78.17_{\pm21.82}$ | $55.65_{\pm11.57}$ | 15.32 | 1.26 |
| SalUn | $26.17_{\pm6.52}$ | $59.47_{\pm1.68}$ | $94.04_{\pm5.95}$ | $61.39_{\pm5.83}$ | 5.00 | 2.13 |
| UGradSL | $33.80_{\pm1.61}$ | $57.38_{\pm2.31}$ | $95.29_{\pm0.11}$ | $56.88_{\pm0.80}$ | **4.98** | 0.95 |
| UGradSL+ | $32.20_{\pm0.49}$ | $49.20_{\pm1.44}$ | $94.47_{\pm0.69}$ | $61.53_{\pm0.97}$ | 5.91 | 0.75 |

Table 19: The difference between GA and FT as well as UGradSL and UGradSL+ on CIFAR-10 regarding the number of epochs. The forgetting paradigm is random forgetting.

| Epoch | Gradient Ascent | | | | | Fine-tuning | | | | |
|---|---|---|---|---|---|---|---|---|---|---|
| | UA | MIA$_{Score}$ | RA | TA | Avg. Gap ($\downarrow$) | UA | MIA$_{Score}$ | RA | TA | Avg. Gap ($\downarrow$) |
| 5 | 0 | 0.32 | 95.31 | 100 | 9.56 | 0.04 | 0.34 | 95.13 | 99.99 | 9.59 |
| 6 | 0 | 0.40 | 95.34 | 100 | 9.53 | - | - | - | - | - |
| 7 | 0.82 | 2.22 | 93.24 | 99.26 | 9.21 | - | - | - | - | - |
| 8 | 3.44 | 4.78 | 90.80 | 96.18 | 7.76 | - | - | - | - | - |
| 9 | 10.34 | 12.76 | 83.42 | 89.00 | 6.53 | - | - | - | - | - |
| 10 | 76.26 | 72.22 | 6.49 | 24.24 | 70.97 | 0.04 | 0.24 | 94.97 | 99.99 | 9.65 |
| 15 | - | - | - | - | - | 0.02 | 0.80 | 94.68 | 99.96 | 9.58 |

| Epoch | UGradSL | | | | | UGradSL+ | | | | |
|---|---|---|---|---|---|---|---|---|---|---|
| | UA | MIA$_{Score}$ | RA | TA | Avg. Gap ($\downarrow$) | UA | MIA$_{Score}$ | RA | TA | Avg. Gap ($\downarrow$) |
| 10 | 14.98 | 33.22 | 77.18 | 84.07 | 13.27 | 6.26 | 14.10 | 93.39 | 99.62 | 4.94 |
| 11 | 24.26 | 34.38 | 68.22 | 75.06 | 20.37 | 6.52 | 11.66 | 93.04 | 99.37 | 5.51 |
| 12 | 28.70 | 24.62 | 68.17 | 74.39 | 19.22 | 21.46 | 27.38 | 89.41 | 97.07 | 9.85 |
| 13 | 38.46 | 72.90 | 61.78 | 64.72 | 37.75 | 29.48 | 31.92 | 87.74 | 94.93 | 12.88 |
| 14 | 99.86 | 86.74 | 0.45 | 0.20 | 88.02 | 31.62 | 32.68 | 86.53 | 93.36 | 13.51 |
| Retrain | 8.07 | 17.41 | 100.00 | 91.61 | - | 8.07 | 17.41 | 100.00 | 91.61 | - |

