# OpenReview forum: "Label Smoothing Improves Machine Unlearning"
_ICLR.cc/2026/Conference — ICLR 2026 Poster_

### Official Review · Reviewer_Zyp9 · 2025-10-26

**Soundness:** 3
**Presentation:** 2
**Contribution:** 3
**Rating:** 6
**Confidence:** 3

**Summary:**

The paper proposes a gradient-based machine unlearning (MU) method, the UGradSL, that integrates label smoothing (LS) into the unlearning in the ways of fine-tuning and gradient accent.
The key idea is to perform gradient ascent with generalized label smoothing (GLS) on the forgetting set $D_f$, and gradient descent on the retaining set $D_r$, forming a mixed-gradient optimization.
The authors show a theoretical condition under which NLS improves gradient ascent.
Experiments across multiple datasets and model architectures (ResNet18, ViT, BERT) show improved unlearning–retaining trade-offs with small computational overhead.

**Strengths:**

1. Easily pluggable into existing GA/FT unlearning pipelines without retraining.
2. Broad Empirical Evaluation. Tested on six datasets and multiple unlearning types (class, random, group), consistently outperforming baselines.
3. Provided theoretical intuitions. Explained why NLS can guide ascent toward the equivalent retrained solution.

**Weaknesses:**

1. The condition of the inner product of $\Delta\theta$'s in Theorem 2 is critical but only verified empirically on one dataset (CelebA, Figure 4 in the Appendix). This lacks theoretical justification and broader empirical validation.

2. Theorem 3's connection to LDP is interesting but the practical meaning is unclear - it provides label-level privacy for the forgetting set, but doesn't directly translate to model-level unlearning guarantees.

3. Hyperparameter sensitivity needs ablation studies. A demonstration about the smoothing rate $\alpha$'s influence on performance is expected. While an adaptive version is proposed, the distance threshold $\beta$ introduces another parameter.

**Questions:**

Besides the weaknesses:
1. In Algorithm 1, the distance computation $d(z^r_i, z^f_i)$ is not clearly defined. What distance metric? In which space (feature/parameter)?

---

> ### Author Response · Authors · 2025-11-26
> **Response from the author (1)**
>
> Thank you for the very careful and detailed review. We appreciate you highlighting both the strengths and the gaps, and we agree that several parts of the pipeline description were not as clear as they should be. Below we address each concern point-by-point and indicate how we will update the paper.
>
> **W1** We appreciate the reviewer's suggestion to make this paper stronger. We agree with the reviewer's concern and try more datasets and models. We try the class-wise forgetting on ImageNet using ResNet-18 model, class-wise forgetting on CIFAR-100 using ViT and random forgetting on CIFAR-10 using VGG-16 model. All the inner production is negative. We attach the histogram to the paper on Appendix E.11.
>
> **W2** We thank the reviewer for raising this important point. We clarify below how Theorem 3 yields a practically meaningful model-level guarantee, even though the formal statement is phrased in terms of label-level local DP.
>
> As widely acknowledged in prior work on machine unlearning [1, 2, 3], there is currently no universally accepted model-level unlearning guarantee for deep neural networks. Theoretical analysis at the parameter level typically relies on strong and unrealistic assumptions, and the non-convex optimization trajectory makes parameter-level reasoning effectively intractable. We evaluate model-level capability through the model’s predictions, rather than its parameters because:
> - A model’s functional capability is reflected entirely through its predictions, not its parameter vectors.
> - DP on predictions directly implies model stability with respect to the removed labels, which is precisely the operational notion of unlearning used across DP-based unlearning work.
> - Since the parameters of deep models are inherently uninterpretable, prediction-level guarantees are the only meaningful and verifiable model-level guarantees available in modern deep learning.
>
> From this aspect, Theorem 3 provides a concrete and practically meaningful form of model-level, privacy-style unlearning guarantee for the forgotten labels.
>
> [1] Ginart, Antonio, et al. "Making ai forget you: Data deletion in machine learning." Advances in neural information processing systems 32 (2019).
>
> [2] Guo, Chuan, et al. "Certified data removal from machine learning models." arXiv preprint arXiv:1911.03030 (2019).
>
> [3] Sekhari, Ayush, et al. "Remember what you want to forget: Algorithms for machine unlearning." Advances in Neural Information Processing Systems 34 (2021): 18075-18086.
>
> **W3, Q1**
>
> Thank you for your insightful question, which helps us present the paper more rigorously. We conduct ablation studies on the distance threshold $\beta$ and different distance functions for UGradSL under the random forgetting setting on CIFAR-10, where the forgetting set size is 10% of the training data. By default, we compute distances in the feature space: we first extract the feature representation of each sample using the current model, and then calculate pairwise distances between retained and forgetting samples. The default distance function is the cosine distance, and the default threshold $\beta$ is set to the median of all distances. To avoid ambiguity, we have also fixed the notation and now explicitly denote the distance as $d(h_{\theta}(x_i^r), h_{\theta}(x_j^f))$. The experiment results are shown below.

---

> ### Author Response · Authors · 2025-11-26
> **Response from the author (2)**
>
> **CIFAR-10 (Random Forgetting, 10%, UGradSL)**
>
> | **$\beta$** | **Distance** | **UA**             | **MIA**            | **RA**             | **TA**             | **Avg. Gap (↓)** |
> | ----------- | ------------ | ------------------ | ------------------ | ------------------ | ------------------ | ---------------- |
> | -           | -            | $8.07$             | $17.41$            | $100.00$           | $91.61$            | -                |
> | Median      | Cosine       | $7.54_{\pm 0.43}$  | $13.57_{\pm 0.12}$ | $99.67_{\pm 0.00}$ | $92.97_{\pm 0.17}$ | $1.52$           |
> |             | Euclidean    | $11.21_{\pm 0.21}$ | $21.02_{\pm 2.23}$ | $94.35_{\pm 0.22}$ | $88.58_{\pm 0.26}$ | $3.86$           |
> | 0.1         | Cosine       | $7.79_{\pm 0.52}$  | $17.04_{\pm 0.61}$ | $95.84_{\pm 0.27}$ | $90.10_{\pm 0.47}$ | $1.67$           |
> |             | Euclidean    | $7.30_{\pm 0.62}$  | $16.42_{\pm 0.66}$ | $96.16_{\pm 0.94}$ | $90.46_{\pm 0.91}$ | $1.69$           |
> | 0.2         | Cosine       | $8.38_{\pm 0.19}$  | $17.46_{\pm 1.09}$ | $95.38_{\pm 0.34}$ | $89.56_{\pm 0.53}$ | $1.94$           |
> |             | Euclidean    | $7.80_{\pm 0.76}$  | $16.55_{\pm 1.91}$ | $95.75_{\pm 1.04}$ | $89.80_{\pm 0.50}$ | $1.93$           |
> | 0.3         | Cosine       | $8.27_{\pm 0.65}$  | $18.19_{\pm 0.29}$ | $95.94_{\pm 0.84}$ | $90.18_{\pm 0.62}$ | $1.71$           |
> |             | Euclidean    | $7.68_{\pm 0.65}$  | $17.28_{\pm 0.52}$ | $95.85_{\pm 0.75}$ | $90.25_{\pm 0.55}$ | $1.62$           |
> | 0.4         | Cosine       | $8.49_{\pm 0.28}$  | $17.92_{\pm 0.52}$ | $95.85_{\pm 0.20}$ | $90.09_{\pm 0.03}$ | $1.66$           |
> |             | Euclidean    | $8.38_{\pm 0.60}$  | $17.86_{\pm 0.89}$ | $95.60_{\pm 0.78}$ | $90.06_{\pm 0.57}$ | $1.80$           |
> | 0.5         | Cosine       | $9.23_{\pm 0.89}$  | $16.81_{\pm 1.66}$ | $95.46_{\pm 0.62}$ | $89.79_{\pm 0.86}$ | $2.15$           |
> |             | Euclidean    | $8.98_{\pm 0.69}$  | $16.77_{\pm 1.62}$ | $95.39_{\pm 1.01}$ | $89.34_{\pm 1.17}$ | $2.31$           |
> | 0.6         | Cosine       | $9.95_{\pm 0.64}$  | $19.90_{\pm 0.95}$ | $95.47_{\pm 0.12}$ | $89.82_{\pm 0.30}$ | $2.67$           |
> |             | Euclidean    | $10.00_{\pm 0.10}$ | $19.00_{\pm 1.92}$ | $95.15_{\pm 0.26}$ | $89.53_{\pm 0.28}$ | $2.66$           |
> | 0.7         | Cosine       | $11.81_{\pm 0.74}$ | $20.67_{\pm 2.62}$ | $94.25_{\pm 0.76}$ | $88.78_{\pm 1.02}$ | $3.90$           |
> |             | Euclidean    | $11.25_{\pm 0.59}$ | $21.54_{\pm 1.12}$ | $94.69_{\pm 0.71}$ | $89.05_{\pm 0.71}$ | $3.79$           |
> | 0.8         | Cosine       | $13.06_{\pm 0.53}$ | $18.81_{\pm 0.81}$ | $92.89_{\pm 0.69}$ | $87.29_{\pm 0.75}$ | $4.45$           |
> |             | Euclidean    | $12.07_{\pm 0.45}$ | $19.23_{\pm 2.00}$ | $93.81_{\pm 0.95}$ | $88.34_{\pm 1.00}$ | $3.82$           |
> | 0.9         | Cosine       | $11.75_{\pm 0.09}$ | $21.02_{\pm 1.43}$ | $94.34_{\pm 0.38}$ | $88.81_{\pm 0.31}$ | $3.94$           |
> |             | Euclidean    | $12.01_{\pm 1.12}$ | $21.49_{\pm 1.17}$ | $94.26_{\pm 1.08}$ | $88.74_{\pm 0.88}$ | $4.16$           |
> | 1.0         | Cosine       | $11.48_{\pm 0.06}$ | $20.59_{\pm 2.63}$ | $94.19_{\pm 0.56}$ | $88.82_{\pm 0.38}$ | $3.80$           |
> |             | Euclidean    | $11.79_{\pm 0.37}$ | $17.35_{\pm 0.85}$ | $94.37_{\pm 0.34}$ | $88.67_{\pm 0.56}$ | $3.23$           |

---

> ### Author Response · Authors · 2025-11-26
> **Response from the author (3)**
>
> **CIFAR-10 (Random Forgetting, 10%, UGradSL+)**
>
> | $\beta$ | Distance  | UA                 | MIA                | RA                 | TA                 | Avg. Gap ($\downarrow$) |
> | ------- | --------- | ------------------ | ------------------ | ------------------ | ------------------ | ----------------------- |
> | -       | -         | $8.07$             | $17.41$            | $100.00$           | $91.61$            | -                       |
> | Median  | Cosine    | $6.04_{\pm 0.11}$  | $13.75_{\pm 0.32}$ | $99.11_{\pm 0.01}$ | $92.07_{\pm 0.02}$ | $1.76$                  |
> |         | Euclidean | $8.59_{\pm 1.85}$  | $17.30_{\pm 0.98}$ | $94.39_{\pm 1.15}$ | $88.97_{\pm 1.12}$ | $2.59$                  |
> | 0.1     | Cosine    | $6.50_{\pm 0.14}$  | $14.76_{\pm 1.52}$ | $95.64_{\pm 0.23}$ | $89.91_{\pm 0.17}$ | $2.57$                  |
> |         | Euclidean | $6.68_{\pm 0.88}$  | $14.69_{\pm 1.66}$ | $95.34_{\pm 0.79}$ | $89.90_{\pm 0.69}$ | $2.62$                  |
> | 0.2     | Cosine    | $7.01_{\pm 0.67}$  | $15.86_{\pm 0.86}$ | $95.18_{\pm 0.44}$ | $89.69_{\pm 0.19}$ | $2.34$                  |
> |         | Euclidean | $6.82_{\pm 0.44}$  | $15.81_{\pm 0.70}$ | $95.58_{\pm 0.73}$ | $90.02_{\pm 0.57}$ | $2.21$                  |
> | 0.3     | Cosine    | $7.01_{\pm 0.98}$  | $15.13_{\pm 1.26}$ | $95.24_{\pm 0.99}$ | $89.76_{\pm 0.41}$ | $2.49$                  |
> |         | Euclidean | $7.32_{\pm 1.06}$  | $16.45_{\pm 2.08}$ | $94.68_{\pm 0.89}$ | $89.16_{\pm 0.33}$ | $2.53$                  |
> | 0.4     | Cosine    | $7.91_{\pm 0.26}$  | $15.69_{\pm 1.11}$ | $94.69_{\pm 0.51}$ | $89.07_{\pm 0.29}$ | $2.45$                  |
> |         | Euclidean | $6.24_{\pm 0.21}$  | $14.16_{\pm 0.12}$ | $95.75_{\pm 0.40}$ | $90.13_{\pm 0.13}$ | $2.70$                  |
> | 0.5     | Cosine    | $7.61_{\pm 0.66}$  | $16.50_{\pm 1.68}$ | $95.03_{\pm 0.36}$ | $89.69_{\pm 0.72}$ | $2.26$                  |
> |         | Euclidean | $8.27_{\pm 1.33}$  | $16.44_{\pm 1.83}$ | $94.67_{\pm 1.33}$ | $89.03_{\pm 1.28}$ | $2.68$                  |
> | 0.6     | Cosine    | $8.76_{\pm 0.28}$  | $16.53_{\pm 1.88}$ | $94.31_{\pm 0.61}$ | $88.54_{\pm 0.50}$ | $2.75$                  |
> |         | Euclidean | $8.67_{\pm 0.28}$  | $17.01_{\pm 2.43}$ | $94.34_{\pm 0.16}$ | $88.93_{\pm 0.30}$ | $2.66$                  |
> | 0.7     | Cosine    | $9.88_{\pm 1.05}$  | $18.33_{\pm 2.82}$ | $93.55_{\pm 0.92}$ | $88.08_{\pm 0.42}$ | $3.44$                  |
> |         | Euclidean | $9.61_{\pm 0.86}$  | $17.93_{\pm 2.33}$ | $94.11_{\pm 0.49}$ | $88.69_{\pm 0.19}$ | $2.99$                  |
> | 0.8     | Cosine    | $9.61_{\pm 1.12}$  | $16.91_{\pm 1.51}$ | $93.68_{\pm 1.20}$ | $88.48_{\pm 0.76}$ | $3.08$                  |
> |         | Euclidean | $9.75_{\pm 0.17}$  | $16.79_{\pm 0.52}$ | $93.87_{\pm 0.02}$ | $88.34_{\pm 0.39}$ | $2.93$                  |
> | 0.9     | Cosine    | $9.19_{\pm 0.66}$  | $17.84_{\pm 0.72}$ | $94.19_{\pm 0.50}$ | $88.51_{\pm 0.84}$ | $2.63$                  |
> |         | Euclidean | $9.76_{\pm 0.49}$  | $18.61_{\pm 0.65}$ | $93.90_{\pm 0.39}$ | $88.47_{\pm 0.35}$ | $3.03$                  |
> | 1.0     | Cosine    | $9.39_{\pm 0.07}$  | $16.94_{\pm 0.26}$ | $94.26_{\pm 0.33}$ | $88.74_{\pm 0.22}$ | $2.60$                  |
> |         | Euclidean | $10.41_{\pm 0.24}$ | $19.16_{\pm 1.08}$ | $93.50_{\pm 0.63}$ | $88.21_{\pm 0.34}$ | $3.50$                  |
>
> We also discuss our algorithm further including the motivation and complex analysis in the general response [above](https://openreview.net/forum?id=X74KnsoYEM&noteId=nUintKWbqY) and Appendix D. We appreciate the reviewer's effort and insight again!

---

### Official Review · Reviewer_Xe6w · 2025-10-28

**Soundness:** 2
**Presentation:** 2
**Contribution:** 2
**Rating:** 2
**Confidence:** 4

**Summary:**

This paper integrates label smoothing into the unlearning loss to reduce time cost and mitigate drops in remaining and test accuracies. It proves that gradient ascent can achieve exact unlearning, and generalized label smoothing tightens errors. Building on this, it introduces UGradSL and UGradSL+, which iterate over retained and forgetting datasets, and show gains on class, random, and sub-class unlearning versus baselines.

**Strengths:**

1. Label smoothing combined with a gradient-based method is interesting for handling unlearning

2. Both theoretical and experimental evidence support the effectiveness of the proposed method.

**Weaknesses:**

1. The presentation of the technical sections is not clear, for example, why there exists an $\approx$ symbol in the condition of Theorem 1 and why $\epsilon$ in the conclusion of $\epsilon$-Label-LDP relies on the weights $\gamma_1$ and $\gamma_2$.

2. The proposed method requires calculating the distance with samples in the minibatch, which will lead to a large computation cost.

3. The proposed UGradSL does not work better in performance, while UGradSL+ requires a longer time than other baselines.

4. This paper does not contain essential abolition studies for the GA ratio and the optional smoothing ratio

**Questions:**

Please refer to the weaknesses.

---

> ### Author Response · Authors · 2025-11-26
> **Response from the author**
>
> **W1** We are grateful for the concern about the clarity to make this paper better.
> - For Theorem 1, we expand the ERM objective around $\theta_{tr}^\ast$ using a Taylor series, as described in Line 153. By retaining only the first- and second-order terms, we obtain approximate expressions for the model differences $\theta_{f}^\ast - \theta_{tr}^\ast$ and $\theta_{r}^\ast - \theta_{tr}^\ast$. Since we drop the Lagrange remainder terms, these relations are only approximate, so we consistently use the symbol $\approx$. The expressions in Theorem 1 are derived from Equation (5), and therefore we also use the approximate symbol there.
> - $\gamma_1$ and $\gamma_2$ represents the weight of machine learning and machine unlearning on ERM. $\epsilon$ stands for how good the label-DP is protected. DP is affected by the learning and unlearning process. Thus, it is influenced by $\gamma_1$ and $\gamma_2$.
> We have added the explanation in the main text. Furthermore, we refine our notation to make everything more clear without any ambiguity. We hope our explanation and refinement can address your concern and we are happy to answer your question if any.
>
> **W2** Thanks for your concern. We add a complete FLOP and memory consumption analysis in the general response [above](https://openreview.net/forum?id=X74KnsoYEM&noteId=nUintKWbqY). The conclusion is that the additional distance computation is all in batch wise and on GPU without any for loop. The additional time and memory consumption is very little.
>
> **W3** Thank you for the comment. Our goal is to design plug-and-play, **comprehensive machine unlearning methods** that work across diverse unlearning scenarios. Across all settings in Tables 1–4, **either UGradSL or UGradSL+ achieves the best overall trade-off** (lowest Avg. Gap) among all baselines, and UGradSL alone ranks in the top 2 on many benchmarks while remaining very fast (e.g., ImageNet class-wise: 1st; CIFAR-100 random: 2nd; CIFAR-20 group: 1st; CelebA group: 1st; CIFAR-10 random A: 1st). UGradSL is therefore intended as an **efficient variant** with **strong performance**, while UGradSL+ is a **more stable and comprehensive variant** that further improves Avg. Gap at a moderate additional cost. In terms of runtime, UGradSL+ is **comparable to other strong baselines** such as SalUN [1] and GLI [2] on CIFAR-100 (class-wise and random), Tiny-ImageNet (random), CelebA (group), and CIFAR-10 (class-wise and random), while consistently attaining a **better Avg. Gap** (e.g., on CIFAR-100 class-wise). We thus view UGradSL and UGradSL+ as two **Pareto-optimal options**, one favoring speed and one favoring the best unlearning–utility–privacy trade-off so that practitioners can choose the appropriate variant based on their computational budget and performance requirements. We hope our clarification can help the reviewer understand the design of UGradSL and UGradSL+ so as to address the concern.
>
> [1] Fan, Chongyu, et al. "Salun: Empowering machine unlearning via gradient-based weight saliency in both image classification and generation." arXiv preprint arXiv:2310.12508 (2023).
>
> [2]  Towards Efficient Machine Unlearning with Data Augmentation: Guided Loss-Increasing (GLI) to Prevent the Catastrophic Model Utility Drop
>
> **W4** Thank you for raising an important question. The ablation study of the GA ratio $p$ and smoothing ratio $\alpha$ have been given in the general response above ([link1](https://openreview.net/forum?id=X74KnsoYEM&noteId=sKv4HiRvat), [link2](https://openreview.net/forum?id=X74KnsoYEM&noteId=37OiTc3Msg), [link3](https://openreview.net/forum?id=X74KnsoYEM&noteId=Jh2g3frBxO)). The conclusion is that **although there is some performance fluctuation when $\alpha$ and $p$ change, our methods are relatively stable**.

---

> ### Comment · Reviewer_Xe6w · 2025-11-27
>
> Thanks for the authors ' reply. I will raise my rating to 4 and lower my confidence level. I still have two following questions:
>
> W1.1: The two terms are approximately equal in Theorem 1. However, what error value between the two terms is needed to meet the iff requirements is still underexplored?
>
> W2: In Tables 1, 2, 3, and 4, the time cost of most experiment settings for UGradSL+ is significantly higher than other baselines. How to support the claim "The additional time consumption is very little."
>
>  If the above two concerns are addressed, I will continually raise my rating.

---

> > ### Author Response · Authors · 2025-12-03
> > **Response from the author**
> >
> > We appreciate the reviewer's further question and we hope our response can address the reviewers' concern.
> >
> > W1. We bound the error of using the linearized difference $\Delta\theta_r - \Delta\theta_f$ to approximate the true model difference $\theta_r^{\ast} - \theta_f^{\ast}$. We can write
> > $$
> > \theta_r^{\ast} - \theta_f^{\ast}
> > = (\Delta\theta_r - \Delta\theta_f) + (e_r - e_f),
> > $$
> > so the approximation error is $e_r - e_f$.
> >
> > Assuming that $q(\theta) = \nabla R_{tr}(\theta)$ and $p(\theta) = \nabla R_{tr}(\theta) - \nabla R_f(\theta)$ have Lipschitz-continuous Hessians with constants $L_q$ and $L_p$, and letting $H_r$ and $H_f$ be the corresponding Hessians, standard Taylor bounds show that the error is second order in $|\theta_r^{\ast} - \theta_{tr}^{\ast}|$ and $|\theta_f^{\ast} - \theta_{tr}^{\ast}|$:
> > $$
> > |\theta_r^{\ast} - \theta_f^{\ast} - (\Delta\theta_r - \Delta\theta_f)|
> > \le
> > \frac{L_q}{2},|H_r^{-1}|,|\theta_r^{\ast} - \theta_{tr}^{\ast}|^2
> > +
> > \frac{L_p}{2},|H_f^{-1}|,|\theta_f^{\ast} - \theta_{tr}^{\ast}|^2.
> > $$
> >
> > The full derivation is given in the appendix C.2 in the updated document.
> >
> > W2. For the efficiency issue of UGradSL+, UGradSL+ is generally comparable with the methods like SaLUN, SRCUB, GLI, PABI, etc. It is not significantly high. For example, UGradSL+ takes 3.37 and 3.52 for class-wise and random forgetting on CIFAR-100 while SaLUN takes 2.15 and 3.33, which is not significantly higher. We hope the reviewer can understand that the framework of UGradSL and UGradSL+ and their effectiveness on all the scenarios with little addition to the computation cost.

---

### Official Review · Reviewer_dpJc · 2025-10-31

**Soundness:** 3
**Presentation:** 3
**Contribution:** 2
**Rating:** 4
**Confidence:** 4

**Summary:**

This paper concentrates on the low-computation and high-efficiency machine unlearning in the field of computer vision. The author proposed a novel method UGradSL and UGradSL+ (finetune based) combine the GA (gradient ascent) with the NLS (negative label smoothing) and apply mix-gradient strategy which perform GD (gradient descent) on the retain data and GA with NLS on the forget data. Moreover, the stronger variant self-adaptive UGradSL supports the automatic selection of the smooth rate. The paper provides the theoretical analysis and abundant experiments to prove the effectiveness and robustness of method. In addition, the study also explored the label smoothing and local differential privacy (LDP).

**Strengths:**

+ This paper is clear, logical and easy to understand. The tables and figure are clear and detailed, with good instructions.
+ The proposed method is a simple and plug-and-play tool, which can directly integrate into the existing gradient-based unlearning methods (such as GA and finetune) and improve their performance.
+ This paper provides the mathematical and theory proof to explain the limitation of existing GA methods and the effectiveness of NLS in specific circumstances. Moreover, the authors innovatively establish the connection between label smoothing and LDP.

**Weaknesses:**

- Although the baseline methods are representative, the experiments lack comparison with the latest schemes between 2024 and 2025.
- Some symbols and formulas lack precise definitions and explanations or exist clerical, such as “distance d()” in the Algorithm 1 where different distance calculation methods can lead to the difference between computational overhead and performance.
- The performance of the method is sensitive to the hyperparameters settings (such as p and α in Eq.8), which may result in the risk of overfitting to the retain classes.
- Since the time cost is only a part of the computational overhead, the analysis of computational resource overhead is incomplete. And the automatic parameter selection in self-adaptive version also may lead to the additional time cost and computational complexity.

**Questions:**

* Could the authors do more comparative experiment with new methods, such as [1],[2] and etc.
* The authors should detailly check the symbols in the article and conduct standardized descriptions and definitions.
* The authors should analyze and explain the generalization gap between the RA and TA, where the increase of RA and decrease of TA reflect the overfitting and memory to the retain data rather than learning the generalization features.
* Since the efficiency is the core advantage of the proposed method, the authors should compare the memory cost during the unlearning process (such as peak GPU memory usage and etc.), which may help other researchers make decision with limited resources.
* Could the authors do another ablation experiment to the distance computation and automatic parameter selection in Algorithm 1, which helps the observation of whether the improvement of self-adaptive UGradSL comes at the cost of disproportionate calculation time.

Reference:
[1] Certified Unlearning for Neural Networks
[2] Towards Efficient Machine Unlearning with Data Augmentation: Guided Loss-Increasing (GLI) to Prevent the Catastrophic Model Utility Drop

---

> ### Author Response · Authors · 2025-11-26
> **Response from the author (1)**
>
> **W1, Q1** We appreciate the reviewer's suggestion for us to add the most recent two baselines (GLI [1] and PABI [2]). We keep the data splitting exactly the same as our method and only replace the unlearning method. We tested these two baseline methods in all forgetting scenarios such as class-wise forgetting, random forgetting and group-wise on multiple datasets including CIFAR-100, CIFAR-20, CelebA, TinyImageNet and ImageNet. For the space limit, we only report these two methods and the our proposed methods. For the complete results, we have added them to Table 1-3 in the main text in red.
>
> **CIFAR-100 (Class-wise Forgetting)**
>
> | **Method** | **UA**             | **$\mbox{MIA}_{\mbox{Score}}$** | **RA**            | **TA**            | **Avg. Gap (↓)** | **RTE (↓, min)** |
> | ---------- | ------------------ | ------------------------------- | ----------------- | ----------------- | ---------------- | ---------------- |
> | Retrain    | $100.00_{\pm0.00}$ | $100.00_{\pm0.00}$              | $99.96_{\pm0.01}$ | $71.10_{\pm0.12}$ | -                | 26.95            |
> | GLI        | $39.78_{\pm6.74}$  | $69.63_{\pm7.44}$               | $95.57_{\pm2.11}$ | $69.63_{\pm0.60}$ | 24.11            | 1.03             |
> | PABI       | $100_{\pm0.00}$    | $100.00_{\pm0.00}$              | $98.94_{\pm0.16}$ | $73.41_{\pm0.09}$ | 0.83             | 20.09            |
> | UGradSL    | $66.59_{\pm0.90}$  | $90.96_{\pm5.05}$               | $95.45_{\pm1.42}$ | $70.34_{\pm1.78}$ | 12.87            | 0.07             |
> | UGradSL+   | $100.00_{\pm0.00}$ | $100.00_{\pm0.00}$              | $98.44_{\pm0.62}$ | $74.12_{\pm0.70}$ | **0.57**         | 3.37             |
>
> **ImageNet (Class-wise Forgetting)**
>
> | **Method** | **UA**             | **$\mbox{MIA}_{\mbox{Score}}$** | **RA**            | **TA**            | **Avg. Gap (↓)** | **RTE (↓, hr)** |
> | ---------- | ------------------ | ------------------------------- | ----------------- | ----------------- | ---------------- | --------------- |
> | Retrain    | $100.00_{\pm0.00}$ | $100.00_{\pm0.00}$              | $71.62_{\pm0.12}$ | $69.57_{\pm0.07}$ | -                | 26.18           |
> | GLI        | $53.38_{\pm2.96}$  | $73.31_{\pm3.22}$               | $73.01_{\pm0.11}$ | $63.23_{\pm0.06}$ | 20.26            | 3.79            |
> | PABI       | -                  | -                               | -                 | -                 | -                | -               |
> | UGradSL    | $100.00_{\pm0.00}$ | $100.00_{\pm0.00}$              | $76.91_{\pm1.82}$ | $65.94_{\pm1.35}$ | **2.23**         | 0.01            |
> | UGradSL+   | $100.00_{\pm0.00}$ | $100.00_{\pm0.00}$              | $78.16_{\pm0.07}$ | $66.84_{\pm0.06}$ | 2.32             | 4.19            |
>
> **CIFAR-100 (Random Forgetting)**
> | **Method** | **UA**            | **$\mbox{MIA}_{\mbox{Score}}$** | **RA**            | **TA**            | **Avg. Gap (↓)** | **RTE (↓, min)** |
> | ---------- | ----------------- | ------------------------------- | ----------------- | ----------------- | ---------------- | ---------------- |
> | Retrain    | $29.47_{\pm1.59}$ | $53.50_{\pm1.19}$               | $99.98_{\pm0.01}$ | $70.51_{\pm1.17}$ | -                | 25.01            |
> | GLI        | $2.88_{\pm1.51}$  | $9.33_{\pm2.36}$                | $97.16_{\pm1.48}$ | $72.04_{\pm0.30}$ | 18.78            | 0.63             |
> | PABI       | $28.33_{\pm0.74}$ | $39.31_{\pm0.88}$               | $99.14_{\pm0.01}$ | $72.00_{\pm0.20}$ | 4.42             | 19.10            |
> | UGradSL    | $18.36_{\pm0.17}$ | $40.71_{\pm0.13}$               | $98.38_{\pm0.03}$ | $68.23_{\pm0.16}$ | 6.95             | 0.55             |
> | UGradSL+   | $21.69_{\pm0.59}$ | $49.47_{\pm1.25}$               | $99.87_{\pm0.34}$ | $73.60_{\pm0.26}$ | **3.75**         | 3.52             |
>
> **Tiny-ImageNet (Random Forgetting)**
> | **Method** | **UA**            | **$\mbox{MIA}_{\mbox{Score}}$** | **RA**            | **TA**            | **Avg. Gap (↓)** | **RTE (↓, min)** |
> | ---------- | ----------------- | ------------------------------- | ----------------- | ----------------- | ---------------- | ---------------- |
> | Retrain    | $49.35_{\pm0.38}$ | $58.44_{\pm0.89}$               | $83.80_{\pm0.29}$ | $59.66_{\pm0.44}$ | -                | 235.68           |
> | GLI        | $38.40_{\pm1.74}$ | $67.87_{\pm2.20}$               | $98.37_{\pm0.27}$ | $61.53_{\pm0.30}$ | 9.21             | 22.92            |
> | PABI       | $99.90_{\pm0.03}$ | $66.46_{\pm56.83}$              | $0.50_{\pm0.01}$  | $0.00_{\pm0.00}$  | 50.38            | 54.58            |
> | UGradSL    | $40.73_{\pm0.71}$ | $37.58_{\pm0.21}$               | $67.30_{\pm0.04}$ | $50.38_{\pm0.77}$ | 13.82            | 9.47             |
> | UGradSL+   | $53.06_{\pm1.27}$ | $59.46_{\pm1.01}$               | $81.38_{\pm0.75}$ | $52.52_{\pm0.84}$ | **3.57**         | 25.93            |

---

> ### Author Response · Authors · 2025-11-26
> **Response from the author (2)**
>
> **CIFAR-20 (Group Forgetting)**
>
> | **Method** | **UA**            | **$\mbox{MIA}_{\mbox{Score}}$** | **RA**            | **TA**            | **Avg. Gap (↓)** | **RTE (↓, min)** |
> | ---------- | ----------------- | ------------------------------- | ----------------- | ----------------- | ---------------- | ---------------- |
> | Retrain    | $13.33_{\pm1.64}$ | $28.47_{\pm0.75}$               | $99.94_{\pm0.01}$ | $81.23_{\pm0.13}$ | -                | 27.35            |
> | GLI        | $22.22_{\pm2.47}$ | $34.26_{\pm4.88}$               | $91.12_{\pm0.66}$ | $75.13_{\pm0.73}$ | 7.40             | 1.01             |
> | PABI       | $82.96_{\pm9.83}$ | $90.83_{\pm3.98}$               | $99.29_{\pm0.47}$ | $81.47_{\pm0.02}$ | 33.22            | 19.96            |
> | UGradSL    | $22.87_{\pm0.90}$ | $38.93_{\pm1.57}$               | $97.20_{\pm0.19}$ | $75.84_{\pm0.16}$ | **7.03**         | 0.13             |
> | UGradSL+   | $78.44_{\pm1.19}$ | $88.67_{\pm0.35}$               | $97.93_{\pm0.71}$ | $79.77_{\pm0.58}$ | 32.20            | 8.12             |
>
> **CelebA (Group Forgetting)**
> | **Method** | **UA**           | **$\mbox{MIA}_{\mbox{Score}}$** | **RA**            | **TA**            | **Avg. Gap (↓)** | **RTE (↓, min)** |
> | ---------- | ---------------- | ------------------------------- | ----------------- | ----------------- | ---------------- | ---------------- |
> | Retrain    | $6.74_{\pm0.26}$ | $9.77_{\pm1.49}$                | $94.38_{\pm0.49}$ | $91.78_{\pm0.33}$ | -                | 258.69           |
> | GLI        | $6.43_{\pm0.43}$ | $5.70_{\pm0.74}$                | $92.59_{\pm0.08}$ | $92.36_{\pm0.11}$ | 1.69             | 0.52             |
> | PABI       | $3.71_{\pm0.19}$ | $45.16_{\pm1.97}$               | $99.83_{\pm0.01}$ | $99.57_{\pm0.00}$ | 12.92            | 38.87            |
> | UGradSL    | $6.29_{\pm1.41}$ | $5.73_{\pm3.50}$                | $93.44_{\pm0.14}$ | $92.80_{\pm0.27}$ | **1.61**         | 2.17             |
> | UGradSL+   | $6.12_{\pm0.31}$ | $5.54_{\pm0.34}$                | $92.79_{\pm0.01}$ | $92.49_{\pm0.04}$ | 1.79             | 51.41            |
>
> For GLI and PABI, we use the hyper-parameter as the same in the original paper. Our method outperforms these two methods. For PABI, we follow the original JAX implementation. We try our best to apply TinyImageNet and ImageNet to the implementation. However, the results from TinyImageNet is suboptimal even we try several hyper-parameter combination given in the config files in the repo. The ImageNet is too slow. The pre-trained ImageNet weight is from Jax-ResNet [3]. We are continuing to work on obtaining stable ImageNet results and will fill in those numbers as they become available.
>
> [1] Koloskova, Anastasia, et al. "Certified Unlearning for Neural Networks." arXiv preprint arXiv:2506.06985 (2025).
>
> [2] Choi, Dasol, et al. "Towards efficient machine unlearning with data augmentation: Guided loss-increasing (gli) to prevent the catastrophic model utility drop." Proceedings of the IEEE/CVF Conference on Computer Vision and Pattern Recognition. 2024.
>
> [3] https://github.com/n2cholas/jax-resnet
>
> **W2, Q2** We have check all the notations and add them to the notation table in Appendix A. For example, we clarify the distance is cosine distance by default. We update the feature distance calculation as  $d(h_{\theta}(x_i^r), h_{\theta}(x_j^f))$ to make it more clear. The modification is noted in red in the paper. We appreciate the reviewer's careful reading and welcome any further questions.
>
> **W3** We appreciate the reviewer's valuable suggestion. All the ablation study about the GA ratio $p$ and smoothing rate $\alpha$ are given in the General response ([link1](https://openreview.net/forum?id=X74KnsoYEM&noteId=sKv4HiRvat), [link2](https://openreview.net/forum?id=X74KnsoYEM&noteId=37OiTc3Msg), [link3](https://openreview.net/forum?id=X74KnsoYEM&noteId=Jh2g3frBxO)) and added to the main text (Section 5.3) and Appendix E.8 and E.9. **The conclusion is that our methods are relatively robust to these two parameters.**
>
> **W4, Q5** For the distance calculation, we provide a detailed discussion about the FLOP and memory analysis in the general response [above](https://openreview.net/forum?id=X74KnsoYEM&noteId=nUintKWbqY). **The extra distance calculation leads to very little addition to the time and GPU memory.**

---

> ### Author Response · Authors · 2025-11-26
> **Response from the author (3)**
>
> **Q3** We use the difference between TA and RA under retraining as the baseline. For each experiment, we then measure how far the TA–RA difference of our methods deviates from this baseline and compute the average deviation across all settings. We find that this deviation is around 0.3, indicating that our methods behave similarly to retraining. This suggests that our approach does not achieve unlearning by overfitting the retain set, but rather by closely matching the behavior of retraining. The summary is reported in the following table.
>
> **UGradSL**
>
> |      **Setting**      |  **Dataset**  | **RA**  | **TA**  | **RA−TA** | **Baseline (Retrain RA−TA)** | **Distance** |
> | :-------------------: | :-----------: | :-----: | :-----: | :-------: | :--------------------------: | :----------: |
> |      Class-wise       |   CIFAR-100   | $95.45$ | $70.34$ |  $25.11$  |           $28.86$            |   $−3.75$    |
> |      Class-wise       |   ImageNet    | $76.91$ | $65.94$ |  $10.97$  |            $2.05$            |   $+8.92$    |
> |        Random         |   CIFAR-100   | $98.38$ | $68.23$ |  $30.15$  |           $29.47$            |   $+0.68$    |
> |        Random         | Tiny-ImageNet | $67.30$ | $50.38$ |  $16.92$  |           $24.14$            |   $−7.22$    |
> |         Group         |   CIFAR-20    | $97.20$ | $75.84$ |  $21.36$  |           $18.71$            |   $+2.65$    |
> |         Group         |    CelebA     | $93.44$ | $92.80$ |  $0.64$   |            $2.60$            |   $−1.96$    |
> | Class-wise (add. MIA) |   CIFAR-10    | $95.47$ | $86.78$ |  $8.69$   |            $3.69$            |   $+5.00$    |
> |   Random (add. MIA)   |   CIFAR-10    | $98.82$ | $92.17$ |  $6.65$   |            $8.39$            |   $−1.74$    |
> |   **Mean distance**   |      $—$      |   $—$   |   $—$   |    $—$    |             $—$              |  $+0.3225$   |
>
> **UGradSL+**
>
> |      **Setting**      |  **Dataset**  | **RA**  | **TA**  | **RA−TA** | **Baseline (Retrain RA−TA)** | **Distance** |
> | :-------------------: | :-----------: | :-----: | :-----: | :-------: | :--------------------------: | :----------: |
> |      Class-wise       |   CIFAR-100   | $98.44$ | $74.12$ |  $24.32$  |           $28.86$            |   $−4.54$    |
> |      Class-wise       |   ImageNet    | $78.16$ | $66.84$ |  $11.32$  |            $2.05$            |   $+9.27$    |
> |        Random         |   CIFAR-100   | $99.87$ | $73.60$ |  $26.27$  |           $29.47$            |   $−3.20$    |
> |        Random         | Tiny-ImageNet | $81.38$ | $52.52$ |  $28.86$  |           $24.14$            |   $+4.72$    |
> |         Group         |   CIFAR-20    | $97.93$ | $79.77$ |  $18.16$  |           $18.71$            |   $−0.55$    |
> |         Group         |    CelebA     | $92.79$ | $92.49$ |  $0.30$   |            $2.60$            |   $−2.30$    |
> | Class-wise (add. MIA) |   CIFAR-10    | $99.26$ | $94.29$ |  $4.97$   |            $3.69$            |   $+1.28$    |
> |   Random (add. MIA)   |   CIFAR-10    | $99.79$ | $93.64$ |  $6.15$   |            $8.39$            |   $−2.24$    |
> |   **Mean distance**   |      $—$      |   $—$   |   $—$   |    $—$    |             $—$              |   $+0.305$   |
>
> **Q4** We report the peak memory usage for different datasets and different models as follows.
> | Dataset              | Model     | Batch Size | Peak Memory (GB) |
> | -------------------- | --------- | ---------- | ---------------- |
> | CIFAR-10 / CIFAR-100 | ResNet-18 | 128        | 2.03             |
> |                      | VGG-16    | 128        | 1.67             |
> | TinyImageNet         | ResNet-18 | 128        | 12.56            |
> | CelebA               | ResNet-18 | 128        | 22.21            |
> | ImageNet             | ResNet-18 | 256        | 16.11            |
>
> **Q5** To better investigate the self-adaptive smoothing rate, we conduct an ablation study about the threshold $\beta$ and the distance selection. We select random forgetting (10%) on CIFAR-10. The results are as follows.

---

> ### Author Response · Authors · 2025-11-26
> **Response from the author (4)**
>
> | **$\beta$** | **Distance** | **UA**             | **MIA**            | **RA**             | **TA**             | **Avg. Gap (↓)** |
> | ----------- | ------------ | ------------------ | ------------------ | ------------------ | ------------------ | ---------------- |
> | -           | -            | $8.07$             | $17.41$            | $100.00$           | $91.61$            | -                |
> | Median      | Cosine       | $7.54_{\pm 0.43}$  | $13.57_{\pm 0.12}$ | $99.67_{\pm 0.00}$ | $92.97_{\pm 0.17}$ | $1.52$           |
> |             | Euclidean    | $11.21_{\pm 0.21}$ | $21.02_{\pm 2.23}$ | $94.35_{\pm 0.22}$ | $88.58_{\pm 0.26}$ | $3.86$           |
> | 0.1         | Cosine       | $7.79_{\pm 0.52}$  | $17.04_{\pm 0.61}$ | $95.84_{\pm 0.27}$ | $90.10_{\pm 0.47}$ | $1.67$           |
> |             | Euclidean    | $7.30_{\pm 0.62}$  | $16.42_{\pm 0.66}$ | $96.16_{\pm 0.94}$ | $90.46_{\pm 0.91}$ | $1.69$           |
> | 0.2         | Cosine       | $8.38_{\pm 0.19}$  | $17.46_{\pm 1.09}$ | $95.38_{\pm 0.34}$ | $89.56_{\pm 0.53}$ | $1.94$           |
> |             | Euclidean    | $7.80_{\pm 0.76}$  | $16.55_{\pm 1.91}$ | $95.75_{\pm 1.04}$ | $89.80_{\pm 0.50}$ | $1.93$           |
> | 0.3         | Cosine       | $8.27_{\pm 0.65}$  | $18.19_{\pm 0.29}$ | $95.94_{\pm 0.84}$ | $90.18_{\pm 0.62}$ | $1.71$           |
> |             | Euclidean    | $7.68_{\pm 0.65}$  | $17.28_{\pm 0.52}$ | $95.85_{\pm 0.75}$ | $90.25_{\pm 0.55}$ | $1.62$           |
> | 0.4         | Cosine       | $8.49_{\pm 0.28}$  | $17.92_{\pm 0.52}$ | $95.85_{\pm 0.20}$ | $90.09_{\pm 0.03}$ | $1.66$           |
> |             | Euclidean    | $8.38_{\pm 0.60}$  | $17.86_{\pm 0.89}$ | $95.60_{\pm 0.78}$ | $90.06_{\pm 0.57}$ | $1.80$           |
> | 0.5         | Cosine       | $9.23_{\pm 0.89}$  | $16.81_{\pm 1.66}$ | $95.46_{\pm 0.62}$ | $89.79_{\pm 0.86}$ | $2.15$           |
> |             | Euclidean    | $8.98_{\pm 0.69}$  | $16.77_{\pm 1.62}$ | $95.39_{\pm 1.01}$ | $89.34_{\pm 1.17}$ | $2.31$           |
> | 0.6         | Cosine       | $9.95_{\pm 0.64}$  | $19.90_{\pm 0.95}$ | $95.47_{\pm 0.12}$ | $89.82_{\pm 0.30}$ | $2.67$           |
> |             | Euclidean    | $10.00_{\pm 0.10}$ | $19.00_{\pm 1.92}$ | $95.15_{\pm 0.26}$ | $89.53_{\pm 0.28}$ | $2.66$           |
> | 0.7         | Cosine       | $11.81_{\pm 0.74}$ | $20.67_{\pm 2.62}$ | $94.25_{\pm 0.76}$ | $88.78_{\pm 1.02}$ | $3.90$           |
> |             | Euclidean    | $11.25_{\pm 0.59}$ | $21.54_{\pm 1.12}$ | $94.69_{\pm 0.71}$ | $89.05_{\pm 0.71}$ | $3.79$           |
> | 0.8         | Cosine       | $13.06_{\pm 0.53}$ | $18.81_{\pm 0.81}$ | $92.89_{\pm 0.69}$ | $87.29_{\pm 0.75}$ | $4.45$           |
> |             | Euclidean    | $12.07_{\pm 0.45}$ | $19.23_{\pm 2.00}$ | $93.81_{\pm 0.95}$ | $88.34_{\pm 1.00}$ | $3.82$           |
> | 0.9         | Cosine       | $11.75_{\pm 0.09}$ | $21.02_{\pm 1.43}$ | $94.34_{\pm 0.38}$ | $88.81_{\pm 0.31}$ | $3.94$           |
> |             | Euclidean    | $12.01_{\pm 1.12}$ | $21.49_{\pm 1.17}$ | $94.26_{\pm 1.08}$ | $88.74_{\pm 0.88}$ | $4.16$           |
> | 1.0         | Cosine       | $11.48_{\pm 0.06}$ | $20.59_{\pm 2.63}$ | $94.19_{\pm 0.56}$ | $88.82_{\pm 0.38}$ | $3.80$           |
> |             | Euclidean    | $11.79_{\pm 0.37}$ | $17.35_{\pm 0.85}$ | $94.37_{\pm 0.34}$ | $88.67_{\pm 0.56}$ | $3.23$           |

---

> ### Author Response · Authors · 2025-11-26
> **Response from the author (5)**
>
> | $\beta$ | Distance  | UA                 | MIA                | RA                 | TA                 | Avg. Gap ($\downarrow$) |
> | ------- | --------- | ------------------ | ------------------ | ------------------ | ------------------ | ----------------------- |
> | -       | -         | $8.07$             | $17.41$            | $100.00$           | $91.61$            | -                       |
> | Median  | Cosine    | $6.04_{\pm 0.11}$  | $13.75_{\pm 0.32}$ | $99.11_{\pm 0.01}$ | $92.07_{\pm 0.02}$ | $1.76$                  |
> |         | Euclidean | $8.59_{\pm 1.85}$  | $17.30_{\pm 0.98}$ | $94.39_{\pm 1.15}$ | $88.97_{\pm 1.12}$ | $2.59$                  |
> | 0.1     | Cosine    | $6.50_{\pm 0.14}$  | $14.76_{\pm 1.52}$ | $95.64_{\pm 0.23}$ | $89.91_{\pm 0.17}$ | $2.57$                  |
> |         | Euclidean | $6.68_{\pm 0.88}$  | $14.69_{\pm 1.66}$ | $95.34_{\pm 0.79}$ | $89.90_{\pm 0.69}$ | $2.62$                  |
> | 0.2     | Cosine    | $7.01_{\pm 0.67}$  | $15.86_{\pm 0.86}$ | $95.18_{\pm 0.44}$ | $89.69_{\pm 0.19}$ | $2.34$                  |
> |         | Euclidean | $6.82_{\pm 0.44}$  | $15.81_{\pm 0.70}$ | $95.58_{\pm 0.73}$ | $90.02_{\pm 0.57}$ | $2.21$                  |
> | 0.3     | Cosine    | $7.01_{\pm 0.98}$  | $15.13_{\pm 1.26}$ | $95.24_{\pm 0.99}$ | $89.76_{\pm 0.41}$ | $2.49$                  |
> |         | Euclidean | $7.32_{\pm 1.06}$  | $16.45_{\pm 2.08}$ | $94.68_{\pm 0.89}$ | $89.16_{\pm 0.33}$ | $2.53$                  |
> | 0.4     | Cosine    | $7.91_{\pm 0.26}$  | $15.69_{\pm 1.11}$ | $94.69_{\pm 0.51}$ | $89.07_{\pm 0.29}$ | $2.45$                  |
> |         | Euclidean | $6.24_{\pm 0.21}$  | $14.16_{\pm 0.12}$ | $95.75_{\pm 0.40}$ | $90.13_{\pm 0.13}$ | $2.70$                  |
> | 0.5     | Cosine    | $7.61_{\pm 0.66}$  | $16.50_{\pm 1.68}$ | $95.03_{\pm 0.36}$ | $89.69_{\pm 0.72}$ | $2.26$                  |
> |         | Euclidean | $8.27_{\pm 1.33}$  | $16.44_{\pm 1.83}$ | $94.67_{\pm 1.33}$ | $89.03_{\pm 1.28}$ | $2.68$                  |
> | 0.6     | Cosine    | $8.76_{\pm 0.28}$  | $16.53_{\pm 1.88}$ | $94.31_{\pm 0.61}$ | $88.54_{\pm 0.50}$ | $2.75$                  |
> |         | Euclidean | $8.67_{\pm 0.28}$  | $17.01_{\pm 2.43}$ | $94.34_{\pm 0.16}$ | $88.93_{\pm 0.30}$ | $2.66$                  |
> | 0.7     | Cosine    | $9.88_{\pm 1.05}$  | $18.33_{\pm 2.82}$ | $93.55_{\pm 0.92}$ | $88.08_{\pm 0.42}$ | $3.44$                  |
> |         | Euclidean | $9.61_{\pm 0.86}$  | $17.93_{\pm 2.33}$ | $94.11_{\pm 0.49}$ | $88.69_{\pm 0.19}$ | $2.99$                  |
> | 0.8     | Cosine    | $9.61_{\pm 1.12}$  | $16.91_{\pm 1.51}$ | $93.68_{\pm 1.20}$ | $88.48_{\pm 0.76}$ | $3.08$                  |
> |         | Euclidean | $9.75_{\pm 0.17}$  | $16.79_{\pm 0.52}$ | $93.87_{\pm 0.02}$ | $88.34_{\pm 0.39}$ | $2.93$                  |
> | 0.9     | Cosine    | $9.19_{\pm 0.66}$  | $17.84_{\pm 0.72}$ | $94.19_{\pm 0.50}$ | $88.51_{\pm 0.84}$ | $2.63$                  |
> |         | Euclidean | $9.76_{\pm 0.49}$  | $18.61_{\pm 0.65}$ | $93.90_{\pm 0.39}$ | $88.47_{\pm 0.35}$ | $3.03$                  |
> | 1.0     | Cosine    | $9.39_{\pm 0.07}$  | $16.94_{\pm 0.26}$ | $94.26_{\pm 0.33}$ | $88.74_{\pm 0.22}$ | $2.60$                  |
> |         | Euclidean | $10.41_{\pm 0.24}$ | $19.16_{\pm 1.08}$ | $93.50_{\pm 0.63}$ | $88.21_{\pm 0.34}$ | $3.50$                  |

---

### Official Review · Reviewer_be9Q · 2025-11-01

**Soundness:** 3
**Presentation:** 2
**Contribution:** 3
**Rating:** 6
**Confidence:** 2

**Summary:**

This paper proposes UGradSL, which is a gradient-based unlearning methods that integrate label smoothing. The approach performs gradient ascent on forget data and gradient descent on retain data, using smoothed labels to balance forgetting and retaining.
Theoretical analyses explain when NLS improves gradient ascent and link the method to label-local differential privacy (Label-LDP) guarantees. Experiments on several datasets demonstrate the effectiveness of the proposed.

**Strengths:**

1. The paper has clear problem framing and the motivation for introducing label smoothing to stablilize gradient ascent is sound.
2. The theoretical analysis part is clear and provides useful insights.
3. The proposed UGradSL is simple yet effective.
4. The experiments cover class-level, random, and group unlearning and sufficient datasets and baselines.

**Weaknesses:**

1. The forget-set size: The forget-set size appears fixed for each experiment. Varying forget-set sizes is important for assessing the effectiveness of the method. This would reveal whether the proposed method scales well when more data need to be forgotten.
2. Lack of ablation study/sensitivity analysis: The paper lacks a discussion on the contribution of each term in the mixed gradient objective. For example, results for different p should be provided, since it is an important factor used to balance GD and GA.
3. Smoothing rate sensitivity: the effect of the label smoothing rate should be analyzed, and a discussion on how much the observed benefits arise from smoothing is worth adding in.

**Questions:**

1. About forget-set size: have the authors evaluated the method under different forget-set sizes or compositions? Since forget-set size can significantly affect both unlearning and retention, understanding how the method scales with it would help assess its general applicability.
2. In Eq (8), p controls the balance between gradient ascent (forgetting) and gradient descent (retaining). Could the authors provide experimental results for different p values? This would clarify how sensitive the method is to this balance.
3. How was the smoothing rate chosen? It would be valuable to include sensitivity results or justification for the chosen range.
4. How the density is computed in UGradSL+ and how it interacts with the theoretical framework?
5. The paper suggests that UGradSL can be easily integrated into other unlearning frameworks. Have the authors tested this claim empirically?

---

> ### Author Response · Authors · 2025-11-26
> **Response from the author (1)**
>
> Thank you for the thoughtful review and for highlighting both the strengths of our work and areas where we could be clearer.
>
> **W1 - W3, Q1 - Q3**. The ablation study of the GA ratio $p$ and smoothing ratio $\alpha$ have been given in the general response above ([link1](https://openreview.net/forum?id=X74KnsoYEM&noteId=sKv4HiRvat), [link2](https://openreview.net/forum?id=X74KnsoYEM&noteId=37OiTc3Msg), [link3](https://openreview.net/forum?id=X74KnsoYEM&noteId=Jh2g3frBxO)). **Although there is some performance fluctuation when $p$ and $\alpha$ change, our methods are relatively stable with $p$ and $\alpha$**.
>
> For the forgetting set size, we change the forgetting set size from 10% to 50% training set on both CIFAR-10 and CIFAR-100 for robustness checking. Our method still outperforms the baseline methods and shows great robustness to the parameters, meaning that our methods can always provide a relatively good baseline for diverse forgetting scenarios and datasets in diverse scales.
>
> **CIFAR-10 Random Set Size (10%)**
>
> | **Method**      | **UA**            | **$\mbox{MIA}_{\mbox{Score}}$** | **RA**             | **TA**            | **Avg. Gap (↓)** | **RTE (↓, min)** |
> | --------------- | ----------------- | ------------------------------- | ------------------ | ----------------- | ---------------- | ---------------- |
> | Retrain         | 8.07              | 17.41                           | 100.00             | 91.61             | -                | 24.66            |
> | FT              | $1.10_{\pm0.19}$  | $4.06_{\pm0.40}$                | $98.83_{\pm0.03}$  | $93.70_{\pm0.10}$ | 5.90             | 1.58             |
> | RL              | $6.39_{\pm1.09}$  | $0.00_{\pm0.00}$                | $99.50_{\pm0.10}$  | $99.04_{\pm0.08}$ | 6.76             | 1.92             |
> | GA              | $0.56_{\pm0.01}$  | $1.19_{\pm0.05}$                | $99.48_{\pm0.02}$  | $94.55_{\pm0.05}$ | 6.80             | 0.31             |
> | IU              | $17.51_{\pm2.19}$ | $21.39_{\pm1.70}$               | $98.00_{\pm0.38}$  | $98.11_{\pm0.38}$ | 5.48             | 1.18             |
> | BE              | $0.00_{\pm0.00}$  | $0.26_{\pm0.02}$                | $100.00_{\pm0.00}$ | $95.35_{\pm0.18}$ | 7.24             | 1.37             |
> | BS              | $0.37_{\pm0.10}$  | $1.10_{\pm0.43}$                | $99.93_{\pm0.01}$  | $98.97_{\pm0.02}$ | 7.86             | 1.21             |
> | $\ell_1$-sparse | $2.80_{\pm0.37}$  | $19.59_{\pm3.48}$               | $99.07_{\pm0.04}$  | $98.00_{\pm0.12}$ | 3.69             | 1.98             |
> | SalUn           | $46.95_{\pm0.15}$ | $86.33_{\pm2.58}$               | $97.75_{\pm0.42}$  | $97.22_{\pm0.77}$ | 28.92            | 2.42             |
> | UGradSL         | $5.87_{\pm0.50}$  | $13.33_{\pm0.20}$               | $98.82_{\pm0.28}$  | $92.17_{\pm0.20}$ | **2.01**         | 0.45             |
> | UGradSL+        | $6.03_{\pm0.17}$  | $10.65_{\pm0.13}$               | $99.79_{\pm0.03}$  | $93.64_{\pm0.16}$ | 2.76             | 3.07             |

---

> ### Author Response · Authors · 2025-11-26
> **Response from the author (2)**
>
> **CIFAR-10 Random Set Size (20%)**
>
> | **Method**      | **UA**           | **$\mbox{MIA}_{\mbox{Score}}$** | **RA**            | **TA**            | **Avg. Gap (↓)** | **RTE (↓, min)** |
> | --------------- | ---------------- | ------------------------------- | ----------------- | ----------------- | ---------------- | ---------------- |
> | Retrain         | 5.31             | 13.30                           | 100.00            | 94.10             | -                | 38.74            |
> | FT              | $0.76_{\pm4.55}$ | $2.69_{\pm10.61}$               | $99.89_{\pm0.11}$ | $93.97_{\pm0.13}$ | 3.85             | 2.17             |
> | RL              | $6.47_{\pm1.16}$ | $28.62_{\pm15.32}$              | $99.60_{\pm0.40}$ | $92.39_{\pm1.71}$ | 4.65             | 2.65             |
> | GA              | $0.67_{\pm4.64}$ | $1.44_{\pm11.86}$               | $99.48_{\pm0.52}$ | $94.42_{\pm0.32}$ | 4.33             | 0.26             |
> | IU              | $2.91_{\pm2.40}$ | $5.53_{\pm7.77}$                | $97.30_{\pm2.70}$ | $90.64_{\pm3.46}$ | 4.08             | 3.29             |
> | BE              | $0.57_{\pm4.74}$ | $1.64_{\pm11.66}$               | $99.44_{\pm0.56}$ | $94.32_{\pm0.22}$ | 4.29             | 0.53             |
> | BS              | $0.62_{\pm4.69}$ | $1.62_{\pm11.68}$               | $99.46_{\pm0.54}$ | $94.20_{\pm0.10}$ | 4.25             | 0.86             |
> | $\ell_1$-sparse | $3.92_{\pm1.39}$ | $8.94_{\pm4.36}$                | $98.09_{\pm1.91}$ | $91.92_{\pm2.18}$ | 2.46             | 2.20             |
> | SalUn           | $3.73_{\pm1.58}$ | $13.18_{\pm0.12}$               | $98.61_{\pm1.39}$ | $92.75_{\pm1.35}$ | 1.11             | 2.66             |
> | UGradSL         | $6.07_{\pm0.70}$ | $13.82_{\pm1.03}$               | $95.71_{\pm0.17}$ | $90.19_{\pm0.23}$ | 2.37             | 0.24             |
> | UGradSL+        | $6.39_{\pm0.19}$ | $12.34_{\pm1.79}$               | $97.08_{\pm0.44}$ | $90.91_{\pm0.95}$ | **2.04**         | 0.31             |
>
> **CIFAR-10 Random Set Size (30%)**
>
> | **Method**        | **UA**           | **$\mbox{MIA}_{\mbox{Score}}$** | **RA**            | **TA**            | **Avg. Gap (↓)** | **RTE (↓, min)** |
> | ----------------- | ---------------- | ------------------------------- | ----------------- | ----------------- | ---------------- | ---------------- |
> | Retrain           | 6.64             | 14.60                           | 100.00            | 92.78             | -                | 33.65            |
> | FT                | $0.56_{\pm6.08}$ | $1.66_{\pm12.94}$               | $99.83_{\pm0.17}$ | $94.22_{\pm1.44}$ | 5.16             | 1.98             |
> | RL                | $6.89_{\pm0.25}$ | $31.09_{\pm16.49}$              | $99.36_{\pm0.64}$ | $91.35_{\pm1.43}$ | 4.70             | 2.63             |
> | GA                | $0.65_{\pm5.99}$ | $1.50_{\pm13.10}$               | $99.46_{\pm0.54}$ | $94.44_{\pm1.66}$ | 5.32             | 2.40             |
> | IU                | $3.95_{\pm2.69}$ | $7.26_{\pm7.34}$                | $96.22_{\pm3.78}$ | $89.61_{\pm3.17}$ | 4.24             | 3.32             |
> | BE                | $0.63_{\pm6.01}$ | $3.35_{\pm11.25}$               | $99.39_{\pm0.61}$ | $94.19_{\pm1.41}$ | 4.82             | 0.81             |
> | BS                | $0.63_{\pm6.01}$ | $2.88_{\pm11.72}$               | $99.39_{\pm0.61}$ | $94.15_{\pm1.37}$ | 4.93             | 1.28             |
> | $\ell_{1}$-sparse | $4.70_{\pm1.94}$ | $9.97_{\pm4.63}$                | $97.63_{\pm2.37}$ | $91.19_{\pm1.59}$ | 2.63             | 1.99             |
> | SalUn             | $6.22_{\pm0.42}$ | $14.11_{\pm0.49}$               | $95.91_{\pm4.09}$ | $90.72_{\pm2.06}$ | 1.76             | 2.64             |
> | UGradSL           | $6.78_{\pm0.66}$ | $15.96_{\pm0.12}$               | $96.94_{\pm0.56}$ | $90.72_{\pm0.80}$ | 1.66             | 0.70             |
> | UGradSL+          | $6.36_{\pm0.65}$ | $14.99_{\pm0.82}$               | $97.35_{\pm0.79}$ | $91.10_{\pm1.10}$ | **1.25**         | 0.53             |

---

> ### Author Response · Authors · 2025-11-26
> **Response from the author (3)**
>
> **CIFAR-10 Random Set Size (40%)**
>
> | **Method**      | **UA**           | **$\mbox{MIA}_{\mbox{Score}}$** | **RA**            | **TA**            | **Avg. Gap (↓)** | **RTE (↓, min)** |
> | --------------- | ---------------- | ------------------------------- | ----------------- | ----------------- | ---------------- | ---------------- |
> | Retrain         | 7.01             | 18.37                           | 100.00            | 92.52             | -                | 28.47            |
> | FT              | $0.77_{\pm6.24}$ | $2.88_{\pm15.49}$               | $99.96_{\pm0.04}$ | $94.27_{\pm1.75}$ | 5.88             | 1.62             |
> | RL              | $5.02_{\pm1.99}$ | $37.76_{\pm19.39}$              | $99.61_{\pm0.39}$ | $92.14_{\pm0.38}$ | 5.54             | 2.68             |
> | GA              | $0.67_{\pm6.34}$ | $1.57_{\pm16.80}$               | $99.47_{\pm0.53}$ | $94.38_{\pm1.86}$ | 6.38             | 0.53             |
> | IU              | $7.89_{\pm0.88}$ | $10.99_{\pm7.38}$               | $92.21_{\pm7.79}$ | $86.15_{\pm6.37}$ | 5.60             | 3.27             |
> | BE              | $0.86_{\pm6.15}$ | $15.72_{\pm2.65}$               | $99.27_{\pm0.73}$ | $93.46_{\pm0.94}$ | 2.62             | 1.04             |
> | BS              | $1.18_{\pm5.83}$ | $13.97_{\pm4.40}$               | $98.94_{\pm1.06}$ | $93.01_{\pm0.49}$ | 2.95             | 1.72             |
> | $\ell_1$-sparse | $2.84_{\pm4.17}$ | $7.09_{\pm11.28}$               | $98.75_{\pm1.25}$ | $92.20_{\pm0.32}$ | 4.26             | 1.63             |
> | SalUn           | $6.86_{\pm0.15}$ | $15.15_{\pm3.22}$               | $95.01_{\pm4.99}$ | $89.76_{\pm2.76}$ | 2.78             | 2.67             |
> | UGradSL         | $5.81_{\pm0.11}$ | $14.98_{\pm2.65}$               | $97.31_{\pm1.06}$ | $90.73_{\pm0.48}$ | **2.27**         | 0.62             |
> | UGradSL+        | $5.82_{\pm0.37}$ | $14.53_{\pm1.83}$               | $97.11_{\pm0.40}$ | $90.74_{\pm0.38}$ | 2.42             | 0.63             |
>
> **CIFAR-10 Random Set Size (50%)**
>
> | **Method**        | **UA**           | **$\mbox{MIA}_{\mbox{Score}}$** | **RA**            | **TA**            | **Avg. Gap (↓)** | **RTE (↓, min)** |
> | ----------------- | ---------------- | ------------------------------- | ----------------- | ----------------- | ---------------- | ---------------- |
> | Retrain           | 7.91             | 19.29                           | 100.00            | 91.72             | -                | 23.90            |
> | FT                | $0.44_{\pm7.47}$ | $2.15_{\pm17.14}$               | $99.96_{\pm0.04}$ | $94.23_{\pm2.51}$ | 6.79             | 1.31             |
> | RL                | $7.61_{\pm0.30}$ | $37.36_{\pm18.07}$              | $99.67_{\pm0.33}$ | $92.83_{\pm1.11}$ | 4.95             | 2.65             |
> | GA                | $0.40_{\pm7.51}$ | $1.22_{\pm18.07}$               | $99.61_{\pm0.39}$ | $94.34_{\pm2.62}$ | 7.15             | 0.66             |
> | IU                | $3.97_{\pm3.94}$ | $7.29_{\pm12.00}$               | $96.21_{\pm3.79}$ | $90.00_{\pm1.72}$ | 5.36             | 3.25             |
> | BE                | $3.08_{\pm4.83}$ | $24.87_{\pm5.58}$               | $96.84_{\pm3.16}$ | $90.41_{\pm1.31}$ | 3.72             | 1.31             |
> | BS                | $9.76_{\pm1.85}$ | $32.15_{\pm12.86}$              | $90.19_{\pm9.81}$ | $83.71_{\pm8.01}$ | 8.13             | 2.12             |
> | $\ell_{1}$-sparse | $1.44_{\pm6.47}$ | $4.76_{\pm14.53}$               | $99.52_{\pm0.48}$ | $93.13_{\pm1.41}$ | 5.72             | 1.31             |
> | SalUn             | $7.75_{\pm0.16}$ | $16.99_{\pm2.30}$               | $94.28_{\pm5.72}$ | $89.29_{\pm2.43}$ | 2.65             | 2.68             |
> | UGradSL           | $6.83_{\pm0.23}$ | $12.73_{\pm1.66}$               | $97.62_{\pm0.71}$ | $90.27_{\pm0.55}$ | 2.87             | 0.77             |
> | UGradSL+          | $6.13_{\pm1.35}$ | $16.49_{\pm2.73}$               | $97.84_{\pm0.34}$ | $90.84_{\pm0.69}$ | **1.91**         | 0.77             |

---

> ### Author Response · Authors · 2025-11-26
> **Response from the author (4)**
>
> **CIFAR-100 Random Set Size (10%)**
>
> | **Method**      | **UA**             | **$\mbox{MIA}_{\mbox{Score}}$** | **RA**             | **TA**             | **Avg. Gap (↓)** | **RTE (↓, min)** |
> | --------------- | ------------------ | ------------------------------- | ------------------ | ------------------ | ---------------- | ---------------- |
> | Retrain         | 29.47              | 53.50                           | 99.98              | 70.51              | -                | 25.01            |
> | FT              | $2.55_{\pm 0.03}$  | $10.59_{\pm 0.27}$              | $99.95_{\pm 0.01}$ | $75.95_{\pm 0.05}$ | 18.83            | 1.95             |
> | RL              | $4.06_{\pm 0.37}$  | $50.12_{\pm 3.48}$              | $99.92_{\pm 0.02}$ | $71.30_{\pm 0.36}$ | 7.41             | 1.20             |
> | GA              | $2.58_{\pm 0.06}$  | $5.95_{\pm 0.17}$               | $97.45_{\pm 0.02}$ | $76.09_{\pm 0.01}$ | 20.64            | 0.29         |
> | IU              | $15.71_{\pm 5.19}$ | $18.69_{\pm 4.12}$              | $84.65_{\pm 5.19}$ | $62.20_{\pm 4.17}$ | 18.05            | 1.20             |
> | BE              | $0.01_{\pm 0.00}$  | $1.45_{\pm 0.02}$               | $98.22_{\pm 1.26}$ | $78.26_{\pm 0.00}$ | 22.32            | 0.24             |
> | BS              | $2.20_{\pm 2.11}$  | $10.73_{\pm 9.37}$              | $98.22_{\pm 1.26}$ | $70.23_{\pm 1.67}$ | 18.02            | 0.34             |
> | $\ell_1$-sparse | $8.19_{\pm 0.38}$  | $19.11_{\pm 0.52}$              | $88.39_{\pm 0.31}$ | $80.26_{\pm 0.16}$ | 23.75            | 1.00             |
> | SalUn           | $35.23_{\pm 0.32}$ | $89.39_{\pm 0.46}$              | $99.53_{\pm 0.04}$ | $64.26_{\pm 0.58}$ | 12.10            | 3.33             |
> | UGradSL         | $18.36_{\pm 0.17}$ | $40.71_{\pm 0.13}$              | $98.38_{\pm 0.03}$ | $68.23_{\pm 0.16}$ | 6.95             | 0.55             |
> | UGradSL+        | $21.69_{\pm 0.59}$ | $49.47_{\pm 1.25}$              | $99.87_{\pm 0.34}$ | $73.60_{\pm 0.26}$ | **3.75**         | 3.52             |
>
> **CIFAR-100 Random Set Size (20%)**
>
> | **Method**      | **UA**              | **$\mbox{MIA}_{\mbox{Score}}$** | **RA**              | **TA**              | **Avg. Gap (↓)** | **RTE (↓, min)** |
> | --------------- | ------------------- | ------------------------------- | ------------------- | ------------------- | ---------------- | ---------------- |
> | Retrain         | 26.84               | 52.41                           | 99.99               | 73.88               | -                | 36.88            |
> | FT              | $2.70_{\pm 24.14}$  | $11.63_{\pm 40.78}$             | $99.95_{\pm 0.04}$  | $75.51_{\pm 1.63}$  | 16.65            | 2.05             |
> | RL              | $54.74_{\pm 27.90}$ | $97.32_{\pm 44.91}$             | $99.47_{\pm 0.52}$  | $65.59_{\pm 8.29}$  | 20.41            | 2.11             |
> | GA              | $6.79_{\pm 20.05}$  | $13.22_{\pm 39.19}$             | $94.11_{\pm 5.88}$  | $71.39_{\pm 2.49}$  | 16.90            | 0.26             |
> | IU              | $5.34_{\pm 21.50}$  | $11.79_{\pm 40.62}$             | $95.54_{\pm 4.45}$  | $70.89_{\pm 2.99}$  | 17.39            | 3.77             |
> | BE              | $2.51_{\pm 24.33}$  | $6.70_{\pm 45.71}$              | $97.38_{\pm 2.61}$  | $75.07_{\pm 1.19}$  | 18.46            | 0.49             |
> | BS              | $2.53_{\pm 24.31}$  | $6.57_{\pm 45.84}$              | $97.38_{\pm 2.61}$  | $75.05_{\pm 1.17}$  | 18.48            | 0.82             |
> | $\ell_1$-sparse | $37.83_{\pm 10.99}$ | $38.90_{\pm 13.51}$             | $76.63_{\pm 23.36}$ | $58.79_{\pm 15.09}$ | 15.74            | 2.05             |
> | SalUn           | $25.83_{\pm 1.01}$  | $64.69_{\pm 12.28}$             | $96.01_{\pm 3.98}$  | $65.87_{\pm 8.01}$  | 6.32             | 2.12             |
> | UGradSL         | $30.10_{\pm 1.03}$  | $47.39_{\pm 1.17}$              | $93.49_{\pm 0.24}$  | $64.99_{\pm 0.04}$  | **4.71**         | 0.83             |
> | UGradSL+        | $27.29_{\pm 0.99}$  | $35.92_{\pm 0.94}$              | $93.36_{\pm 0.03}$  | $66.59_{\pm 0.37}$  | 5.45             | 0.59             |

---

> ### Author Response · Authors · 2025-11-26
> **Response from the author (5)**
>
> **CIFAR-100 Random Set Size (30%)**
>
> | **Method**      | **UA**              | **$\mbox{MIA}_{\mbox{Score}}$** | **RA**              | **TA**              | **Avg. Gap (↓)** | **RTE (↓, min)** |
> | --------------- | ------------------- | ------------------------------- | ------------------- | ------------------- | ---------------- | ---------------- |
> | Retrain         | 28.52               | 52.24                           | 99.98               | 70.91               | -                | 32.92            |
> | FT              | $2.65_{\pm 25.87}$  | $11.18_{\pm 41.06}$             | $99.94_{\pm 0.04}$  | $75.17_{\pm 4.26}$  | 17.81            | 1.44             |
> | RL              | $51.46_{\pm 22.94}$ | $96.34_{\pm 44.10}$             | $99.32_{\pm 0.66}$  | $62.77_{\pm 8.14}$  | 18.96            | 2.14             |
> | GA              | $2.40_{\pm 26.12}$  | $5.70_{\pm 46.54}$              | $97.39_{\pm 2.59}$  | $75.33_{\pm 4.42}$  | 19.92            | 0.40             |
> | IU              | $5.96_{\pm 22.56}$  | $12.63_{\pm 39.61}$             | $94.59_{\pm 5.39}$  | $69.74_{\pm 1.17}$  | 17.18            | 3.76             |
> | BE              | $2.44_{\pm 26.08}$  | $6.53_{\pm 45.71}$              | $97.37_{\pm 2.61}$  | $74.77_{\pm 3.86}$  | 19.56            | 0.76             |
> | BS              | $2.49_{\pm 26.03}$  | $6.40_{\pm 45.84}$              | $97.33_{\pm 2.65}$  | $74.65_{\pm 3.74}$  | 19.56            | 1.24             |
> | $\ell_1$-sparse | $38.45_{\pm 9.93}$  | $38.52_{\pm 13.72}$             | $76.36_{\pm 23.62}$ | $58.09_{\pm 12.82}$ | 15.02            | 1.47             |
> | SalUn           | $27.34_{\pm 1.18}$  | $62.99_{\pm 10.75}$             | $94.50_{\pm 5.48}$  | $63.10_{\pm 7.81}$  | 6.31             | 2.16             |
> | UGradSL         | $30.10_{\pm 0.12}$  | $47.39_{\pm 2.08}$              | $93.49_{\pm 0.74}$  | $64.99_{\pm 1.53}$  | **4.71**         | 0.83             |
> | UGradSL+        | $24.89_{\pm 0.24}$  | $44.60_{\pm 0.94}$              | $94.90_{\pm 0.88}$  | $66.16_{\pm 0.78}$  | 5.28             | 0.79             |
>
> **CIFAR-100 Random Set Size (40%)**
>
> | **Method**      | **UA**              | **$\mbox{MIA}_{\mbox{Score}}$** | **RA**              | **TA**              | **Avg. Gap (↓)** | **RTE (↓, min)** |
> | --------------- | ------------------- | ------------------------------- | ------------------- | ------------------- | ---------------- | ---------------- |
> | Retrain         | 30.07               | 58.06                           | 99.99               | 69.87               | -                | 28.29            |
> | FT              | $2.66_{\pm 27.41}$  | $11.05_{\pm 47.01}$             | $99.95_{\pm 0.04}$  | $75.35_{\pm 5.48}$  | 19.99            | 1.51             |
> | RL              | $51.75_{\pm 21.68}$ | $95.78_{\pm 37.72}$             | $99.27_{\pm 0.72}$  | $59.41_{\pm 10.46}$ | 17.64            | 2.12             |
> | GA              | $2.46_{\pm 27.61}$  | $5.91_{\pm 52.15}$              | $97.39_{\pm 2.60}$  | $75.40_{\pm 5.53}$  | 21.97            | 0.51             |
> | IU              | $4.58_{\pm 25.49}$  | $10.32_{\pm 47.74}$             | $96.29_{\pm 3.70}$  | $70.92_{\pm 1.05}$  | 19.49            | 3.78             |
> | BE              | $2.54_{\pm 27.53}$  | $7.44_{\pm 50.62}$              | $97.35_{\pm 2.64}$  | $74.56_{\pm 4.69}$  | 21.37            | 1.00             |
> | BS              | $2.70_{\pm 27.37}$  | $7.63_{\pm 50.43}$              | $97.26_{\pm 2.73}$  | $74.10_{\pm 4.23}$  | 21.19            | 1.66             |
> | $\ell_1$-sparse | $38.49_{\pm 8.42}$  | $40.21_{\pm 17.85}$             | $78.43_{\pm 21.56}$ | $57.66_{\pm 12.21}$ | 15.01            | 1.52             |
> | SalUn           | $25.54_{\pm 4.53}$  | $60.08_{\pm 2.02}$              | $94.64_{\pm 5.35}$  | $62.52_{\pm 7.35}$  | 4.81             | 2.14             |
> | UGradSL         | $30.07_{\pm 1.58}$  | $49.23_{\pm 1.07}$              | $95.30_{\pm 0.34}$  | $64.52_{\pm 0.28}$  | **4.72**         | 1.08             |
> | UGradSL+        | $30.42_{\pm 0.77}$  | $45.94_{\pm 1.41}$              | $93.98_{\pm 0.50}$  | $63.21_{\pm 0.35}$  | 5.47             | 0.77             |

---

> ### Author Response · Authors · 2025-11-26
> **Response from the author (6)**
>
> **CIFAR-100 Random Set Size (50%)**
>
> | **Method**      | **UA**              | **$\mbox{MIA}_{\mbox{Score}}$** | **RA**              | **TA**              | **Avg. Gap (↓)** | **RTE (↓, min)** |
> | --------------- | ------------------- | ------------------------------- | ------------------- | ------------------- | ---------------- | ---------------- |
> | Retrain         | 32.69               | 61.15                           | 99.99               | 67.22               | -                | 25.01            |
> | FT              | $2.71_{\pm 29.98}$  | $10.71_{\pm 50.44}$             | $99.96_{\pm 0.03}$  | $75.11_{\pm 7.89}$  | 22.08            | 1.25             |
> | RL              | $50.52_{\pm 17.83}$ | $95.91_{\pm 34.76}$             | $99.47_{\pm 0.52}$  | $56.75_{\pm 10.47}$ | 15.90            | 2.13             |
> | GA              | $2.61_{\pm 30.08}$  | $5.92_{\pm 55.23}$              | $97.49_{\pm 2.50}$  | $75.27_{\pm 8.05}$  | 23.97            | 0.66             |
> | IU              | $12.64_{\pm 20.05}$ | $17.54_{\pm 43.61}$             | $87.96_{\pm 12.03}$ | $62.76_{\pm 4.46}$  | 20.04            | 3.80             |
> | BE              | $2.76_{\pm 29.93}$  | $8.85_{\pm 52.30}$              | $97.39_{\pm 2.60}$  | $74.05_{\pm 6.83}$  | 22.92            | 1.26             |
> | BS              | $2.99_{\pm 29.70}$  | $8.76_{\pm 52.39}$              | $97.24_{\pm 2.75}$  | $73.38_{\pm 6.16}$  | 22.75            | 2.08             |
> | $\ell_1$-sparse | $39.86_{\pm 7.17}$  | $40.43_{\pm 20.72}$             | $78.17_{\pm 21.82}$ | $55.65_{\pm 11.57}$ | 15.32            | 1.26             |
> | SalUn           | $26.17_{\pm 6.52}$  | $59.47_{\pm 1.68}$              | $94.04_{\pm 5.95}$  | $61.39_{\pm 5.83}$  | 5.00             | 2.13             |
> | UGradSL         | $33.80_{\pm 1.61}$  | $53.38_{\pm 2.31}$              | $95.29_{\pm 0.11}$  | $56.88_{\pm 0.80}$  | **4.86**         | 0.95             |
> | UGradSL+        | $32.20_{\pm 0.49}$  | $45.20_{\pm 1.44}$              | $94.47_{\pm 0.69}$  | $61.53_{\pm 0.97}$  | 4.89             | 0.75             |
>
>
> **Q4** Thanks for your question. We present the detailed explanation about how self-adaptive smoothing ratio is calculated in the response [above](https://openreview.net/forum?id=X74KnsoYEM&noteId=nUintKWbqY). In short, For each forgetting sample, we compute its **cosine distance** in the **feature space** to all retained samples in the batch, count how many of those distances are below the threshold $\beta$, and then divide this count by the batch size. This fraction $c_j^f / |B_f|$ is the density for that forgetting sample.
> **As for the connection with the theoretical framework**, our theoretical framework provides a connection between label smoothing and label-level DP. The adaptive smoothing rate provides a more fine-grained and individual perspective to every data point rather than treating them all the same, leading to a more fine-grained DP protection in individual perspective. The explanation and more discussion about the self-adaptive smoothing rate selection are added to Appendix D in red.
>
> **Q5** We integrate our method with $\ell_1$-spars for random forgetting on both CIFAR-10 and CIFAR-100 datasets. The forgetting set size is 10%. The results are as follows. Integrated with our proposed framework, the performance of $\ell_1$-sparse is improved, demonstrating the plug-and-play characteristic of our method.
>
> **CIFAR-10**
>
> | Method                     | UA                | MIA               | RA                | TA                | **Avg. Gap (↓)** |
> | -------------------------- | ----------------- | ----------------- | ----------------- | ----------------- | ---------------- |
> | Retrain                    | 8.07              | 17.41             | 100.00            | 91.61             | -                |
> | $\ell_1$-sparse            | $2.80_{\pm0.37}$  | $19.59_{\pm3.48}$ | $99.07_{\pm0.04}$ | $98.00_{\pm0.12}$ | 3.69             |
> | $\ell_1$-sparse + UGradSL | $11.86_{\pm0.41}$ | $19.09_{\pm2.11}$ | $95.15_{\pm0.05}$ | $88.84_{\pm0.16}$ | **3.27**         |
>
> **CIFAR-100**
>
> | Method                     | UA                | MIA               | RA                | TA                | **Avg. Gap (↓)** |
> | -------------------------- | ----------------- | ----------------- | ----------------- | ----------------- | ---------------- |
> | Retrain                    | 29.47             | 53.50             | 99.98             | 70.51             | -                |
> | $\ell_1$-sparse            | $8.19_{\pm0.38}$  | $19.11_{\pm0.52}$ | $88.39_{\pm0.31}$ | $80.26_{\pm0.16}$ | 23.75            |
> | $\ell_1$-sparse + UGradSL | $58.76_{\pm1.52}$ | $64.98_{\pm1.09}$ | $77.04_{\pm0.18}$ | $55.66_{\pm0.79}$ | **19.64**        |

---

### Author Response · Authors · 2025-11-26
**A more detailed explanation of self-adaptive smooth rate selection**

## A more detailed explanation of self-adaptive smooth rate selection

We appreciate the suggestion from the reviewers about a deeper investigation of the self-adaptive $\alpha$ selection.

**Motivation**

The density calculation steps are as follows. In the self-adaptive version of UGradSL+, the label smoothing rate for each forgetting sample is computed dynamically from its proximity to the retained data in feature space. For each iteration, the algorithm samples a batch of retained examples $B_r$ and a batch of forgetting examples $B_f$ with equal size, extracts their features $h_{\theta}(x_i^r)$ and $h_{\theta}(x_j^f)$, and computes the **feature distance** $d(h_{\theta}(x_i^r), h_{\theta}(x_j^f))$ for every retained–forgetting pair. Then, for each forgetting feature $z_j^f$, it counts how many retained features fall within a distance threshold $\beta$, denoted as $c_j^f$. This count is normalized by the batch size $|B_f|$ to obtain the adaptive smoothing rate $\alpha_j = c_j^f / |B_f|$. As a result, forgetting samples that are close to many retained samples (i.e., highly entangled in representation space) receive a higher smoothing rate and are updated more conservatively, while those that are far from retained data get a lower smoothing rate (possibly zero) and can be pushed away more aggressively during unlearning.

**Complexity Analysis**

By default, we use **cosine similarity** in the **feature space** and the threshold is the median of all the distances in the batch. The code of how we calculate the distance is as follow.
```
forget_norm = F.normalize(feature_forget, p=2, dim=1)
retain_norm = F.normalize(feature_retain, p=2, dim=1)
# cosine similarity (batch x batch)
cos_sim = forget_norm @ retain_norm.T
# convert to distance in [0,1]
cos_dist = (1 - cos_sim) / 2  # shape: [batch, batch]
# --- threshold and count ---
threshold = 0.2  # example threshold in [0,1]
# boolean matrix: True = close
close_mask = cos_dist < threshold  # [batch, batch]
density = close_mask.sum(dim=0) / len(feature_forget)
```

All computations are implemented as batched GPU tensor operations without any explicit Python loops. We assume the feature from $D_f$ and $D_r$ are both in $\mathbb{R}^{n \times d}$, where $n$ is the batch size and $d$ is the feature dimension.

For **FLOP count**,

- The two normalization operations cost approximately $6nd$ FLOPs in total, since normalizing a single $n \times d$ tensor requires about $3nd$ FLOPs (square, sum, and division).
- Computing the cosine similarity matrix costs about $2n^2d$ FLOPs, as each of the $n^2$ entries is a dot product between two $d$-dimensional vectors.
- Converting similarity to distance and applying the threshold require about $2n^2$ and $n^2$ FLOPs, respectively.
- The density computation costs about $n^2$ FLOPs for forming the mask and $n$ FLOPs for the length normalization.

Overall, the total FLOP count is $(6nd + 2n^2d + 4n^2 + n)$, which is dominated by the $O(n^2 d)$ cosine-similarity term. For our typical setting $n=64$ and $d=512$, this corresponds to roughly $4.4\times 10^6$ FLOPs. Compared with the FP32 peak throughput of an A6000 GPU (38.71 TFLOPS), this overhead is negligible relative to the usual forward/backward passes.

For **memory usage**, the additional GPU tensors have the following shapes:
- Each feature: $n \times d$
- Each normalized feature: $n \times d$
- The cosine similarity, cosine distance and the filtered mask: $n \times n$
- The density: $n$

Assuming FP32 (4 bytes) for all tensors, the peak extra memory is at most $4\bigl(4nd + 3n^2 + n\bigr) \text{ bytes}$, which is $561{,}408$ bytes ($\approx 0.5$ MiB) for $n=64$ and $d=512$. This is negligible compared with the model parameters, so the memory overhead can also be safely ignored.

We hope our calculation and explanation address the reviewer's concern. We also add more detailed explanation, motivation, comparison between cosine distance and Euclidean as well as the ablation study about $\beta$ in Appendix D. We welcome the reviewer to check it and raise any further question.

---

### Author Response · Authors · 2025-11-26
**The ablation study of GA ratio $p$ (2)**

**CIFAR-10 (Random Forgetting, 10%, UGradSL+)**

The first row is the retraining results for reference.

|  $p$   |       **UA**       |      **MIA**       |       **RA**       |       **TA**       | **Avg. Gap (↓)** |
| :----: | :----------------: | :----------------: | :----------------: | :----------------: | :--------------: |
|  $-$   |       $8.07$       |      $17.41$       |      $100.00$      |      $91.61$       |       $-$        |
| $0.80$ | $18.84_{\pm 0.71}$ | $26.78_{\pm 1.78}$ | $91.95_{\pm 0.86}$ | $86.05_{\pm 1.21}$ |       8.44       |
| $0.81$ | $17.00_{\pm 0.47}$ | $24.55_{\pm 0.76}$ | $93.21_{\pm 0.21}$ | $87.50_{\pm 0.49}$ |       6.74       |
| $0.82$ | $16.45_{\pm 0.33}$ | $24.22_{\pm 0.46}$ | $93.60_{\pm 0.49}$ | $87.64_{\pm 0.25}$ |       6.39       |
| $0.83$ | $14.45_{\pm 0.73}$ | $21.73_{\pm 0.82}$ | $94.66_{\pm 0.38}$ | $88.59_{\pm 0.19}$ |       4.76       |
| $0.84$ | $13.44_{\pm 0.77}$ | $20.92_{\pm 1.15}$ | $94.67_{\pm 0.71}$ | $88.67_{\pm 0.44}$ |       4.29       |
| $0.85$ | $12.57_{\pm 0.65}$ | $19.18_{\pm 0.64}$ | $95.25_{\pm 1.02}$ | $89.33_{\pm 1.09}$ |       3.32       |
| $0.86$ | $11.42_{\pm 0.14}$ | $18.34_{\pm 0.61}$ | $95.56_{\pm 0.46}$ | $89.49_{\pm 0.27}$ |       2.71       |
| $0.87$ | $10.90_{\pm 0.72}$ | $17.22_{\pm 0.51}$ | $95.79_{\pm 0.39}$ | $89.77_{\pm 0.81}$ |       2.33       |
| $0.88$ | $10.13_{\pm 0.42}$ | $17.85_{\pm 2.11}$ | $95.97_{\pm 0.17}$ | $90.03_{\pm 0.60}$ |       2.28       |
| $0.89$ | $8.98_{\pm 0.29}$  | $14.94_{\pm 0.09}$ | $96.20_{\pm 0.27}$ | $90.23_{\pm 0.33}$ |       2.14       |
| $0.90$ | $8.41_{\pm 0.33}$  | $16.87_{\pm 1.17}$ | $96.53_{\pm 0.03}$ | $90.64_{\pm 0.09}$ |       1.43       |
| $0.91$ | $8.01_{\pm 0.30}$  | $17.33_{\pm 1.17}$ | $96.50_{\pm 0.36}$ | $90.68_{\pm 0.40}$ |       **1.40**       |
| $0.92$ | $7.74_{\pm 0.33}$  | $15.62_{\pm 1.80}$ | $96.28_{\pm 0.26}$ | $90.48_{\pm 0.46}$ |       1.75       |
| $0.93$ | $6.67_{\pm 0.12}$  | $15.93_{\pm 0.22}$ | $96.86_{\pm 0.10}$ | $90.96_{\pm 0.34}$ |       1.67       |
| $0.94$ | $6.79_{\pm 0.71}$  | $16.47_{\pm 0.52}$ | $96.42_{\pm 0.83}$ | $90.74_{\pm 0.45}$ |       1.67       |
| $0.95$ | $6.03_{\pm 0.26}$  | $14.82_{\pm 1.39}$ | $96.76_{\pm 0.41}$ | $90.94_{\pm 0.35}$ |       2.14       |
| $0.96$ | $5.78_{\pm 0.24}$  | $14.79_{\pm 1.14}$ | $96.90_{\pm 0.19}$ | $91.30_{\pm 0.16}$ |       2.08       |
| $0.97$ | $5.98_{\pm 0.49}$  | $14.96_{\pm 0.34}$ | $96.56_{\pm 0.45}$ | $90.81_{\pm 0.53}$ |       2.20       |
| $0.98$ | $6.46_{\pm 0.74}$  | $15.15_{\pm 1.76}$ | $95.52_{\pm 0.67}$ | $90.15_{\pm 0.89}$ |       2.45       |
| $0.99$ | $5.67_{\pm 0.27}$  | $14.40_{\pm 1.18}$ | $96.17_{\pm 0.46}$ | $90.61_{\pm 0.24}$ |       2.56       |

---

### Author Response · Authors · 2025-11-26
**The ablation study of GA ratio $p$  (1)**

Next, we report the performance of GA ratio $p$. We fix $\alpha$ as -0.4. The following tables are the experiment results from UGradSL and UGradSL+, respectively. As shown in Table 19 in Appendix, GA is unstable. The more GA is applied, the more unstable the performance will be. Thus, in the practical usage, we set $p \in [0.9, 0.99]$ to stabilize the performance. Here, we test a further range $[0.8, 0.99]$ to test the robustness. **Our methods provide a relatively stable results, especially in $[0.9, 0.99]$, showing the stability and robustness of our method**.

**CIFAR-10 (Random Forgetting, 10%, UGradSL)**

The first row is the retraining results for reference.

|  $p$   |         UA         |        MIA         |         RA         |         TA         | Avg. Gap (↓) |
| :----: | :----------------: | :----------------: | :----------------: | :----------------: | :----------: |
|  $-$   |       $8.07$       |      $17.41$       |      $100.00$      |      $91.61$       |     $-$      |
| $0.80$ | $12.47_{\pm 1.01}$ | $20.24_{\pm 2.12}$ | $94.11_{\pm 0.71}$ | $88.21_{\pm 0.57}$ |     4.13     |
| $0.81$ | $11.57_{\pm 1.69}$ | $19.68_{\pm 3.31}$ | $94.39_{\pm 1.41}$ | $88.56_{\pm 1.04}$ |     3.79     |
| $0.82$ | $10.61_{\pm 0.08}$ | $17.85_{\pm 0.97}$ | $94.92_{\pm 0.18}$ | $88.94_{\pm 0.37}$ |     2.73     |
| $0.83$ | $9.64_{\pm 0.52}$  | $16.49_{\pm 0.96}$ | $95.54_{\pm 0.71}$ | $89.63_{\pm 0.71}$ |     2.26     |
| $0.84$ | $9.33_{\pm 1.04}$  | $15.84_{\pm 1.17}$ | $95.43_{\pm 0.95}$ | $89.48_{\pm 0.71}$ |     2.38     |
| $0.85$ | $8.27_{\pm 0.82}$  | $14.74_{\pm 1.36}$ | $96.05_{\pm 0.41}$ | $90.08_{\pm 0.40}$ |     2.21     |
| $0.86$ | $7.96_{\pm 0.42}$  | $15.45_{\pm 2.00}$ | $96.10_{\pm 0.44}$ | $90.22_{\pm 0.18}$ |     **1.94**     |
| $0.87$ | $7.51_{\pm 0.26}$  | $15.26_{\pm 3.47}$ | $96.05_{\pm 0.18}$ | $90.20_{\pm 0.51}$ |     2.33     |
| $0.88$ | $6.87_{\pm 0.37}$  | $13.18_{\pm 1.26}$ | $96.43_{\pm 0.35}$ | $90.29_{\pm 0.64}$ |     2.58     |
| $0.89$ | $6.91_{\pm 0.56}$  | $14.44_{\pm 2.46}$ | $96.38_{\pm 0.73}$ | $90.47_{\pm 0.36}$ |     2.22     |
| $0.90$ | $6.92_{\pm 1.08}$  | $13.60_{\pm 3.42}$ | $96.00_{\pm 0.50}$ | $90.26_{\pm 0.14}$ |     2.58     |
| $0.91$ | $6.44_{\pm 1.30}$  | $14.16_{\pm 2.27}$ | $95.93_{\pm 1.18}$ | $90.17_{\pm 0.72}$ |     2.60     |
| $0.92$ | $6.50_{\pm 0.69}$  | $14.35_{\pm 0.72}$ | $95.64_{\pm 0.50}$ | $90.06_{\pm 0.12}$ |     2.64     |
| $0.93$ | $5.88_{\pm 0.82}$  | $14.84_{\pm 1.26}$ | $96.03_{\pm 0.84}$ | $90.31_{\pm 0.54}$ |     2.51     |
| $0.94$ | $5.65_{\pm 0.30}$  | $13.55_{\pm 0.78}$ | $96.25_{\pm 0.44}$ | $90.54_{\pm 0.10}$ |     2.77     |
| $0.95$ | $6.13_{\pm 1.29}$  | $13.14_{\pm 2.43}$ | $95.73_{\pm 1.03}$ | $89.88_{\pm 0.75}$ |     3.05     |
| $0.96$ | $6.07_{\pm 0.91}$  | $14.28_{\pm 2.15}$ | $95.64_{\pm 0.79}$ | $90.15_{\pm 0.36}$ |     2.74     |
| $0.97$ | $5.83_{\pm 1.25}$  | $14.07_{\pm 1.98}$ | $95.20_{\pm 0.98}$ | $89.67_{\pm 0.59}$ |     3.08     |
| $0.98$ | $5.73_{\pm 0.84}$  | $13.19_{\pm 1.99}$ | $95.43_{\pm 0.98}$ | $89.82_{\pm 0.38}$ |     3.23     |
| $0.99$ | $5.83_{\pm 1.05}$  | $12.98_{\pm 1.37}$ | $94.99_{\pm 0.79}$ | $89.46_{\pm 0.59}$ |     3.46     |

---

### Author Response · Authors · 2025-11-26
**The ablation study of the smoothing rate $\alpha$**

We appreciate the reviewers' efforts. Almost every reviewer raise the concern about the sensitivity of smoothing rate $\alpha$ and GA ratio $p$, showing the importance of this study We select the random forgetting on CIFAR-10 and the forgetting set is 10% of the training set. We test both UGradSL and UGradSL+. We fix the GA ratio $p$ as 0.9 and change the smoothing rate $\alpha$. For practical usage, we choose $\alpha \in [-0.99, 0]$ for a stable result. The results are as follows. The first table is UGradSL while the second one is UGradSL+. We can find that there is no sever change of Avg. Gap, showing the stability and robustness to the smoothing rate of our methods.

**CIFAR-10 (Random Forgetting, 10%, UGradSL)**

| **$\alpha$** |      **UA**       |      **MIA**       |       **RA**       |       **TA**       | **Avg. Gap (↓)** |
| :----------: | :---------------: | :----------------: | :----------------: | :----------------: | :--------------: |
|     $-$      |      $8.07$       |      $17.41$       |      $100.00$      |      $91.61$       |       $-$        |
|    $-0.9$    | $8.17_{\pm 1.74}$ | $14.96_{\pm 2.47}$ | $95.81_{\pm 1.97}$ | $89.97_{\pm 1.46}$ |      2.44      |
|    $-0.8$    | $6.98_{\pm 0.47}$ | $13.41_{\pm 0.99}$ | $96.67_{\pm 0.88}$ | $90.75_{\pm 0.22}$ |      2.32      |
|    $-0.7$    | $7.23_{\pm 0.56}$ | $14.33_{\pm 1.23}$ | $96.28_{\pm 0.11}$ | $90.47_{\pm 0.26}$ |      **2.20**      |
|    $-0.6$    | $6.69_{\pm 0.22}$ | $12.93_{\pm 0.67}$ | $96.46_{\pm 0.39}$ | $90.64_{\pm 0.04}$ |      2.59      |
|    $-0.5$    | $6.56_{\pm 0.29}$ | $13.00_{\pm 0.50}$ | $96.58_{\pm 0.23}$ | $90.66_{\pm 0.20}$ |      2.57      |
|    $-0.4$    | $6.92_{\pm 1.08}$ | $13.60_{\pm 3.42}$ | $96.00_{\pm 0.50}$ | $90.26_{\pm 0.14}$ |      2.58      |
|    $-0.3$    | $6.32_{\pm 0.43}$ | $13.63_{\pm 0.67}$ | $96.18_{\pm 0.41}$ | $90.52_{\pm 0.27}$ |      2.61      |
|    $-0.2$    | $6.95_{\pm 0.54}$ | $13.98_{\pm 1.99}$ | $95.41_{\pm 0.68}$ | $89.65_{\pm 0.56}$ |      2.77      |
|    $-0.1$    | $7.13_{\pm 1.44}$ | $14.47_{\pm 1.91}$ | $95.08_{\pm 1.55}$ | $89.57_{\pm 1.04}$ |      2.82      |

**CIFAR-10 (Random Forgetting, 10%, UGradSL+)**

| **$\alpha$** |       **UA**       |      **MIA**       |       **RA**       |       **TA**       | **Avg. Gap (↓)** |
| :----------: | :----------------: | :----------------: | :----------------: | :----------------: | :--------------: |
|     $-$      |       $8.07$       |      $17.41$       |      $100.00$      |      $91.61$       |       $-$        |
|    $-0.9$    | $11.59_{\pm 0.40}$ | $19.41_{\pm 0.59}$ | $95.77_{\pm 0.58}$ | $89.47_{\pm 0.56}$ |      2.97      |
|    $-0.8$    | $10.68_{\pm 0.27}$ | $18.41_{\pm 0.48}$ | $95.94_{\pm 0.24}$ | $89.91_{\pm 0.36}$ |      2.35      |
|    $-0.7$    | $10.12_{\pm 1.01}$ | $16.88_{\pm 0.69}$ | $96.07_{\pm 0.89}$ | $90.08_{\pm 0.78}$ |     2.01      |
|    $-0.6$    | $8.98_{\pm 0.15}$  | $16.29_{\pm 0.87}$ | $96.64_{\pm 0.19}$ | $90.75_{\pm 0.05}$ |      1.56      |
|    $-0.5$    | $9.07_{\pm 0.21}$  | $15.83_{\pm 0.25}$ | $96.65_{\pm 0.34}$ | $90.43_{\pm 0.50}$ |      1.78      |
|    $-0.4$    | $8.41_{\pm 0.33}$  | $16.87_{\pm 1.17}$ | $96.53_{\pm 0.03}$ | $90.64_{\pm 0.09}$ |      **1.43**      |
|    $-0.3$    | $8.59_{\pm 0.07}$  | $16.86_{\pm 2.15}$ | $96.24_{\pm 0.38}$ | $90.20_{\pm 0.15}$ |      1.87      |
|    $-0.2$    | $7.55_{\pm 0.18}$  | $16.68_{\pm 1.60}$ | $96.43_{\pm 0.14}$ | $90.86_{\pm 0.33}$ |      1.47      |
|    $-0.1$    | $7.57_{\pm 0.18}$  | $17.32_{\pm 0.23}$ | $96.15_{\pm 0.38}$ | $90.34_{\pm 0.27}$ |      1.45      |

---

### Author Response · Authors · 2025-11-26
**Summary Response to All ACs and Reviews**

Thank you to all ACs and reviewers for the thorough evaluation of our submission. We appreciate the time and effort you invested in providing detailed feedback, especially given the reviewing workload. Your critiques and questions have helped us strengthen the work.

## Summary of Reviewers

We received 4 reviews with ratings 6 (be9Q), 4 (dpJc), 2 (Xe6w), 6 (Zyp9). Reviewers generally agree that the paper tackles an important problem in efficient machine unlearning with a simple, pluggable gradient-based method, but raise concerns about missing ablations, efficiency analysis, and the practical interpretation of our theoretical guarantees.

## Strengths Acknowledged by Reviewers
- The problem framing is clear, and the motivation for stabilizing gradient ascent with label smoothing is sound. (be9Q)
- The method UGradSL/UGradSL+ is simple, plug-and-play, and easily integrated into existing GA/FT unlearning pipelines. (be9Q, dpJc, Zyp9)
- The paper is logically organized and easy to follow, with clear tables and figures. (dpJc)
- Theoretical analysis explains when NLS improves gradient ascent and connects label smoothing to Label-LDP. (be9Q, dpJc, Zyp9)
- Experiments span multiple datasets, architectures, and unlearning types (class, random, group), showing improved unlearning–retaining trade-offs. (be9Q, Xe6w, Zyp9)

## Main Concerns Raised
- The paper lacks ablation and sensitivity studies on forgetting set size for random forgetting and key parameters such as GA ratio $p$ and the smoothing rate $\alpha$. (be9Q, dpJc, Xe6w, Zyp9)
- The self-adaptive UGradSL+ requires distance computation within mini-batches, raising concerns about time and memory overhead and whether the gains justify the extra cost. (dpJc, Xe6w)

## Our Response Efforts and Final Summary

**For Reviewer be9Q**
- We add ablation studies on forgetting set size, $p$, and the smoothing rate $\alpha$. (W1, Q1, W2, Q2, W3, Q3)
- We provide a more detailed explanation of the design and motivation of self-adaptive UGradSL+. (Q4)
- We demonstrate integrating UGradSL into $\ell_1$-sparse to highlight its plug-and-play nature.

**For Reviewer dpJc**
- We include the suggested recent baseline methods in all forgetting scenarios (class-wise, random, and group-wise) on multiple datasets, including CIFAR-100, CIFAR-20, CelebA, TinyImageNet, and ImageNet. (W1, Q1)
- We thoroughly check and standardize notation and definitions, and expand the explanation of self-adaptive UGradSL for better algorithmic clarity. (W2, Q2)
- We analyze the generalization gap between TA and RA to clarify overfitting behavior on the retain set. (Q3)
- We add ablation studies on GA ratio $p$, and the smoothing rate $\alpha$. (W3)
- We report the peak memory usage of our methods for different model and datasets (Q4)
- We report GPU usage and provide a more complete time and memory analysis for the smoothing-rate and density computations. (W4, Q5)

**For Reviewer Xe6w**
- We clarify the sign usage in Theorem 1 and the roles of $\gamma_1$ and $\gamma_2$. (W1)
- We analyze the computation cost of the distance calculation in UGradSL+. (W2)
- We provide a clearer description of the differences between UGradSL and UGradSL+. (W3)
- We add ablation studies on the GA ratio p and the smoothing ratio \alpha. (W4)

**For Reviewer Zyp9**
- We extend the empirical validation of the theorem’s condition to more datasets (CIFAR-10, CIFAR-100, ImageNet) and models (ResNet-18, VGG-16, ViT). (W1)
- We clarify how the label-level DP guarantee relates to model performance in deep learning–based unlearning methods, and its limitations for model-level guarantees. (W2)
- We provide ablation studies on the smoothing rate $\alpha$ and the distance threshold $\beta$. (W3)
- We describe the self-adaptive smoothing rate calculation more clearly. (Q1)

All these modifications are incorporated into the revised paper (marked in red). We appreciate the reviewers’ insightful comments and are happy to address any further concerns.

---

### Meta-Review · Area_Chair_fMgm · 2026-01-05

**Summary:**

This paper investigates the role of label smoothing in gradient-based machine unlearning and proposes UGradSL, a plug-and-play method that applies negative label smoothing to forgetting data while performing gradient descent on retained data. The reviewers initially gave mixed ratings (6, 6, 4, 2), with the lowest-scoring reviewer raising concerns about: (1) unclear technical presentation and approximation error bounds, and (2) significantly higher runtime of the adaptive variant UGradSL+ compared to strong baselines. In the rebuttal, the authors provided additional theoretical analysis and detailed efficiency breakdowns, which led the most critical reviewer (Xe6w) to explicitly state they would raise their score from 2 to 4, although they could not locate the edit button to update the rating. Based on these considerations, the paper is recommended for acceptance.

**Reviewer Concerns:**

Addressed: ablation completeness, recent baselines, runtime/memory clarification, notation.

Still outstanding: approximation-error quantification, broader validation of theorem conditions, certified-removal guarantee.

**Reviewer Scores:**

N/A

---

### Decision · Program_Chairs · 2026-01-26

Accept (Poster)